



# Intercomparison of low and high resolution infrared spectrometers for ground-based solar remote sensing measurements of total column concentrations of CO$_2$, CH$_4$ and CO

Mahesh Kumar Sha[1], Martine De Mazière[1], Justus Notholt[2], Thomas Blumenstock[3], Huilin Chen[4], Angelika Dehn[5], David W T Griffith[6], Frank Hase[3], Pauli Heikkinen[7], Christian Hermans[1], Alex Hoffmann[8], Marko Huebner[8], Nicholas Jones[6], Rigel Kivi[7], Bavo Langerock[1], Christof Petri[2], Francis Scolas[1], Qiansi Tu[3], and Damien Weidmann[8]

[1]Royal Belgian Institute for Space Aeronomy (BIRA-IASB), Brussels, Belgium
[2]Institute of Environmental Physics, University of Bremen, Bremen, Germany
[3]Karlsruhe Institute of Technology, IMK-ASF, Karlsruhe, Germany
[4]Centre for Isotope Research, University of Groningen, Groningen, The Netherlands
[5]European Space Agency, ESA/ESRIN
[6]University of Wollongong, Wollongong, Australia
[7]Finnish Meteorological Institute, Sodankylä, Finland
[8]Rutherford Appleton Laboratory, United Kingdom

**Correspondence:** Mahesh Kumar Sha (mahesh.sha@aeronomie.be)

**Abstract.** The Total Carbon Column Observing Network (TCCON) has been the baseline network of instruments that record solar absorption spectra from which accurate and precise column-averaged dry air mole fractions of CO$_2$ (XCO$_2$), CH$_4$ (XCH$_4$), CO (XCO) and other gases are retrieved. The TCCON data have been widely used for carbon cycle science and validation of satellites measuring greenhouse gas concentrations globally. The number of stations in the network (currently about 25) is limited and the stations are distributed mostly in Northern America, Europe, Japan and Oceania leaving gaps in the global coverage. A denser distribution of ground-based solar absorption measurements is needed to cover various atmospheric conditions (humid, dry, polluted, presence of aerosol), various surface conditions (high and low albedo) and a larger latitudinal distribution. More stations in the southern hemisphere are also needed but a further expansion of the network is limited by its costs and logistical requirements. For this reason several groups are investigating supplemental portable low-cost instruments. The European Space Agency (ESA) funded campaign Fiducial Reference Measurements for Ground-Based Infrared Greenhouse Gas Observations (FRM4GHG) at the Sodankylä TCCON site in northern Finland aims at characterising the assessment of several low-cost portable instruments for precise solar absorption measurements of XCO$_2$, XCH$_4$ and XCO. The test instruments under investigation are three Fourier transform spectrometers (FTS): a Bruker EM27/SUN, a Bruker IRcube and a Bruker Vertex70; as well as a Laser Heterodyne spectro-Radiometer (LHR) developed by the UK Rutherford Appleton Laboratory. All four remote sensing instruments performed measurements simultaneously next to the reference TCCON instrument, a Bruker IFS 125HR, for a full year in 2017. The TCCON FTS was operated in its normal high-resolution mode (TCCON data set) and in a special low-resolution mode (HR125LR data set), similar to the portable spectrometers. The remote sensing measurements have been complemented by regular AirCore launches performed from the same site. They provide



in-situ vertical profiles of the target gas concentrations as auxiliary reference data for the column retrievals which is traceable

to the WMO SI standards. The timeseries, the bias relative to the reference instrument and its scatter and the seasonal and the day-to-day variations of the target gases are shown and discussed. The comparisons with the HR125LR data set gave useful analysis of the resolution dependent effects on the target gas retrieval. The solar zenith angle dependence of the retrievals is shown and discussed. The reference measurements performed with the Bruker IFS 125HR (TCCON and HR125LR data sets) were found to be affected by non-linearity. A non-linearity correction of the TCCON data was performed and compared with

the test instruments and AirCore. The non-linearity corrected TCCON data show a better match with the test instruments and AirCore data as compared to the reference TCCON data. The intercomparison results show that the LHR data have a large scatter and biases with a strong diurnal variation relative to the TCCON and other FTS instruments. The LHR is a new instrument under development and these biases are being currently investigated and addressed. The campaign helped to characterise and identify the instrumental biases and possibly retrieval biases which are currently under investigation. Further improvements of

the instrument are ongoing. The EM27/SUN, the IRcube, the modified Vertex70 and the HR125LR provided stable and precise measurements of the target gases during the campaign with quantified small biases. The bias dependence on the humidity along the measurement line-of-sight has been investigated and no dependence was found. These three portable low-resolution FTS instruments are suitable to be used for campaign deployment or long-term measurements from any site and offer the ability to complement the TCCON and expand the global coverage of ground-based reference measurements of the target gases.

## 1  Introduction

Carbon dioxide ($CO_2$) and methane ($CH_4$) are the two main components of the carbon cycle of the earth's atmosphere. They absorb and retain the heat in the atmosphere causing the greenhouse effect and global warming. $CH_4$ has a global warming potential of about 28 times greater than $CO_2$ over a 100 year time period. It however exists in much lower concentrations and has a significantly shorter lifetime compared to $CO_2$. $CH_4$ also plays an important role in atmospheric chemistry by reacting with

hydroxyl radicals (OH), thereby reducing the oxidation capacity of the atmosphere and producing ozone (Kirschke et al., 2013). The atmospheric concentration of both these gases has been steadily increasing in the recent years caused by anthropogenic activities (Stocker et al., 2013; Dlugokencky and Tans). The third gas focused on is carbon monoxide (CO). It is a poisonous reactive gas considered principally as a man-made pollutant. It plays an important role in atmospheric chemistry by reacting with the atmospheric oxidants, ozone ($O_3$), hydroperoxy ($HO_2$) and hydroxyl radicals (OH). The lifetime of CO ranges from

weeks to months (Novelli et al., 1998). An increase of CO would imply that more OH will be lost through chemical reaction with CO and that less OH will be available for reaction with $CH_4$. Therefore CO has an indirect but important influence in determining the chemical composition and radiative properties of the atmosphere. The emissions of CO are virtually certain to have a positive radiative forcing. It is therefore considered as an indirect greenhouse gas (Stocker et al., 2013). Continuous monitoring of precise and accurate measurements of these gases is of utmost importance to determine their sources and sinks,

and trends. Currently this is one of the major challenges within climate research which will help in understanding the carbon cycle.



Atmospheric measurements of $CO_2$, $CH_4$ and CO have been performed by in-situ surface based networks for many decades. These have been complemented by sparse aircraft measurement campaigns providing important additional measurements. However, both these measurement types have been performed at only a few locations and the atmosphere sampled non-uniformly. In recent years, satellite based remote sensing measurements have been able to provide global coverage of these gases. The nadir looking satellites detecting scattered sunlight in the near-infrared (NIR) spectral region provide the most powerful method for global mapping of these gases. These measurements cover the whole atmospheric column providing the total column concentrations of the trace gases of interest and add important measurements to the in-situ networks. However satellite measurements require accurate validation. These accurate reference measurements can be performed from surface based, air-borne (e.g., balloon or aircraft) or already validated satellites. To ensure equal dependency on the input spectral data, the best validation method is to use the total column amounts of the trace gases calculated from the solar absorption measurements performed from the surface and the satellite in the same spectral region. Moreover, the total column observations are much less sensitive to boundary layer effects compared to the in-situ surface measurements.

The current state-of-the-art validation system for greenhouse gases (GHGs) is the Total Carbon Column Observing Network (TCCON). TCCON is a network of ground-based Fourier transform spectrometers (FTS), of the type Bruker IFS 125HR, that record solar absorption spectra in the NIR spectral range to retrieve accurate and precise column-averaged abundances of atmospheric constituents including $CO_2$, $CH_4$ and CO amongst other species (Wunch et al., 2011). There are currently about 25 TCCON stations distributed globally which form the backbone of the validation data set for the GHG measuring satellites (e.g., GOSAT, OCO-2, Sentinel-5 Precursor, ...) and model comparisons (Inoue et al., 2016; Wunch et al., 2016; Borsdorff et al., 2018; Kivimäki et al., 2019; Ostler et al., 2016; Jing et al., 2018; Kong et al., 2019). The distribution of the TCCON stations currently lacks global coverage with a majority of its stations located in Northern America, Europe and Japan, and currently only five stations in the southern hemisphere. The lack of stations close to important source areas and the limited number of stations in general is unable to resolve global GHG gradients. Furthermore, for the complete validation of the satellite data set, measurements covering a wide range of conditions (e.g., high and very low surface albedo, pollution, aerosol presence) are missing within the current set up of the TCCON network.

An extension of the TCCON network is limited by high start-up, maintenance and operational costs, as well as difficulties of campaign based transportability. The maintenance of the instrument requires skill and experience. All these factors resulted in the development of a number of cheap and easily deployable instruments for remote sensing measurements of greenhouse gases, mainly driven by scientific research institutes in collaboration with industrial partners. Some of these instruments have been in operation for several years. However, there has been little characterisation, intercomparison and harmonisation of these new instruments in comparison to the standard instrument used in TCCON. These comparisons however are mandatory for using these individual data sets independently for science.

For this reason in 2017 an intercomparison campaign, initiated by the European Space Agency (ESA) within the project Fiducial Reference Measurements for Ground-Based Infrared Greenhouse Gas observation (FRM4GHG), was performed in Sodankylä (Finland) to assess different spectrometric instruments for remote sensing of atmospheric trace gases as to their performances regarding precise measurements of column-averaged dry-air volume mole fractions of $CO_2$, $CH_4$ and CO. The





instruments were deployed at the meteorological observatory Sodankylä where measurements took place between March and October 2017. The remote sensing measurements were complemented by regular AirCore launches from the same site. Air-Core measurements provide vertical profiles of the target gas concentrations as auxiliary reference data for the column mea-

surements. The performances of the instruments were compared between themselves and to a reference TCCON instrument. The goal of this campaign was the characterisation of less expensive and more portable FTSs to complement TCCON for the establishment of a wider and denser network.

This paper is organised as follows: Section 2 gives a description of the campaign site, lists the details of the instruments taking part in the campaign and their evolution. Section 3 gives a description of the measurement strategy that was used to

ensure comparable observations. Section 4 gives the description of the data and its availability. Section 5 gives the campaign results showing the intercomparison between the test instruments with respect to the reference TCCON and low-resolution measurements performed by the TCCON instrument. It also gives the intercomparison results between the TCCON and AirCore data and the results using the AirCore profile as a priori for the FTS retrievals. Section 6 concludes the paper by giving a summary of the results.

## 100  2  Measurements at Sodankylä and campaign instrumentation

### 2.1  Description of the campaign site

The Finnish Meteorological Institute (FMI) Sodankylä facility was selected as the campaign site as it fulfilled all selection criteria: (I) availability of TCCON measurements at the site, (II) possibility to launch, retrieve and analyse AirCore, (III) infrastructure to host all participating instruments and (IV) local support by scientists / engineers in case of problems occurring

with the instruments during the campaign. The Sodankylä facility is located above the Arctic Circle in northern Finland, 67.3668° N, 26.6310° E, 188 m.a.s.l about 6 km south of Sodankylä. The site is equipped with a stratospheric balloon launch facility. The AirCore system has been operated by FMI to perform regular balloon launches since early September 2013. AirCore and other balloon payloads can be launched within 200 meters from the TCCON instrument. In addition, the site also has a mobile system to launch payloads from an upwind site in order to retrieve them in the vicinity of the TCCON site. Upon its

recovery, the analysis of the AirCore is done on site using a Picarro G2401 analyser. Continuous surface in-situ measurements of $CO_2$, $CH_4$ and CO are performed from a 50 meter tower located at 500 meters away from the TCCON instrument. Further details on the site can be found in Kivi and Heikkinen (2016). An air-conditioned laboratory container ($\sim$ 9.1 meter long) was set up for the deployment of the visiting instruments for the campaign. The laboratory was placed about 30 meters south from the building hosting the TCCON instrument.

### 115  2.2  Instruments

The TCCON spectrometer, a Bruker IFS 125HR, was the main reference instrument for this campaign. Four low resolution portable instruments participated in the campaign: a Bruker EM27/SUN, a Bruker Vertex70, a Bruker IRcube, and a homemade





Laser Heterodyne spectro-Radiometer (LHR). Each of the three Bruker low resolution instruments is based on a RockSolid$^{TM}$ corner-cube pendulum interferometer. This allows for comparable sampling quality and robustness amongst the instruments. However, the instruments differ in the use of the surrounding imaging optics and their geometric arrangement which defines the interferometric field-of-view (FOV) and thus determines the instrumental line shape (ILS) of the respective instrument. The position of the centre burst which determines the resolution differs for each instrument. The EM27/SUN records double sided and the IRcube single sided interferograms yielding a maximum resolution of $0.5 \, cm^{-1}$. The Vertex70 records single sided interferograms giving a maximum resolution of $0.16 \, cm^{-1}$. The number of usable detector positions differs for the three instruments. All instruments used solar trackers with an active feedback loop to track the sun with an accuracy better than 0.1 mrad either with the help of active quadrant diodes or by active camera positioning. All low-resolution test instruments have the advantage that they do not need to be disassembled for transport. A detailed description of the instruments is given in the following sub-sections and some of the key features of the instruments, measurement properties and retrieval strategies during the campaign are listed in Tables 1 & 2.

### 2.2.1 Bruker IFS 125HR

The instrumental and operational setting of the Bruker IFS 125HR in the TCCON mode of operation can be found in detail in Kivi and Heikkinen (2016). The TCCON instrument's operation, maintenance and data analysis was performed by FMI. The measurements were performed at a spectral resolution of $0.02 \, cm^{-1}$ in vacuum (<1 hPa) to improve the stability and to reduce water vapour in the system. They were recorded using room-temperature (RT) Indium Gallium Arsenide (InGaAs) and RT Silicon (Si) detectors. The recorded signal (interferogram) was stored in DC mode in order to make corrections for the solar intensity variations. The interferogram upon DC correction was then Fourier transformed to get the corresponding spectrum. Column abundances of $CO_2$, $CH_4$, CO, $N_2O$, $H_2O$, HDO, $O_2$ and HF were retrieved from the spectra based on the TCCON GFIT retrieval code (Wunch et al., 2015). The instrument was also equipped with a liquid nitrogen (LN2) cooled Indium Antimonide (InSb) detector. This detector enhances the possibilities to expand the wavelength region covered by the instrument (see Table 2) and to retrieve more atmospheric species. In addition to the TCCON and InSb measurements, the instrument was also used to record double-sided DC coupled interferograms at $0.5 \, cm^{-1}$ using the InGaAs detector. These measurements are henceforth called HR125LR. These measurements provide low-resolution data sets performed with the same TCCON instrument to be compared to the results of the other tested low-resolution instruments. The sequence of measurements was that first one InGaAs / Si forward-backward scan (standard TCCON measurement) was recorded, then two forward-backward HR125LR scans, after that again one standard TCCON measurement and two forward-backward HR125LR scans followed by one InSb forward-backward scan. This cycle was repeated for the whole measurement day. This paper focuses on the measurements performed with only the InGaAs detector (standard TCCON and HR125LR data sets). The instrument was operated in an automated way with the possibility of manual intervention. The ILS characterisation was performed using a HCl (hydrogen chloride) gas cell following the recommendations of TCCON (Hase et al., 2013) using the LINEFIT software (Hase et al., 1999).





### 2.2.2 Bruker EM27/SUN

The EM27/SUN spectrometer has been developed by Karlsruhe Institute of Technology (KIT) in cooperation with Bruker starting in 2011 (Gisi et al., 2012). The spectrometer is available as commercial item from Bruker since 2014, an additional channel for CO detection has been assigned in 2016 (Hase et al., 2016). Today already more than 40 units are operated by working groups around the globe (Frey et al., 2019). The EM27/SUN used during the campaign was provided by KIT. The EM27/SUN records double-sided DC coupled interferograms making an average of 10 scans in about 58 sec at a spectral resolution of $0.5$ cm$^{-1}$. A double-sided recording of the interferograms largely reduces the sensitivity to residual phase error. The measurements were performed using a RT InGaAs detector (5500–11000 cm$^{-1}$) and a DC-coupled wavelength extended RT InGaAs detector (4000–5500 cm$^{-1}$) (Hase et al., 2016). In this extended configuration, the EM27/SUN covers the same spectral region as TCCON and encompasses the spectral section as observed by TROPOspheric Monitoring Instrument (TROPOMI) (TRO, a, b). Spectra were generated from raw interferograms using the preprocessor tool developed by KIT in the framework of the COCCON-PROCEEDS project funded by ESA. Column abundances of $CO_2$, $CH_4$, CO, $H_2O$, and $O_2$ were retrieved from the resulting spectra using the PROFFAST retrieval code. The $XCO_2$ and $XCH_4$ products are bias-corrected based on the extensive COCCON development. The bias correction is only done for the EM27/SUN and not for any other test data sets. The codes are open source and licence free and can be downloaded from the KIT webpage http://www.imk-asf.kit.edu/english/3225.php. The characterisation of the ILS was performed using an open path measurement as described in Frey et al. (2015). The solar tracker of the EM27/SUN is attached to the body of the spectrometer. It was operated outside the FRM4GHG laboratory container at ambient conditions for the whole campaign period. This mode of deployment showed the capability of the instrument to be operated even under harsh campaign conditions. The day-to-day instrument operation was performed by KIT with local support from FMI for some measurement days. Once deployed, the instrument operation is automated. The EM27/SUN was supported by a pressure sensor and a GPS sensor for accurate timekeeping and position acquisition.

### 2.2.3 Bruker Vertex70

The Vertex70 spectrometer was purchased from Bruker to take part in the campaign. It records single-sided DC coupled interferograms making an average of 2 scans in about 17.3 sec at a spectral resolution of $0.2$ cm$^{-1}$. The intensity of the interferogram varies during the scan and the incident angle on the two interferometer mirrors of the pendulum changes during the scan due to the large optical path covered by the pendulum drive, leading to self-apodisation. Both these factors were taken into account while performing the retrieval. Several scans were co-added for one measurement ($\sim 2.5$ min) with a comparable signal-to-noise ratio (SNR) to the reference TCCON measurements. The Vertex70 has the advantage of accommodating and measuring with two detectors covering a wide spectral range. An extended RT InGaAs detector (3500–15000 cm$^{-1}$) and a LN2 cooled InSb detector (2500–10000 cm$^{-1}$) were used. This paper focuses on the measurements performed with only the InGaAs detector. The GFIT retrieval code was used to analyse the measured spectra and retrieve column abundances of $CO_2$, $CH_4$, CO, $H_2O$, and $O_2$. The characterisation of the ILS was performed using a HCl gas cell similar to TCCON. The Vertex70 was operated from inside the dedicated FRM4GHG air-conditioned laboratory container regulated at about $20\,^\circ$C with the solar



beam being fed to the instrument using a homemade BIRA-IASB solar tracker mounted on top of the container. The distance between the solar tracker and the spectrometer was 3 m. The tracking of the sun was performed using a camera-based active feedback option. The instrument operation was automated using the BARCOS system (Neefs et al., 2007) and a homemade automated control unit system build by BIRA-IASB with the possibility of a manual intervention at any time. The solar tracker was equipped with sun intensity and rain detection sensors which facilitated the automatic opening and closing of the solar tracker cover depending on the weather conditions. This facilitated performing atmospheric measurements on every occasion with good weather conditions. The data analysis was performed by University of Bremen and the maintenance was shared between BIRA-IASB and University of Bremen.

### 2.2.4 Bruker IRcube

The IRcube is a compact portable FTS manufactured by Bruker Optics. It records single-sided DC coupled interferograms using a RT extended InGaAs detector ($4500$–$15000$ cm$^{-1}$) making an average of 33 scans (17 forward and 16 backward) in about 1.7 mins at a spectral resolution of $0.5$ cm$^{-1}$. It has an internal full angle FOV of 72 mrad. The novel design of the IRcube for this field campaign was the use of a fibre-optic (FO) feed from an independent solar tracker (a STR-21G, Eko instruments Ltd of Japan) mounted on top of the FRM4GHG laboratory container to receive the solar beam. A 50 cm focal length F/5 telescope (glass lens) focuses the solar beam onto a 20 m long, 600 $\mu$m core fibre with a numerical aperture (NA) of 0.22. This defines the external FOV on the solar disk at 1.2 mrad. The coupling of light from the optical fibre to the IRcube was chosen to optically match the input optics of the IRcube as closely as possible; coupling the power from the FO to the spectrometer so that the signal-to-noise ratio is comparable to TCCON, while avoiding unwanted spectral features that are present in NIR optical fibres. There is a limited range of NAs commercially available, out of which the best compromise for the IRcube with good spectral characteristics was the low OH Thorlabs FG550LEC. A glass lens and aperture in front of the IRcube refocuses the solar beam from the fibre into the entrance aperture (0.5 mm). A small part of the main beam reflected from the CaF$_2$ entrance window was used to monitor the solar radiation for cloud filtering. The IRcube can be housed anywhere within the length of the fibre-optic cable (here 20 m). This design concept is of significant importance for certain applications where the spectrometer can be placed far away from the solar tracker e.g., inside a weather proof enclosure. During this campaign the IRcube was set up by the University of Wollongong inside the FRM4GHG container and the operation of both tracker and IRcube was automatic. The characterisation of the ILS was performed using an open path measurement similar to the procedures followed by the EM27/SUN. The data analysis was performed by the University of Wollongong using the GFIT retrieval code.

### 2.2.5 Laser Heterodyne spectro-Radiometer (LHR)

The LHR is a research instrument developed by the Spectroscopy Group of the Space Science and Technology Department of the Rutherford Appleton laboratory (RAL) (Weidmann et al., 2007; Tsai et al., 2012; Hoffmann et al., 2016). The principle of operation is similar to that of a heterodyne radio-receiver; however the LHR operates in the mid-infrared region of the spectrum. The benefits of such an approach to spectroscopy include: (I) High spectral resolution (up to >500,000 resolving power), (II)



ideally shot noise limited radiometric noise, (III) intrinsic narrow FOV and (IV) scalable down to ultra-miniaturised packages through optical integration.

Compared to the laboratory instrument reported in (Hoffmann et al., 2016), the LHR was re-engineered to the requirements of the FRM4GHG campaign with the following modifications: I) The optical path was reworked to bring the instrument package down to 40x40x20 cm$^3$. II) A secondary laser channel (to be equipped in future) was integrated. III) A thermoelectrically cooled Mercury Cadmium Telluride (HgCdTe) photodiode for photomixing was installed to avoid LN2 usage. IV) A solar disk imager was installed for FOV monitoring and optional solar tracking operations. V) Acquisition, instrument control hardware and software were integrated to allow full unattended operation, except for switch on and off procedures.

The LHR was installed inside the FRM4GHG container and operated under ambient conditions. The incoming solar beam had a 12 mm diameter and was side-sampled from the BIRA-IASB solar tracker. The LHR has no entrance window. Inside the instrument, the incoming beam is split into a transmitted mid-infrared component for heterodyning and a visible component for solar imaging. To that end, a Germanium (Ge) long-wave infrared bandpass filter is used. To carry out the fine spectral analysis, the incoming mid IR field is superimposed with that of an optical local oscillator (LO) by a Zinc Selenide (ZnSe) beamsplitter. The LO consists of a continuously tunable semiconductor laser source, in this case a quantum cascade laser, operating in the narrow spectral range between 952–955 cm$^{-1}$ ($v_1 \leftarrow v_3$ $CO_2$ band) optimised through prior analysis to optimise atmospheric state retrieval information. The spectra were resolved through LO continuous frequency tuning. The superimposed atmospheric beam and local oscillator one are mixed onto the high speed photodiode, effectively transposing the middle infrared spectral information into the radio-frequency (RF) domain. The spectral resolution is determined by electronic filters. For the FRM4GHG campaign, the spectral resolution was set to 0.02 cm$^{-1}$. Each spectrum was recorded over 30 s. The start and stop operation of the LHR was performed manually by the local support staff at the measurement site. A typical atmospheric spectrum showing the $CO_2$ window as measured by the LHR can be seen in Fig. 6 in Hoffmann et al. (2016). The data analysis was performed by the RAL team using the optimum estimation atmospheric retrieval method, in which the Reference Forward Model was used (Dudhia, 2017).

### 2.2.6 AirCore

The AirCore is a novel innovative technique to sample high-altitude profiles of atmospheric concentrations of trace gases. A detailed description of the technique can be found in Karion et al. (2010). The AirCore system used for this campaign was originally built by the University of Groningen (RUG) and was further developed together with the Finnish Meteorological Institute (FMI). The total length of the AirCore is 100 m. It consists of two types of stainless steel tubing with outer diameters of 1/4" and 1/8", respectively. The vertical resolution of measurements from the AirCore is 13.4 mbar for an ambient pressure higher than 232 mbar, and 3.9 mbar for an ambient pressure lower than 232 mbar. A custom-made data logger by FMI was used to record temperature and ambient pressure of the AirCore tubing. An automatic valve was developed and installed prior to the campaign, which closed the inlet valve of the AirCore system upon landing. The AirCore was packed in a styrofoam box to protect it from damage during landing, with its inlet valve protruding through the styrofoam box. Magnesium perchlorate ($Mg(ClO_4)_2$) was used as dryer in the AirCore. The AirCore package includes tubing, connectors, valves, data logger and a





box. The air volume of the AirCore is approximately 1400 ml. The AirCore was launched hanging on a 3000 g meteorological balloon (Totex TX3000). The payload included a Vaisala RS92-SGPL radiosonde (Dirksen et al., 2014), an Iridium and GPS/GSM positioning device and a light-weight transponder. The balloon burst after reaching the ceiling height (typically about 30–35 km). A large parachute was used to slow down the descent speed of the AirCore while a tracking system located its position. Upon landing, the AirCore was recovered and brought to the laboratory to obtain mole fractions of $CO_2$, $CH_4$

and CO with a Picarro G2401-m Cavity Ring-Down Spectrometer (CRDS). The precision/accuracy of the $CO_2$, $CH_4$ and CO are 0.05 ppm/0.1 ppm, 0.5 ppb/1 ppb and 8 ppb/3 ppb, respectively. An orifice (Sapphire, Type A, size 0.18 mm) was placed between the pump and the analyser to achieve a constant flow of 40 ml/min. The sample was analysed starting from the stratospheric part (the closed end) to minimise the diffusion. Before each flight, the AirCore was flushed with dry air from a fill cylinder for several hours. This procedure dries the inner surface of the AirCore and fills it with air of known mole fractions.

The mole fraction of CO in the fill cylinder was ∼12000 ppb. The fill air was used as an indicator of air mixing and as a diagnostic tool. Radiosonde (Vaisala RS92-SGPL) ambient pressure, temperature and AirCore temperature were available for each AirCore flight. AirCore vertical profiles were retrieved based on the measured time series of mole fractions and the recorded in-flight information, e.g., coil temperature, ambient pressure and ambient altitude using a custom made retrieval software by RUG.

### 2.2.7 In-situ

The in-situ measurements used for this work were provided by the FMI. The concentration of $CO_2$, $CH_4$ and CO were measured on a 50 meter tower at 3 levels (2 m, 22 m and 48 m) above surface using a Picarro G2401 system. More information about the site can be found on the webpage at http://fmiarc.fmi.fi/index.php.

## 3  Description of the measurement strategy to ensure comparable observations

### 3.1  Measurement set-up

The campaign took place between March and October 2017. The site is located at high latitude therefore it was not possible to measure beyond this period due to the high solar zenith angle (SZA). Solar measurements were recorded between sunrise and sunset, depending on the SZA limits set by the local scene and weather conditions (cloud, fog and strong winds). The FMI team monitored the operation of the instruments during the campaign period. Depending on the weather conditions, all

spectrometers performed as many measurements as possible to improve the measurement statistics. The continuous operation in the presence of the sun and good weather condition helped to observe diurnal variation of the target gases. The campaign began with an initial blind intercomparison phase where the instruments were operated with the optimised settings best known to their PIs to get a good SNR comparable to the TCCON instrument. The measurements performed by the different remote sensing instruments were submitted to the chosen referee BIRA-IASB.





The intercomparison study of the blind phase showed that the Vertex70 instrument needed a modification. The aperture was reduced such that the beam diameter changed from 40 mm to 20 mm, reducing the intensity of the light reaching the detector. This helped to reduce the scatter in the retrieved column values by ensuring the operation of the instrument in the linear region of the detector.

    The IRcube did not have to undergo any internal modifications however a broken optical fibre had to be replaced in

April 2017. The first FO used for the IRcube was an ultra-low OH silica optical fibre from Polymicron Technologies, part FIA8008801100 with a NA of 0.22 and a core diameter of 800 $\mu$m. Due to a long delivery time of this FO, a replacement FO as discussed in section 2.2.4 was ordered and used after April 2017.

    The EM27/SUN was operated without any modifications during the whole campaign period. The exact dates of all performed modifications are shown in Table 3.

A total of 10 AirCore launches were performed during the campaign and these were used as an in-situ reference data set to better understand the intercomparison of the remote sensing data. Further details are discussed in section 5.2.

### 3.2   Instrument characterisation

All teams performed full functionality test of their respective instruments and accessories before shipping and upon arrival at the campaign site in Sodankylä. The functionality test included quality checks as well as performing ILS measurements of the

instruments. These measurements serve as reference to check the effects (if any) of transport on the instrumental properties and to ensure nominal operation in case of new set ups. During the campaign all teams performed ILS measurements when possible to monitor the long-term stability of the participating instruments. The modulation efficiency (ME) of the TCCON instrument at the highest OPD was <1.02 with a phase error (PE) in the range of $\pm 2$ mrad throughout the year. The ME of the EM27/SUN at the highest OPD was about 1.02 with a PE in the range between -3 mrad and 1 mrad throughout the year. The

ME of the Vertex70 before shipping and upon arrival at the Sodankylä site was about 0.935 at 4.5 cm OPD and the PE was changing between -16 and -36 mrad. The ME improved significantly from 0.935 to about 0.973 and the PE improved to about -13 mrad after the modification of the Vertex70 with the introduction of the additional aperture. The IRcube has a ME of about 0.95 with the PE in the range between -5 and +1.5 mrad. A summary of the ILS properties of the FTS is given in Table 3. The ILS of the LHR was determined by the radio frequency (RF) filter characteristics used to limit the detector bandwidth and

hence the spectral resolution of the instrument and is therefore an inherent property of the instrument. A detailed description of the ILS validation of the LHR with $C_2H_4$ gas cell measurements can be found in a technical document by Hoffmann et al. (2017). None of the instruments showed any sign of degradation of the instrumental properties during the whole campaign.

### 4   Data description

The raw measurements (level 0 data) from all participating remote sensing instruments are made publicly available with the

DOI https://doi.org/10.18758/71021040 (Sha et al., 2018). The atmospheric concentration of the trace gases (level 2 data) together with the auxiliary data are made publicly available with the DOI https://doi.org/10.18758/71021048 (Sha et al., 2019).





All data sets and the documentations are also made publicly available via the project webpage (http://frm4ghg.aeronomie.be) as well as via the ESA Atmospheric Validation Data Centre (EVDC).

## 5 Campaign results

### 5.1 Intercomparison data

Sodankylä is located within the Arctic Circle therefore solar measurements with sufficiently low SZA are only possible from the beginning of March to the end of October. During the month of September and October we had mostly overcast sky. Only three days of measurements were possible with the TCCON instrument during the period between the middle of September and the end of October. These measurements however have to be filtered out from the intercomparison study as they were measured at SZA > 75°.

Based on the measurement capabilities by the individual instruments, the groups were asked to provide some or preferably all of the following parameters: Measurement day/time; ground pressure; total column amounts of $O_2$, $H_2O$, $CO_2$, $CH_4$, $CO$; and column averaged dry air mole fraction of the gas (Xgas) values for $XCO_2$, $XCH_4$, $XCO$. Xgas is defined by the following equation:

$$Xgas = \frac{gas_{column,dry}}{O_{2,column,dry}} \times 0.2095 \tag{1}$$

where 0.2095 is the dry air $O_2$ mole fraction.

For the FTIR instruments also Xair was submitted, which is a measure of the instrument's performance. Xair is calculated from the scaled ratio of the surface pressure to the $O_2$ column amount analogous to equation 1 described in Deutscher et al. (2010) with the value of 1.0 for a perfect measurement and a perfect pressure measurement. A summary of the data sets and the corresponding retrieval methods is provided in Table 2. The Xgas values which were calculated using GFIT were scaled to the WMO standards using the calibration factors used by TCCON and as discussed in Wunch et al. (2015). The recent values of the correction factors (airmass dependent correction factor (ADCF) and airmass independent correction factor (AICF)) for the respective gases were taken from Table 4 in Wunch et al. (2015). The scaling factors for the Xgas values which were calculated using PROFFAST for the EM27/SUN are discussed in detail in Frey et al. (2015).

The optical fibre of the solar tracker of the Bruker IRcube was broken on 23 March 2017, which led to an interruption of measurements until it was replaced by the end of April. The blind intercomparison study revealed that the optical setting of the Bruker Vertex70 was not optimised. It was further improved on 06 July 2017 when a 20 mm aperture stop was introduced in the parallel light beam falling on the detector. This configuration was used until almost the end of the measurement period when the aperture stop was further reduced with an iris to 9 mm. However, we did not have any solar measurements with this setting due to unsuitable weather conditions. All three interventions on the respective instruments are marked in the time-series plots with vertical lines and the colours corresponding to the respective instrument. The dates are given in Table 3. In the following sections the intercomparison results will be shown, the long-term stability will be discussed and cases where clear deviations of the retrieval results from the participating instruments w.r.t. the reference data set are observed will be explained.





## 5.2 Methodology for data intercomparisons

The data acquisition of the level 2 products were different for each instrument (see Table 2 for details). In order to make the intercomparison, data from each instrument were sorted and all data within a time interval of a 5 min sequence were averaged and associated to the respective start time of the bin. The timestamp of the reference data set (e.g., TCCON) was matched with the same timestamp of the other instruments to find the coincident data pairs which were used for the difference and the correlation calculation. The TCCON as well as the low-resolution instruments showed a strong air-mass dependence for measurements with SZA>75°; these data were therefore not included in this study. Filtering these data removed only a very limited fraction of the data set (about 5% for EM27/SUN and LHR, about 10% for IRcube and about 13% for Vertex70). Statistical values were computed from the coincident data set to obtain the bias, scatter and seasonal variation of the individual instruments with respect to a reference data set from the Bruker IFS 125HR. A linear regression line was fitted to the correlation data set for each gas. The slope, the intercept, the correlation coefficient and the standard error are shown on the respective correlation plots.

## 5.3 Intercomparisons with reference TCCON data

The intercomparison results with the TCCON data as reference and data from other low resolution remote sensing instruments are discussed in this section species-by-species. All instruments performed the retrievals following their standard procedure and using TCCON a priori as the common prior. The statistical values for the intercomparison results (mean of the bias, the standard deviation of the difference and the Pearson correlation coefficient) are given in Table 4 and plotted in the overview summary plot of Fig. 16.

### 5.3.1 $XCO_2$ intercomparison results

The timeseries of the coincident $XCO_2$ values measured during the year 2017 by each test instrument and the reference TCCON instrument are shown in the top-panel of Fig. 1. The corresponding differences relative to the TCCON instrument are shown in the second row panel. The correlation plots between the test instruments and the TCCON instrument are shown in the individual panels of the third and the fourth rows of Fig. 1. The measured $XCO_2$ values are high during the early winter and low during the summer season which represents the annual seasonal cycle at the site. All instruments captured the annual summer drawdown.

Amongst the test FTS instruments, the EM27/SUN has the lowest mean bias of -0.18 ppm with a standard deviation of 0.45 ppm and a very high correlation coefficient of 0.995. The difference plot (second row panel of Fig. 1) as well as the correlation plot (bottom-right panel of Fig. 1) show a small seasonal dependency of the bias relative to the TCCON instrument.

The correlation plot of the bottom-left panel of Fig. 1 shows a step change of the $XCO_2$ values for the IRcube in March as a result of the replacement of the fibre-optic which caused a change of the ILS of the instrument. The IRcube data show high bias and have a small seasonal dependency. This may be because of the poorly defined ILS due to compact short focal length optics or detector non-linearity.





The Vertex70 also has shown a step change relative to the TCCON instrument since its modification in July 2017. The data set after the instrument modification shows a significant reduction in scatter and bias as compared to the earlier data from the campaign. As a result, data from the period between 06 July and 12 September 2017 are compared separately to characterise the behaviour of the Vertex70 relative to the TCCON instrument and the other test instruments. The statistics for the data for the selected period are shown in the lower part of Table 4 and are plotted in the summary plot of Fig. 16. The data from the Vertex70

show a significant reduction in bias from 1.93 ppm to 0.22 ppm, and the standard deviation from 1.72 ppm to 0.58 ppm, while the correlation coefficient remained still high. The Vertex70 and EM27/SUN measurements are comparable to each other for this period. The mean bias and the standard deviation of the other instruments are quite similar for the July-September period as compared to the full year. However due to the limited data set, the correlation coefficient is slightly poorer for the shorter period.

The LHR instrument is in its developmental phase and measured only $CO_2$ and $H_2O$. $XCO_2$ data were found to be affected by two clearly different noise processes: A high frequency random error, mostly determined by detector noise, was found ranging from 2 to 5 ppm (one sigma) depending on the instrument SNR. On top of this random error, large slowly varying diurnal biases were observed to be up to ~10 ppm. All bias included and averaged, the biases against the TCCON instrument for the full year were found to be $18.4 \pm 5.3$ ppm. These biases were found to be inherent to the re-engineered LHR instrument

in contrast to the better controlled laboratory one (Hoffmann et al., 2016). They are under studies; some instrumental ones have been identified to step from laser optical feedback and laser excess noise producing a variable offset in the heterodyne demodulated signal.

### 5.3.2    XCH$_4$ intercomparison results

The timeseries of the coincident $XCH_4$ values measured during the year 2017 by each test instrument and the reference

TCCON instrument (top panel plot), the corresponding differences relative to the TCCON instrument (middle row panel) and the correlation plots (bottom row panels) are shown in Fig. 2. $XCH_4$ values are high during the late winter followed by a dip during the spring and further rising during the summer period. The annual cycle can be seen for the TCCON, EM27/SUN and IRcube measurements. The Vertex70 data, after the instrument modification in July, are also representative of the TCCON measurements.

The EM27/SUN has the lowest mean bias of 0.003 ppm with a standard deviation of 0.005 ppm and a correlation coefficient of 0.962. The difference plot shown in the middle row panel of Fig. 2 shows that both the EM27/SUN and IRcube have a high bias of about 0.01 ppm with respect to the TCCON instrument in the period between early March and the end of May. Also the correlation plots relative to the TCCON data shown in the bottom-middle and the bottom-right panel of Fig. 2 for the IRcube and the EM27/SUN show the monthly deviation very clearly.

The Vertex70 data show a step change in bias of about 0.03 ppm and a significant reduction in the measurement standard deviation after the instrument modification. The statistical values for all instruments between 06 July and 12 September 2017 are shown in Table 4 and are plotted in Fig. 16. The Vertex70 data have the largest bias of 0.011 ppm followed by the IRcube with a bias of -0.008 ppm and the EM27/SUN with a zero bias relative to the TCCON instrument. The positive bias of the





Vertex70 still remains after the instrument modification and the annual cycle is also captured. The standard deviations of the

measurements from the three test instruments are comparable. The correlation coefficient for this shorter period is slightly

improved for the EM27/SUN relative to the full period. This points to the large difference in the $XCH_4$ values between March

and May as mentioned earlier. The reasons for this will be discussed in section 5.4.2.

### 5.3.3 XCO intercomparison results

Carbon monoxide is measured by TCCON, Vertex70 and EM27/SUN instruments. The timeseries of the coincident XCO

values measured during the year 2017 by these instruments (top panel), the corresponding differences (middle row panel) and

the correlation plots (bottom row panels) relative to the TCCON instrument are shown in Fig. 3. XCO values during the start

of the measurement period in late winter are high, followed by a dip during summer and rising values during the late summer

period. The annual cycle is seen by all instruments.

The EM27/SUN has the lowest mean bias of 4.54 ppb with a standard deviation of 1.37 ppb and a high correlation coefficient

of 0.993. The difference plot shows that the bias is seasonally dependent with high scatter during the summer period due to

measurements with large SZA variation performed on long summer days.

The Vertex70 data show a significant improvement in the scatter after the instrument modification. The statistics showing

the mean, standard deviation and correlation coefficient are given in the bottom part of Table 4. The Vertex70 result has a

high correlation coefficient of 0.991 for the period after the instrument modification. The scatter and outliers are reduced, the

comparison shows mean bias of 1.47 ppb and standard deviation of 1.04 ppb relative to the TCCON instrument.

### 5.3.4 Xair intercomparison results

Xair values were submitted by all FTIR instruments. The timeseries of the coincident Xair values for the year 2017 for the

instruments are shown in the top panel plot of Fig. 4 and the corresponding differences relative to the TCCON instrument are

shown in the middle panel plot. Ideally Xair being the scaled ratio of the surface pressure divided by the retrieved total column

of oxygen values should be 1. Any difference relative to the ideal case is an indicator for the instrument and retrieval code

performance.

The Xair values of the Vertex70 show two distinct groups due to the instrument modification in July. After the instrument

modification the scatter in the Xair values is significantly reduced. The EM27/SUN shows a slightly lower scatter compared

to the IRcube for the full year. However the scatter of the EM27/SUN, Vertex70 and IRcube are similar for the shorter time

period. There is a small offset relative to the TCCON instrument. However the small offset in bias is less important than a stable

Xair over the long timeseries. The correlation plots between the test instruments and the TCCON instrument are shown in the

bottom panel plots of Fig. 4. The bottom-left and bottom-middle panels show that the spread of the Xair values in the y-axis

(representing Vertex70 and IRcube, respectively) are higher than those in the x-axis (representing TCCON). The bottom-right

panel shows that the spread in the x-axis (representing TCCON) is slightly higher than the spread in the y-axis (representing

EM27/SUN) except for a few outliers. The EM27/SUN shows the smallest airmass dependence whereas the Vertex70 and the

IRcube show decreasing Xair values with an increasing SZA similar to the TCCON instrument. This may reflect the difference





between the GFIT and PROFFAST results.

The Xgas biases between the low resolution test instruments and the TCCON instrument as reference may be due to ef-
fects such as different responses to a priori profiles, interfering species in the retrieval windows or different averaging kernels.
Furthermore it is important to note that, TCCON uses a network-wide constant scaling factor to scale its Xgas values to the
WMO standards. The scaling factors specific to each gas for TCCON had been determined from several measurement cam-
paigns where vertically distributed measurements of the gases were performed from airborne platforms using WMO calibrated
instruments. The EM27/SUN uses species dependent scaling factors for $XCO_2$ and $XCH_4$ which had been calculated from
long-term intercomparison measurements performed at the KIT TCCON site. However, no such instrument specific calibration
factors were applied for the other instruments and also for the XCO results from the EM27/SUN measurements. This also
contributes to the residual bias which is observed in this intercomparison result. The biases which are purely due to resolution
differences are addressed by performing low resolution measurements with the same TCCON instrument. These data are then
used for an intercomparison relative to the standard TCCON as well as for the intercomparison with other low resolution test
instruments. Further details of the intercomparison results are given in sections 5.4 and 5.5, respectively.

### 5.4 Comparisons between TCCON and low resolution measurements performed with Bruker IFS 125HR

The Bruker IFS 125HR was configured to record regular low resolution measurements (spectral resolution of 0.5 cm$^{-1}$) to-
gether with the standard TCCON measurements (spectral resolution of 0.02 cm$^{-1}$). The low resolution measurements are
henceforth referred to as HR125LR. The HR125LR measurements, which are similar double-sided interferograms to the
EM27/SUN, were processed by the KIT-group using PROFFAST. The results were post-processed in the same way as the
results of the EM27/SUN. The comparison results with TCCON data as reference and HR125LR data recorded with the same
instrument are discussed in this section species-by-species.

The timeseries of the coincident $XCO_2$, $XCH_4$, XCO and Xair values measured during the year 2017 for the HR125LR and
the reference TCCON instrument are shown in the top panel, third row panel, fifth row panel and the seventh row panel plots
of Fig. 5. The corresponding difference for each species relative to the TCCON data are shown in the second row panel, fourth
row panel, sixth row panel and eight row panel plots, respectively. The mean bias, the standard deviation of the difference and
the correlation coefficient between HR125LR and TCCON data sets for the full year of measurements in 2017 and those for
the period between 6 July and 12 September 2017 (henceforth referred to as the shorter period) are given in Table 5 and are
plotted in the summary plot of Fig. 16. The shorter time period was chosen in order to compare the results with the improved
Vertex70 instrument.

### 5.4.1 $XCO_2$ comparison results

The seasonal cycle including the summer drawdown of the $CO_2$ was captured well by the HR125LR. The mean bias for the
full year and the shorter period of measurements are -0.69 ppm and -0.4 ppm, respectively. The relatively small difference in
the bias for the two time scales indicates that the bias is quite constant over the year. The high bias of -0.69 ppm may be due





to the choice of the constant scaling factor used for the calculation of the $XCO_2$ values. The same calibration factors as used by the EM27/SUN were used for the scaling of gases retrieved from the HR125LR measurements. However these calibration factors are specific to the EM27/SUN and therefore may not be accurate for the HR125LR. This is the reason for the high bias which is understood and not a problem as long as it remains constant and is not varying over the season. The difference plot (second row panel plot of Fig. 5) shows that there is a quite constant bias over the whole period with no seasonal dependencies.

The standard deviation of the difference (0.53 ppm) as well as the correlation coefficient (0.993) between the HR125LR and the TCCON data sets are very similar to those when comparing the EM27/SUN and the modified Vertex70 data sets relative to the TCCON data. This implies similar behaviour of the above mentioned low resolution instruments.

### 5.4.2  XCH$_4$ comparison results

The seasonal cycle of CH$_4$ is well captured by the HR125LR except for the March–May period. The mean bias for the full
year is -0.005 ppm with a standard deviation of -0.004 ppm and a correlation coefficient of 0.975. The difference plot of the $XCH_4$ values (see fourth row panel plot of Fig. 5) shows a relatively high bias during the March–May period. This feature is also seen in the intercomparison results between other low resolution instruments and the TCCON (see middle panel plot Fig. 2). The reason for this is the difference in resolution and the column averaging kernel (AK) between the TCCON and the low resolution instruments. During the March–May period the TCCON a priori profiles show large differences to the AirCore
profiles (see top three middle panels of Fig. 17) where the latter give a better representation of the true atmospheric state. The AK represents the sensitivity of the retrieved total column to the true partial column profile. An AK value of 1 for all altitudes is the ideal case, which implies perfect sensitivity for the whole atmosphere. In such case the retrieved total column represents the true atmospheric state. An AK value of <1 or >1 for a given altitude implies that the retrieval underestimates or overestimates the contribution from that particular layer in the total column calculation budget, respectively. The TCCON
AK for CH$_4$ is shown in the middle plot of the bottom-row of Fig. 17. The AKs for the high SZAs at the lower layers are underestimating and those above the troposphere are overestimating the contribution. Any deviation of the CH$_4$ a priori profile from the true atmospheric state will be affecting the retrieval results with the higher SZA (mostly during the winter seasons) more as compared to those with the lower SZA. The deviation of the TCCON a priori from the true profile in combination with the overestimation of the retrieval values due to AKs >1 for high SZAs during the spring season is the reason for the
high bias for the March–May period. This is further discussed in detail in section 5.9.1. The mean bias of the shorter time period is -0.007 ppm with a standard deviation of 0.002 ppm and a correlation coefficient of 0.97. The bias difference and the low standard deviation for the shorter period are due to the selected data set being outside the March–May period which does not cover the high values. The standard deviation of the difference between the HR125LR and TCCON data are very similar to those when comparing the EM27/SUN, IRcube and modified Vertex70 data sets relative to the TCCON data. This implies
again similar behaviour of the low resolution instruments.



### 5.4.3   XCO comparison results

The seasonal cycle of CO is captured well by the HR125LR. The mean bias for the full year is 0.03 ppb with a standard deviation of 1.02 ppb and a correlation coefficient of 0.996. The values for the shorter time period are similar to those of the full year. However a slight seasonal dependency is seen. The bias is due to the choice of the scaling factor used for the

calculation of the XCO values. The TCCON AK for CO is shown in the right panel plot of the bottom-row of Fig. 17. The AK for most of the atmospheric layers overestimates its contribution to the retrieval results. However the concentration of CO in the atmosphere decreases rapidly with increasing altitude implying that the contribution in the total column is low and not strongly dependent on the SZA of the measurements. The intercomparison results of the low resolution data set from all instruments relative to the TCCON data show that their performance was very similar in relation to the standard deviation of the difference

and the correlation coefficient relative to the TCCON data.

### 5.4.4   Xair comparison results

The Xair values for the HR125LR over the whole year are constant with a mean bias of 0.03 relative to the TCCON data and a standard deviation of 0.003. The mean bias for the shorter time period is the same as that for the full year. As the Xair values should be constant over the year, there is no correlation of Xair values expected between the two data sets. This is seen for the

shorter time period. The complete time period shows a slightly negative correlation between the two data sets. The constant Xair values over the longer time period show that the performance of the Bruker IFS 125HR operated in the low resolution mode was stable.

### 5.5   Intercomparisons with HR125LR data

In this section we discuss the intercomparison results between the HR125LR as reference in relation to other low resolu-

tion remote sensing instruments species-by-species. The timeseries of the coincident $XCO_2$, $XCH_4$ and XCO values for the HR125LR and the other test instruments are shown in the top panel, third row panel and fifth row panel plots of Fig. 6. The corresponding differences relative to the HR125LR are shown in the second row panel, fourth row panel and sixth row panel plots, respectively. The mean bias, standard deviation of the difference and the correlation coefficient between the individual instruments and the HR125LR for the full year of measurements in 2017 and that for the shorter time period are given in

Table 6 and are plotted in the summary plot of Fig. 16.

The mean biases of the target species for the test instruments (see Table 6) are close to the difference of the biases of the species in Table 4 minus the biases in Table 5 for the full year of 2017 and for the shorter time period. The Vertex70 shows a significant improvement of the bias, scatter and correlation coefficient for the intercomparison results during the shorter time period performed after the instrument modification.



### 5.5.1 $XCO_2$ intercomparison results

The EM27/SUN and the IRcube show slight improvement, while the modified Vertex70 shows a slight degradation of the standard deviation of the difference and the correlation coefficient for the intercomparison results relative to the HR125LR as compared to TCCON for the full year and the shorter time period (see Table 6 and Table 4). The scatter of the LHR instrument is very high and it is the dominating component of the intercomparison with other reference instruments.

### 5.5.2 $XCH_4$ intercomparison results

The high bias observed in the March–May period for the intercomparison of the test instruments with the TCCON instrument is not seen in the intercomparison results of the test instruments with the HR125LR. This indicates that the high resolution and the AK of TCCON (Fig. 17) is the cause of the large bias during this period where the TCCON a priori is further away from the true atmospheric state. The standard deviation of the difference and the correlation coefficient improved for the EM27/SUN comparison. The IRcube has no significant bias, also the scatter and the correlation coefficient improved for the HR125LR intercomparison in relation to the TCCON intercomparison. The modified Vertex70 has the same standard deviation of the difference and a similar correlation coefficient for the HR125LR as compared to the TCCON intercomparison results.

### 5.5.3 XCO intercomparison results

The EM27/SUN results for the full period and the Vertex70 results for the short period show similar values for the standard deviation of the difference and the correlation coefficient for the intercomparison results relative to the HR125LR as compared to the TCCON results.

The intercomparison results show the Xgas dependence on the resolution of the instrument, the averaging kernels and on the a priori. The low resolution measurements helped to identify that the high bias in the XCH4 during the March–May period was caused by the resolution difference and its sensitivity to the different averaging kernels.

### 5.6 SZA dependencies of bias

The TCCON Xgas values are known to be affected by the SZA during the measurements. In this section we check the SZA dependence of the low resolution instruments with respect to the TCCON and HR125LR data sets.

### 5.6.1 $XCO_2$ intercomparison results

The $XCO_2$ biases as a function of the measurement SZA for the low resolution test instruments relative to the TCCON and HR125LR for all measurements in 2017 are shown in Fig. 7 and Fig. 8, respectively.

The top-left panel plots in the two figures show no SZA dependence of the bias for the LHR. The scatter is very high and the dominant instrumental biases mask any possible SZA dependence.





The plots for the Vertex70 are shown in the top-right panels. The TCCON comparison results show no SZA dependence of the bias after the instrument modification which implies that the two instruments have the same SZA dependence. However the HR125LR comparison results show decreasing bias values for an increasing SZA of the measurements. This implies that the $XCO_2$ values retrieved from the HR125LR measurements have a smaller SZA dependency as compared to the Vertex70.

The plots for the IRcube and the EM27/SUN are shown in the bottom-left and the bottom-right panels. The IRcube bias shows an increase with SZA relative to the TCCON data set and is quite constant relative to the HR125LR data set. The EM27/SUN bias shows a slight increase with SZA relative to the TCCON and it shows a rather constant bias relative to the HR125LR. This implies that the $XCO_2$ values retrieved from the HR125LR measurements show a similar SZA dependence as compared to the retrieval results for the EM27/SUN and the IRcube.

The retrievals for the HR125LR and the EM27/SUN were performed by PROFFAST and the retrievals of other instruments were performed by GFIT. The data set of the HR125LR, the IRcube and the EM27/SUN were all measured at the same spectral resolution of $0.5$ cm$^{-1}$, whereas the data set for the Vertex70 were measured at a higher spectral resolution of $0.2$ cm$^{-1}$ and that of the TCCON were measured at $0.02$ cm$^{-1}$. From the plots it can be seen that the SZA dependency of the retrievals is related to the spectral resolution and the AK of the instruments. This explains the decrease of the standard deviation of the bias for the EM27/SUN from 0.45 ppm to 0.4 ppm while the correlation coefficient improved slightly where the first values are intercomparison results relative to the TCCON and the latter values are relative to the HR125LR. For the IRcube the standard deviation of the bias decreased from 1.06 ppm to 1.03 ppm while the correlation coefficient improved from 0.971 to 0.978. For the modified Vertex70 we see an increase in the standard deviation of the bias from 0.582 ppm to 0.657 ppm while the correlation coefficient decreased from 0.931 to 0.911.

### 5.6.2   XCH$_4$ intercomparison results

The XCH$_4$ biases as a function of the measurement SZA for the low resolution test instruments relative to the TCCON and the HR125LR for all measurements in 2017 are shown in the left panels and right panels of Fig. 9, respectively.

The Vertex70 XCH$_4$ biases are shown in the top panel plots. The TCCON comparison results show a slight decrease in the bias values for an increasing SZA of the measurements whereas the bias remains constant with an increasing SZA for the HR125LR comparison for the measurements performed after the instrument modification. The correlation coefficient and the scatter of the bias for the two comparison results are very similar.

The plots for the IRcube are shown in the middle panels. The IRcube bias shows a stronger dependence on SZA leading to a slightly poorer correlation coefficient of 0.924 for the TCCON comparison relative to a correlation coefficient of 0.949 for the HR125LR comparison. The standard deviation of the difference is slightly better for the HR125LR comparison.

The bottom-left panel plot shows a dependence of the bias as a function of the SZA for the EM27/SUN comparison relative to the TCCON. The bottom-right panel plot however does not show any dependence of the EM27/SUN comparison relative to the HR125LR. The correlation coefficient of the TCCON comparison was 0.962 and improved slightly to 0.978 for the HR125LR comparison. The scatter in the bias is slightly low leading to a significant improvement of the standard deviation of the difference from 0.005 ppm for the TCCON to 0.003 ppm for the HR125LR.





### 5.6.3 XCO intercomparison results

The XCO biases as a function of the measurement SZA for the Vertex70 and the EM27/SUN relative to the TCCON and the
HR125LR for all measurements in 2017 are shown in the left panels and right panels of Fig. 10, respectively.

The Vertex70 comparison results relative to the TCCON show a slight decrease in the bias with an increasing SZA of the
measurements performed after the instrument modification. However, the bias increases slightly with an increasing SZA of the
measurement for the HR125LR comparison. The standard deviation of the bias changed from 1.04 ppb to 1.17 ppb whereas
the correlation coefficient changed from 0.991 to 0.988 when comparing the Vertex70 results relative to TCCON and relative
to HR125LR. This shows that the Vertex70 comparison relative to the TCCON is slightly better than that of the HR125LR.

The EM27/SUN comparison results show a larger bias dependency on the SZA for the TCCON relative to the HR125LR
comparison. The standard deviation of the bias decreased from 1.37 ppb to 1.27 ppb and the correlation coefficient improved
from 0.993 to 0.995 when comparing the results relative to the HR125LR as compared to the TCCON.

This shows that the Vertex70 comparison with the TCCON shows better results whereas the EM27/SUN shows better results
in comparison with the HR125LR.

### 5.6.4 Xair intercomparison results

The Xair values as a function of the SZA of the measurements for the low resolution instruments and the TCCON for all measurements in 2017 are shown in Fig. 11. The plots show the SZA dependence of the retrieved oxygen column for the performed
measurements. The EM27/SUN and the HR125LR show no SZA dependence. However, the TCCON, the Vertex70 and the
IRcube show a decreasing Xair value for an increasing SZA of the measurements.

The EM27/SUN and the HR125LR do not show a SZA dependence for species where an airmass correction factor, which
was previously determined, was applied except for carbon monoxide where no correction was applied. The other instruments
show a SZA dependence to some degree. In order to minimise the effect of the SZA, measurements with an SZA<75° should
be used for the instruments.

### 5.7 Humidity dependencies of bias

The presence of water vapour lines in the retrieval windows can lead to errors in the determination of the Xgas values unless they are fitted well in the forward model. It is therefore necessary to check the influence of the water vapour lines for
retrievals performed with the low resolution instruments. In this section the bias dependence on the humidity present along
the measurement line-of-sight is discussed for the intercomparison of the low resolution test instruments with respect to the
HR125LR.





### 5.7.1 XCO$_2$ intercomparison results

The XCO$_2$ bias as a function of the humidity along the measurement line-of-sight for the low resolution test instruments relative to the HR125LR for all measurements in 2017 are shown in Fig. 12.

The top-right panel plot shows no bias dependency on the humidity for the Vertex70. The scatter in the bias shows a significant reduction after the instrument modification.

    The IRcube bias plot is shown in the bottom-left panel. The measurements in March, which were performed with the original optic fibre, show a high scatter in the values. The scatter is reduced after the change of the fibre and no humidity dependency of the bias is seen.

The bottom-right panel shows the bias plot for the EM27/SUN. The measurements in March show a higher scatter in the bias values compared to the rest of the year however no dependency on the humidity is seen.

    The top-left panel plot shows no bias dependency on the humidity for the LHR. However, owing to the instrumental biases the scatter is very high.

    This concludes that the XCO$_2$ retrieved from the low resolution instruments have no dependencies on the humidity in the

line-of-sight of the measurements.

### 5.7.2 XCH$_4$ intercomparison results

The XCH$_4$ bias as a function of the humidity along the measurement line-of-sight for the low resolution test instruments relative to the HR125LR for all measurements in 2017 are shown in Fig. 13.

    The top-left panel plot shows no bias dependency on the humidity for the Vertex70. Also here we see a significant reduction

of the scatter in the bias values after the instrument modification.

    The top-right panel plot shows no dependency of the bias on the humidity for the IRcube. Also here we see a high scatter before the change of the fibre.

    The middle row-left panel plot shows the bias for the EM27/SUN. The scatter in the bias values during the dry period is slightly higher as compared to the measurements from other months but there is no dependency on the humidity seen. The

high scatter of the bias during the dry period (March–May) may be due to the difference of the a prioris as compared to the true atmospheric state. This concludes that the XCH$_4$ retrieved from the low resolution instruments has no dependencies on the humidity in the line-of-sight of the measurements.

### 5.7.3 XCO intercomparison results

The XCO bias as a function of the humidity along the measurement line-of-sight for the low resolution test instruments relative

to the HR125LR for all measurements in 2017 are shown in Fig. 13.

    The middle row-right panel plot shows no bias dependency on the humidity for the Vertex70. A reduction in the scatter of the bias values after the instrument modification is also seen here.





The bottom panel shows the bias plot for the EM27/SUN. The scatter in the values is high, however no dependency on the humidity can be seen. This concludes that the XCO retrieved from the low resolution instruments has no dependencies on the humidity in the line-of-sight of the measurements.

### 5.7.4 Xair intercomparison results

The Xair values as a function of the humidity along the measurement line-of-sight for the low resolution test instruments, the HR125LR and the TCCON for all measurements in 2017 are shown in Fig. 14. The plot shows the humidity dependency of the retrieved oxygen column for the measurements performed. The Vertex70 and the IRcube show a significant reduction in the scatter of the Xair values after the instrument modification. The low resolution instruments show no dependencies on the humidity.

### 5.8 Detector non-linearity effects

The TCCON measurements performed during the campaign in 2017 are found to be affected by the non-linearity of the detector. The non-linearity was identified towards the very end of the campaign in 2017 while checking the interferogram signal measured by the TCCON and comparing it to the EM27/SUN. The detector non-linearity is dependent on the photon load incident on the detector and influences the Xgas values dependent on the signal strength of the measurements. The non-linearity being a signal dependent function, can be avoided by keeping the signal level within the linear domain of the detector. To test the non-linearity, a metal grid was placed in the parallel light beam at the entrance port to reduce the signal by about 20%. Figure 15 shows two spectra measured with the standard TCCON configuration with no grid (red) and with a grid (black) placed in the parallel light beam. These spectra cover the complete spectral regions measured by the detector and are zoomed in to highlight the signal of the out-of-band spectral regions. The non-linearity effect leads to out-of-band artifacts in the spectrum falsely indicating the presence of energy where the detector is insensitive. The signal between $0 \text{ cm}^{-1}$ and the lower cutoff of the detector at $4000 \text{ cm}^{-1}$ as well as the signal between the upper cutoff at about $12000 \text{ cm}^{-1}$ and the end of the detector bandpass at about $16000 \text{ cm}^{-1}$ show non-zero values for the no grid case indicating that the measurements performed were affected by the detector non-linearity. However the measurements performed with the reduced intensity by introducing the grid in the parallel beam do not show such high out-of-band intensities. The lower wavenumber out-of-band region shows only noise values and the higher wavenumber region close to the detector bandpass shows values which are higher than the noise but much lower than the signal of the standard measurements. These higher values could come from the spectral double passing of the signal within the interferometer. The high signal in the out-of-band spectral regions confirms that the TCCON measurements performed during 2017 are affected by the detector non-linearity. A correction method has been developed based on the method described in Hase (2000, chap. 5), it has been tested and applied to the TCCON data. The results of this are shown in the annex 1. In this section the intercomparison results of the low resolution test instruments are shown in comparison to the non-linearity corrected data measured by the TCCON, henceforth referred to as TCCONmod.

The non-linearity correction strongly affects the measured signal at both high and very low intensities. The mean bias, the standard deviation of the difference and the correlation coefficient between the individual low resolution data sets and the





TCCONmod for the full year of measurements in 2017 and for the period between 06 July and 12 September 2017 are shown in Table 7 and are plotted in the summary plot of Fig. 16. The comparison of the shorter time period has been made in order to check the statistics relative to the period where the Vertex70 was operated with improved settings.

### 5.8.1 XCO$_2$ intercomparison results

The mean bias changed by -0.5 ppm with a standard deviation change of 0.23 ppm for the intercomparison between the TCCON and the TCCONmod data sets for the full measurement period. The intercomparison of the low resolution measurements with TCCONmod and TCCON as reference also show similar changes in the mean bias values. The standard deviation of the difference and the correlation coefficient remained similar for the intercomparison with TCCONmod as compared to TCCON. As an example, the mean bias values for the EM27/SUN changed from -0.18 ppm to -0.73 ppm, the standard deviation of the

difference changed from 0.45 ppm to 0.47 ppm and the correlation coefficient changed from 0.995 to 0.996 for the comparison relative to TCCON where the latter values are for the comparison relative to TCCONmod.

### 5.8.2 XCH$_4$ intercomparison results

The mean bias changed by -0.003 ppm with a standard deviation change of 0.001 ppm for the intercomparison between the TCCON and the TCCONmod data sets for the full measurement period. The intercomparison of the low resolution measurements

with TCCONmod and TCCON as reference also show similar changes in the mean bias values. The standard deviation of the difference and the correlation coefficient remained similar or improved slightly for the intercomparison with TCCONmod as compared to TCCON. As an example, the mean bias values for the EM27/SUN changed from 0.003 ppm to 0.000 ppm, the standard deviation of the difference changed from 0.005 ppm to 0.004 ppm and the correlation coefficient changed from 0.962 to 0.973 for the comparison relative to TCCON where the latter values are for the comparison relative to TCCONmod.

### 5.8.3 XCO intercomparison results


The mean bias changed by -0.14 ppm with a standard deviation change of 0.08 ppm for the intercomparison between the TCCON and the TCCONmod data sets for the full measurement period. The intercomparison of the low resolution measurements with TCCONmod and TCCON as reference also show similar changes in the mean bias values. The standard deviation of the difference and the correlation coefficient remained similar for the intercomparison with TCCONmod as compared to TCCON.

As an example, the mean bias values for the EM27/SUN changed from 4.54 ppb to 4.38 ppb, the standard deviation of the difference changed from 1.37 ppb to 1.36 ppb and the correlation coefficient remained the same at 0.993 for the comparison relative to TCCON where the latter values are for the comparison relative to TCCONmod.

All of the results above for the individual species show that the non-linearity corrected TCCONmod data set has a different bias value which has to be taken into account. The standard deviation of the difference and the correlation coefficient remained

similar or improved slightly for the target species. The TCCONmod data set is a better representation of the true atmospheric signal. As TCCON is our primary data reference for the intercomparison study for this campaign, the non-linearity correction





has been applied to the TCCON data. Having this method implemented and tested for one year of data during this campaign, it will help in dealing with many years of historic TCCON data measured at the Sodankylä site. The HR125LR data were compromised by the non-linearity in a similar way to the high resolution TCCON spectra. This explains part of the residual

deviations of the low resolution test instruments compared to the HR125LR data.

## 5.9    Comparisons with AirCore data

In this section the comparison results between the low resolution test instruments and the TCCON are shown using the AirCore data. Table 8 gives a list of all AirCore measurements performed in 2017 at the Sodankylä site. The standard retrieval of each remote sensing instrument taking part in the intercomparison exercise were performed using the TCCON a priori. The daily

a priori files were automatically generated during the GFIT run. In addition, the tool to generate the daily TCCON a priori for any given location is available using a stand-alone program via a DOI link provided by Toon and Wunch (2017). The AirCore measurements are in-situ measurements of the targeted species calibrated to the WMO scale and serve as a better reference for the vertical profile of the measured species. However, the AirCore profiles are limited to a vertical sampling height of about 25–30 km depending on the ceiling height reached by the launching balloon. Given this height limitation, the AirCore profiles

cover only a part of the atmosphere relative to the TCCON a priori profile which covers a larger range starting from the site altitude up to 70 km. The lowermost layer of an AirCore profile is contaminated as the sampled air of the lowermost part of the atmosphere gets mixed with the reference push-gas. The push-gas is needed to let the sampled air pass through the analyser. The in-situ measurements performed at 2 m height above ground-level from a nearby ICOS (Integrated Carbon Observation System; https://www.icos-ri.eu/) ecosystem tower (https://en.ilmatieteenlaitos.fi/GHG-measurement-sites#Sodankyla) were used

to substitute the concentrations of the lowermost layer of the measured AirCore profile. The AirCore profile above the top most measured layer was further extended by a scaled TCCON a priori profile to cover the missing profile information up to 70 km of altitude. This is equivalent to a filling of < 5% of the total column above the top height of an AirCore measurement. The modified profile constructed using the ground-based in-situ measurement, AirCore measurement and the scaled TCCON a priori profile for each AirCore measurement day was further used for the intercomparison study.

The first three rows of Fig. 17 show the measured AirCore profiles (small blue dots), the a priori profiles from the GFIT run (small red dots), the tower mast measurements (green rectangle) and the extended AirCore profiles (large blue dots) for three sample days on 24 April, 15 May and 28 August 2017. The above mentioned three days were chosen to show the variability of the a priori profile during the different seasons at the Sodankylä site. The left-column represents the data plotted for $XCO_2$, the middle-column represents the plots for $XCH_4$ and the right-column represents the plots for XCO as a function of the altitude

for 24 April (top-row), 15 May (2nd-row) and 28 August 2017 (3rd-row), respectively. The column averaging kernels for $CO_2$, $CH_4$ and CO retrievals for TCCON plotted as a function of pressure and altitude are shown in the bottom-row of Fig. 17 from left to right. The different colours of the AK correspond to the varying SZAs of the measurements.



### 5.9.1 Intercomparison results using AirCore as a priori profile

The extended AirCore vertical profiles for the targeted gases derived from the AirCore flights have been fed as input a priori
profiles for the retrieval of the respective gases from the measurements performed with the remote sensing instruments on the
respective days. The retrieval results with the modified AirCore profiles have been given the suffix "AC" at the end of the
instrument name. As the remote sensing instruments covered a larger range of SZAs on 15 May and 28 August than on 24
April, those two days were selected for the intercomparison study.

### $XCO_2$ intercomparison results

The intercomparison results for $XCO_2$ retrieved using TCCON a priori and modified AirCore a priori for the measurements
performed with the TCCON and other low resolution FTIR spectrometers are shown in Fig. 18. The left and right columns
show the results for measurements performed on 15 May and on 28 August 2017, respectively. The difference between the
TCCON a priori and the modified AirCore a priori profiles is relatively small on 15 May as compared to the high difference of
the profiles on 28 August (see Fig. 17). This implies that the TCCON a priori is closer to the true atmospheric state on 15 May
than on 28 August. As a result the difference between the standard retrievals from each instrument using TCCON a priori and
the retrievals using the modified AirCore a priori is smaller on 15 May compared to the difference on 28 August. The retrieval
results for all instruments for the measurements on 28 August (see Fig. 18) show a bias between the TCCON a priori and the
modified AirCore a priori retrievals. The bias shows a strong dependency of the retrieval on the SZA of the measurements.
This is due to the TCCON $CO_2$ AK dependence on the SZA as seen in the left panel plot of the bottom-row of Fig. 17. With
these AK the a priori information is very relevant. The AirCore a priori is in principle the closest a priori to the truth. When
applying the AirCore a priori and doing the retrieval we see that the airmass dependence is much reduced.

For example the AK values for $CO_2$ for lower altitudes are >1 for higher SZA measurements which means that the retrieval
will overcompensate any over- or underestimation of the a priori: If the a priori is underestimating the lower partial column
values, then these will be overestimated by the retrieval in the total column amount; and vice versa if the a priori overestimates
the lower partial columns then the retrieval will underestimate their contribution in the total column amount. Similar reason-
ing is applicable to the case where the AK>1 for lower SZA measurements typically at local noon. Therefore we can see in
Fig. 17 that the TCCON a priori underestimates the lower partial columns during the summer months and that the TCCON
columns are too high compared to the TCCONAC. The plots also show the scatter of the retrieval results from the individual
instruments for two days. With these AK the a priori information is very relevant. The AirCore a priori is the closest a priori
to the truth. When applying the AirCore a priori and doing the retrieval we see that the airmass dependence is much reduced.
The EM27/SUN shows a lower scatter as compared to the TCCON. The Vertex70 measurements on 15 May were performed
before the instrument modifications. As a result a high bias relative to the TCCON was seen. This bias is not present for the
measurements performed after the instrument modification on 28 August. The scatter in the IRcube and Vertex70 is comparable
to the TCCON.



**XCH$_4$ intercomparison results**

The intercomparison results for XCH$_4$ retrieved using TCCON a priori and modified AirCore a priori for the measurements performed with TCCON and other low resolution FTIR spectrometers are shown in Fig. 19. The left and right columns show the results for measurements performed on 15 May and on 28 August 2017. The difference between the TCCON a priori and the modified AirCore a priori profiles of CH$_4$ is the highest for 24 April followed by 15 May and the smallest for 28 August (see Fig. 17). The vertical distribution of the CH$_4$ concentration during the winter and spring period is poorly modelled by the

TCCON a priori tool. The a priori during the summer is in better agreement with the AirCore measurements as seen for the 28 August profiles. As a result the difference between the standard retrievals from each instrument using TCCON a priori and the retrievals using the modified AirCore as a priori is smaller for 28 August than for 15 May.

The middle panel plot of the bottom-row of Fig. 17 shows the TCCON CH$_4$ AK dependence as a function of the SZA. The AK values are >1 for measurements at a lower SZA which means that the retrieval overestimates the contribution from all layers

above 10 km. However, the AK values are <1 for measurements with SZA >65° which means that the retrieval underestimates the contribution from all layers above 10 km. The TCCONAC results are higher than the TCCON results for the lower SZA values and vice versa. This effect is stronger for the retrieval results of 15 May compared to the results of 28 August where the TCCON a priori is closer to the AirCore a priori. The retrieval results for 15 May measurements for all instruments show a bias between the TCCON a priori and the modified AirCore a priori. The bias shows a strong dependency of the retrieval on the

SZA. The EM27/SUNAC results show a small bias compared to the EM27/SUN. The difference plot shows that the change of the retrieved XCH$_4$ values with the modified AirCore a priori has the same sign as compared to the TCCON. The same feature is also seen by the Vertex70 and IRcube results. The bias for the 28 August is largely reduced compared to that of the 15 May. The small remaining bias is due to the difference in the a priori and the AK of the instruments.

**XCO intercomparison results**

The intercomparison results for XCO retrieved using TCCON a priori and modified AirCore a priori for the measurements performed with the TCCON and the other low resolution FTIR spectrometers are shown in Fig. 20. The left and right columns show the results for measurements performed on 15 May and on 28 August 2017. The TCCON a priori and modified AirCore

a priori profiles of CO for three days in 2017 are shown in Fig. 17. The AirCore measured CO profiles are provided for altitudes up to 17 km and in some cases as high as 19 km. The AirCore profile measured on 28 August captured a large signal in the troposphere but it is not seen in the TCCON a priori. The difference in the profiles in the stratosphere is the largest for 24 April followed by 15 May and the difference is the smallest for 28 August. As a result the difference between the standard retrievals using TCCON a priori and the retrievals using the modified AirCore as a priori is slightly higher for 15 May than

for 28 August. The right panel plot of the bottom-row of Fig. 17 shows the TCCON CO AK dependence as a function of the SZA. The AK contribution to the retrieval results are underestimated (AK values <1) for layers below 5 km and overestimated

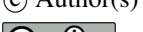



for layers above 5 km with AK values >1, even increasing up to above 2 for higher layers. The bias dependence on SZA is significant for measurements performed only at high SZA. The TCCONAC XCO retrievals show a constant bias relative to the TCCON XCO retrievals for most of the SZA and the deviation is seen only for measurements performed at the high SZAs.

The EM27/SUN and the Vertex70 results also show a slight dependency of the XCO retrieved using the TCCON a priori and the modified AirCore a priori on the measurements performed at a high SZA and a constant bias for measurements performed at a low SZA.

### 5.9.2 Direct intercomparison results of the Xgas calculated from AirCore relative to the TCCON and non-linearity corrected TCCON data sets

In this section we discuss the intercomparison results of the Xgas values calculated directly from the modified AirCore profiles by using the TCCON AKs in relation to the Xgas values retrieved from the standard TCCON and the non-linearity corrected TCCON data sets. Any difference in the intercomparison results is a direct reflection of the difference between the measured AirCore profile and the ground-based in-situ data relative to the TCCON a priori for the same altitude coverage. The time corresponding to 90% of the profile (starting at the top of the atmosphere) acquisition time is taken as the AirCore timestamp

for the intercomparison of the Xgas values. A 3h time window around the AirCore measurement time was used as the co-incidence limit. All Xgas values from TCCON data and TCCONmod data in this time window were averaged and taken as the coincident data sets for the intercomparison. The 3hr time window was selected for the remote sensing measurement as it is a good representation of the AirCore measurements. Reducing the time window resulted in the reduction of co-located measurement days and increasing the time window introduced the true variability of the atmospheric state in the remote sensing

data.

The mean bias, the standard deviation of the difference and the correlation coefficient of the Xgas values calculated from the AirCore relative to the TCCON and the TCCONmod are shown in Table 9. The $XCO_2$ mean bias between AirCore and TCCON is 0.47 ppm with a standard deviation of 0.66 ppm and a correlation coefficient of 0.994. The mean bias is reduced significantly to -0.03 ppm for the intercomparison between the AirCore and the TCCONmod. The standard deviation of the

difference is very similar, however the correlation coefficient improved slightly for the TCCONmod. This shows that the $XCO_2$ values from the TCCONmod data set are a better representation of the true atmospheric state.

The $XCH_4$ mean bias between the AirCore and the TCCONmod increases to -0.007 ppm as compared to the mean bias of -0.004 ppm between AirCore and TCCON. The scatter remains the same with an improvement in the correlation for the TCCONmod. The improvement in the correlation indicates that the TCCONmod data is a better representation of the true

atmospheric state. The increase in the mean bias is due to the difference in the TCCON a priori profiles used for the retrieval relative to the true atmospheric profiles. The top-panel of Fig. 21 shows the time series of a 30 min averaged TCCONmod $XCH_4$ data set and $XCH_4$ calculated from the AirCore measurements. The middle-panel plot shows the difference in the $XCH_4$ bias. The large difference between the two data sets in April is due to the difference between the a priori from the true atmospheric state. The bias reduces significantly for all later AirCore measurement days.





The XCO mean bias between AirCore and TCCONmod is slightly reduced to 6.25 ppb as compared to the mean bias of 6.4 ppb between AirCore and TCCON. The scatter is almost the same with very similar correlation coefficients. The CO retrieval from the AirCore has a large uncertainty. As a result the impact due to the change of the data set from the TCCON to the TCCONmod is within the uncertainty budget of the AirCore measurements.

The direct intercomparison results of the Xgas calculated from AirCore relative to the TCCON and non-linearity corrected
TCCON data sets indicate clearly that the non-linearity corrected data set gives Xgas amounts which are closer to the AirCore amounts and hence closer to our best estimate of the true atmospheric conditions.

# 6 Summary and outlook

The FRM4GHG campaign was successfully executed by comparing four portable remote sensing instruments against the reference TCCON instrument at the Sodankylä site during the year 2017. The EM27/SUN was set up every day at the ambient
temperature and pressure and was operated without configuration changes during the whole campaign. The other low resolution FTIR and the LHR were operated from inside a dedicated temperature controlled container. The instruments needed optimisation and behaved better with a low bias and a high correlation relative to the TCCON instrument afterwards.

The EM27/SUN Xgas biases relative to the TCCON data were low for the target species except for the high $XCH_4$ bias during the March–May period which is due to the difference in the sensitivity of the high and low resolution instruments
and the a prioris not matching well with the actual profile shape. The EM27/SUN measurements include a bias correction determined independently prior to this study.

The IRcube Xgas values show relatively high biases which are related to the possible dependence of the signal level on the extended InGaAs detector known to have non-linearity characteristics. The ILS of the IRcube is also less ideal compared to other larger instruments due to the compact short focal length optics. The impact of the ILS on the biases is being further
investigated. However the comparison shows low scatter and a good correlation relative to the TCCON data.

The Vertex70 was equipped with an extended InGaAs detector which led to identifiable non-linearity effects. The optical path was modified by introducing an aperture stop to avoid saturation and operate in the linear region of the detector to improve the ILS. The bias of the Xgas values, the standard deviation of the difference and the correlation of the modified Vertex70 instrument relative to the TCCON data were significantly lower after this instrument modification and comparable to
the EM27/SUN results relative to the TCCON data.

The LHR was a new instrument deployed under test during this campaign. It showed large scatter and large biases with a strong diurnal variation relative to the TCCON and other FTS instruments. The LHR data for the 2017 campaign are not yet able to provide meaningful geophysical information. However this comparison has proven to be invaluable to characterise and understand the instrumental biases and possibly the retrieval biases. Both aspects are currently under investigation and
improvements are being developed.

The intercomparison results of the low resolution measurements performed with the Bruker IFS 125HR relative to the standard TCCON and other low resolution instruments provided useful analysis of the resolution dependent effects on the





Xgas retrieved for the target gases. The low resolution measurements performed with the Bruker IFS 125HR also helped to identify that the high bias in the XCH$_4$ during the March–May period was caused by the resolution difference and the
corresponding different sensitivities to the vertical profile shape as seen in the averaging kernels.

The airmass dependence of the retrievals is an effect of the software. The EM27/SUN and the HR125LR results retrieved with PROFFAST do not show SZA dependence for species where an airmass correction factor was applied. All other results show SZA dependence to some degree. The correction for the SZA dependence is a long-standing and ongoing issue for TCCON and relevant to all instruments. In order to minimise the effect of the SZA, measurements with SZA <75° were used
for the intercomparison of the different data sets.

The bias dependence on the humidity along the measurement line-of-sight was investigated for each target species. However no dependence was found.

In the course of the campaign not only the Vertex70, IRcube and LHR instruments have been improved but also the TCCON instrument by detecting and correcting non-linearity of the detector response. Detecting this issue by comparison with the
EM27/SUN shows the potential of this instrument as a traveling standard for TCCON. The intercomparison results showed that the non-linearity corrected TCCON data gave a better match with the low resolution instruments. The standard deviation of the bias and the correlation coefficient was similar for the target species for the non-linearity corrected TCCON data relative to the standard TCCON data.

The intercomparison results using AirCore profiles as a priori showed interesting insights to the FTS retrievals, its sensitivity
to the resolution and the averaging kernels. The AirCore profiles also showed the differences relative to the TCCON a prioris and the resulting biases in the retrievals of the target species. The Xgas calculated from AirCore and compared to the TCCON and the non-linearity corrected TCCON data sets show that the latter data set is a better representation of the true atmospheric state.

The EM27/SUN, the IRcube, the modified Vertex70 and the HR125LR provided stable and precise measurements of the tar-
get gases during the campaign. The portable low resolution instruments can be used for campaigns or long-term measurements from any site, and complement the TCCON network. The Xgas measurements from these instruments will be of similar quality as the TCCON Xgas data.

## Appendix A:  Non-linearity of the Sodankylä TCCON InGaAs detector

The TCCON measurements performed during the campaign in 2017 are found to be affected by the non-linearity of the InGaAs
detector used for the measurements. A correction method has been developed based on the method described in Hase (2000, chap. 5) to correct the signal in the interferogram domain using the following equation:

$$I' = I + a \times I^2 + b \times I^3 + c \times I^4 \qquad\qquad\qquad (A1)$$

where I is the original interferogram and I$'$ is the non-linearity corrected interferogram. The coefficients a, b and c are determined for each measurement by minimising the cost function. The cost function is defined as the ratio of the signal sum in





the out-of-band regions covering 100–3600 cm$^{-1}$ and 14200–15750 cm$^{-1}$ to the signal sum for the in-band spectral region covering 4100–9700 cm$^{-1}$. The individual correction functions are plotted in Fig. A1 as red curves, their mean is plotted as blue open circles and the black line is the fourth order polynomial fit of the mean with the coefficient values of a = 0.00138, b = 0.00302 and c = 0.0166. The coefficients of the fitted curve representing the averaged correction function are used to correct the non-linearity of all TCCON measurements. The top-left and bottom-left panel plots of Fig. A2 show the original and the

non-linearity corrected spectra for the measurement day of 06 September 2017. The colours of the spectrum depend on the interferogram maximum signal at the centre burst. The highest values represented by the dark red colour are for measurements performed with the highest solar intensity during noon time and the measurements performed with the lowest solar intensity are represented by blue as the minimum. The top-right and bottom-right panel plots of Fig. A2 show the original and the non-linearity corrected spectra for the out-of-band region between 100 and 3600 cm$^{-1}$. The signal reduction in the out-of-band

spectral region clearly shows that the non-linearity correction worked for the spectra. In order to quantify the effect of the non-linearity correction on the Xgas values, explicit results are shown in detail for the 06 September 2017. The top panel plot of Fig. A3 shows the retrieved XCO$_2$ values from the original spectra (red) and the non-linearity corrected spectra (black). The middle and bottom row panels show the difference in ppm and the relative difference in percentage for the individual measurements. The non-linearity correction is dependent on the signal intensity of the recorded interferograms. The interferograms

with less signal intensity are affected less as compared to the signal with high intensity. The maximum of the correction is therefore applied for the measurements with the highest signal during the noon time and the minimum correction is needed for measurements with the lowest signal when the sun is near the horizon.

*Author contributions.* MKS, MDM and JN designed the study. MKS wrote the paper and produced the intercomparison analysis and results with input from all authors. All authors contributed in the generation of data used for this study. All authors read and provided comments on

the paper.

*Competing interests.* The authors declare that they have no conflict of interest.

*Acknowledgements.* The FRM4GHG project received financial support from the European Space Agency under the grant agreement number ESA-IPL-POE-LG-cl-LE-2015-1129. The authors wish to thank the instrument operators as well as the scientists doing the retrieval for the delivery of the data which was used for the intercomparison and discussed in this paper. We thank Tuomas Laurila and Juha Hatakka from

FMI for providing in-situ measurements from the 50 m tower at the site and for providing critical instrumentation needed for the AirCore analysis, including gas analyser, calibration gases and fill gases. We thank Nicholas Kumps for checking on the automatic operation and data recording of the Vertex70 instrument. KIT would like to thank the FMI staff and their student Elina Paulus for supporting the operation of the EM27/SUN spectrometer. The UK team also wishes to acknowledge support from the Centre of Earth Observation Instrumentation (CEOI) and the UK Space Agency (UKSA). We thank Paolo Castracane for his contribution during the project meetings and discussions.



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



**Figure 1.** Timeseries of $XCO_2$ retrievals for TCCON, LHR, Vertex70, IRcube and EM27/SUN using the standard procedure using TCCON a priori for measurements performed at Sodankylä in 2017 (top row panel), the difference of $XCO_2$ time series for each instrument relative to the reference TCCON results (second row panel). The correlation plots of $XCO_2$ from LHR, Vertex70, IRcube and EM27/SUN instruments vs TCCON for all measurements with SZA < 75° : Third row-left panel: LHR vs TCCON; third row-right panel: Vertex70 vs TCCON; bottom row-left panel: IRcube vs TCCON; bottom row-right panel: EM27/SUN vs TCCON measurements. The colours represent the measurements performed during the different months of the year.



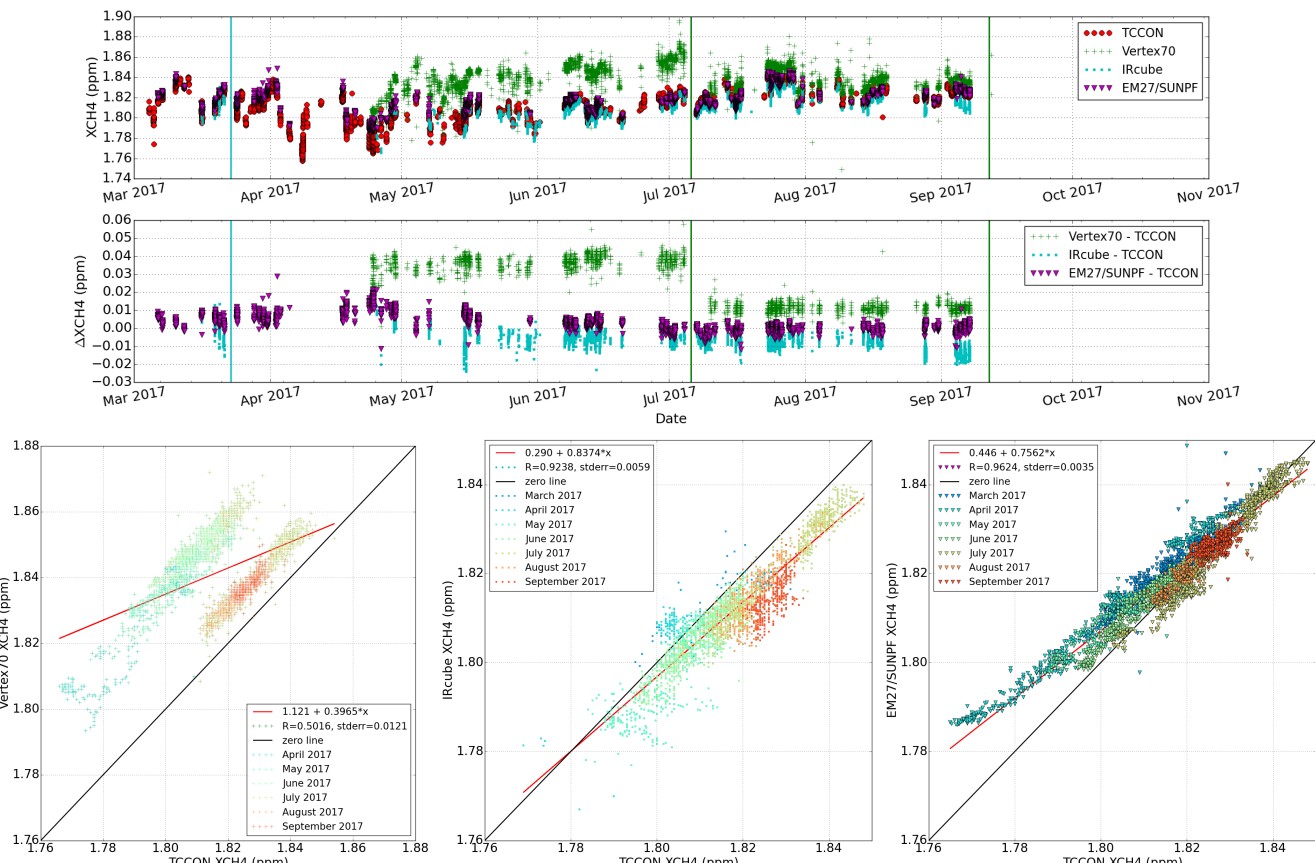

**Figure 2.** Timeseries of XCH$_4$ retrievals for TCCON, Vertex70, IRcube and EM27/SUN using the standard procedure using TCCON a priori for measurements performed at Sodankylä in 2017 (top row panel) and the difference of XCH$_4$ time series for each instrument relative to the reference TCCON results (middle row panel). The correlation plots of XCH$_4$ from Vertex70, IRcube and EM27/SUN instruments vs TCCON for all measurements with SZA < 75° : Bottom row-left panel: Vertex70 vs TCCON; bottom row-middle panel: IRcube vs TCCON; bottom row-right panel: EM27/SUN vs TCCON measurements. The colours represent the measurements performed during the different months of the year.



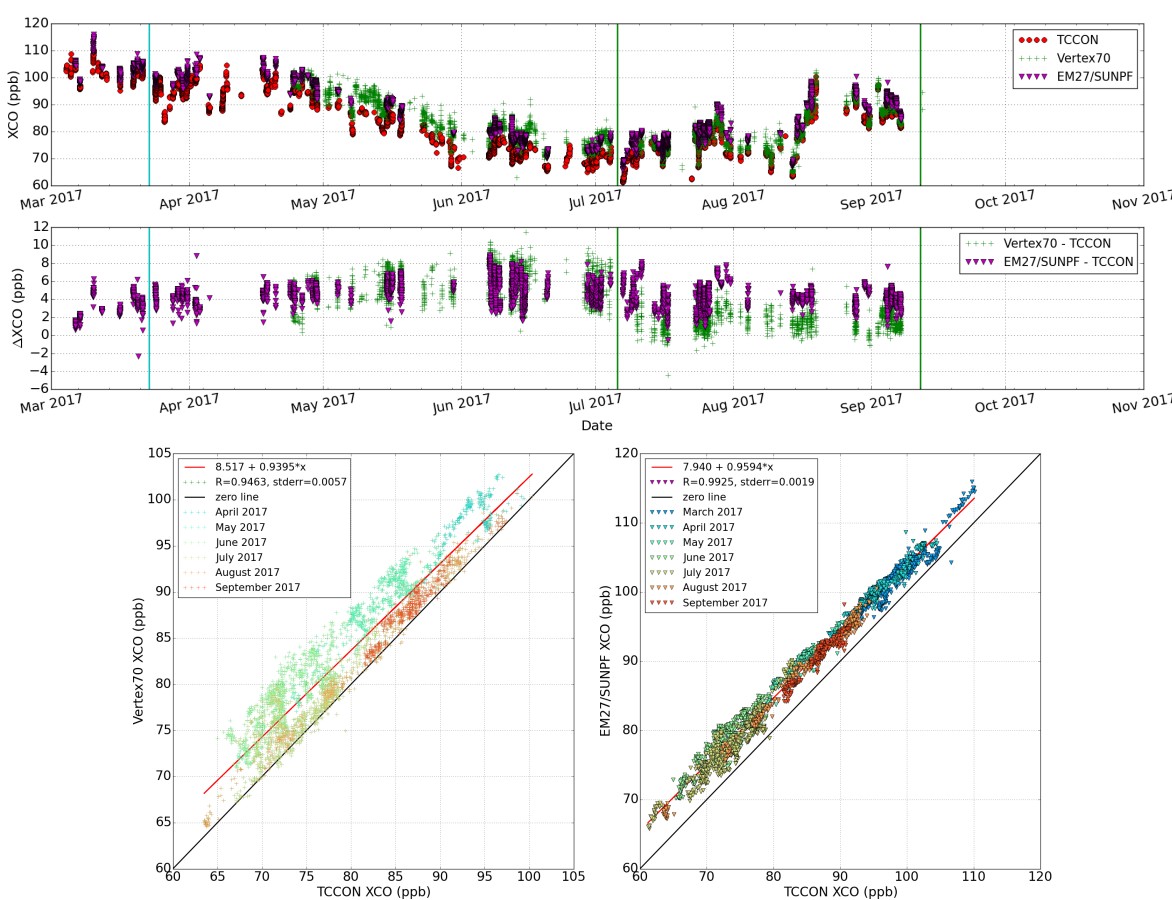

**Figure 3.** Timeseries of XCO retrievals for TCCON, Vertex70 and EM27/SUN using the standard procedure using TCCON a priori for measurements performed at Sodankylä in 2017 (top row panel) and the difference of XCO time series for each instrument relative to the reference TCCON results (middle row panel). The correlation plots of XCO from Vertex70 and EM27/SUN instruments vs TCCON for all measurements with SZA < 75° : Bottom row-left panel: Vertex70 vs TCCON; bottom row-right panel: EM27/SUN vs TCCON measurements. The colours represent the measurements performed during the different months of the year.



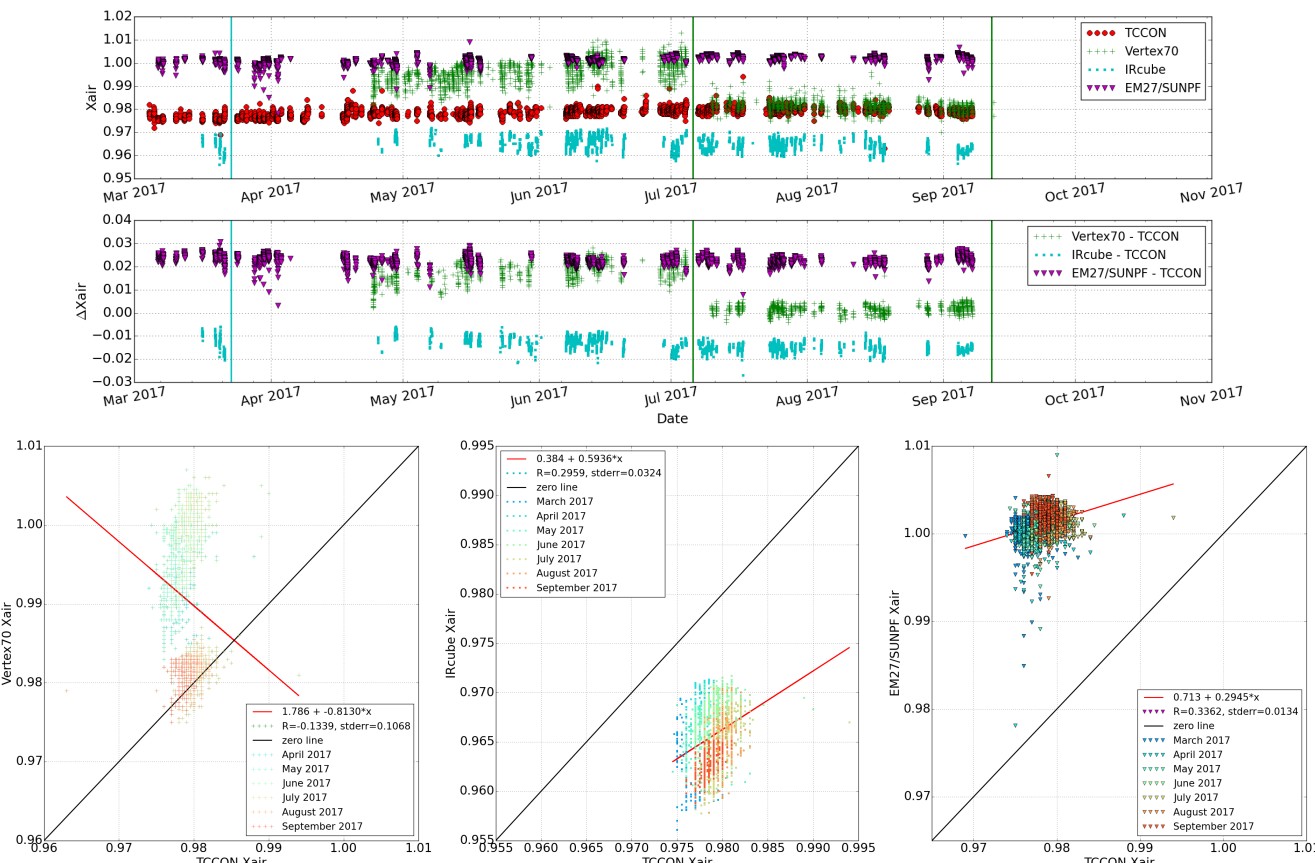

**Figure 4.** Timeseries of Xair for TCCON, Vertex70, IRcube and EM27/SUN using the standard procedure using TCCON a priori for measurements performed at Sodankylä in 2017 (top row panel) and the difference of Xair time series for each instrument relative to the reference TCCON results (middle row panel). The correlation plots of Xair from Vertex70, IRcube and EM27/SUN instruments vs TCCON for all measurements with SZA < 75° : Bottom row-left panel: Vertex70 vs TCCON; bottom row-middle panel: IRCUBE vs TCCON; bottom row-right panel: EM27/SUN vs TCCON measurements. The colours represent the measurements performed during the different months of the year.





**Figure 5.** Timeseries of $XCO_2$ (first row panel), $XCH_4$ (third row panel), XCO (fifth row panel) and Xair (seventh row panel) retrievals for TCCON and HR125LR using the standard procedure using TCCON a priori for measurements performed at Sodankylä in 2017. The difference of $XCO_2$ (second row panel), $XCH_4$ (fourth row panel), XCO (sixth row panel) and Xair (eighth row panel) time series for HR125LR relative to the reference TCCON results.



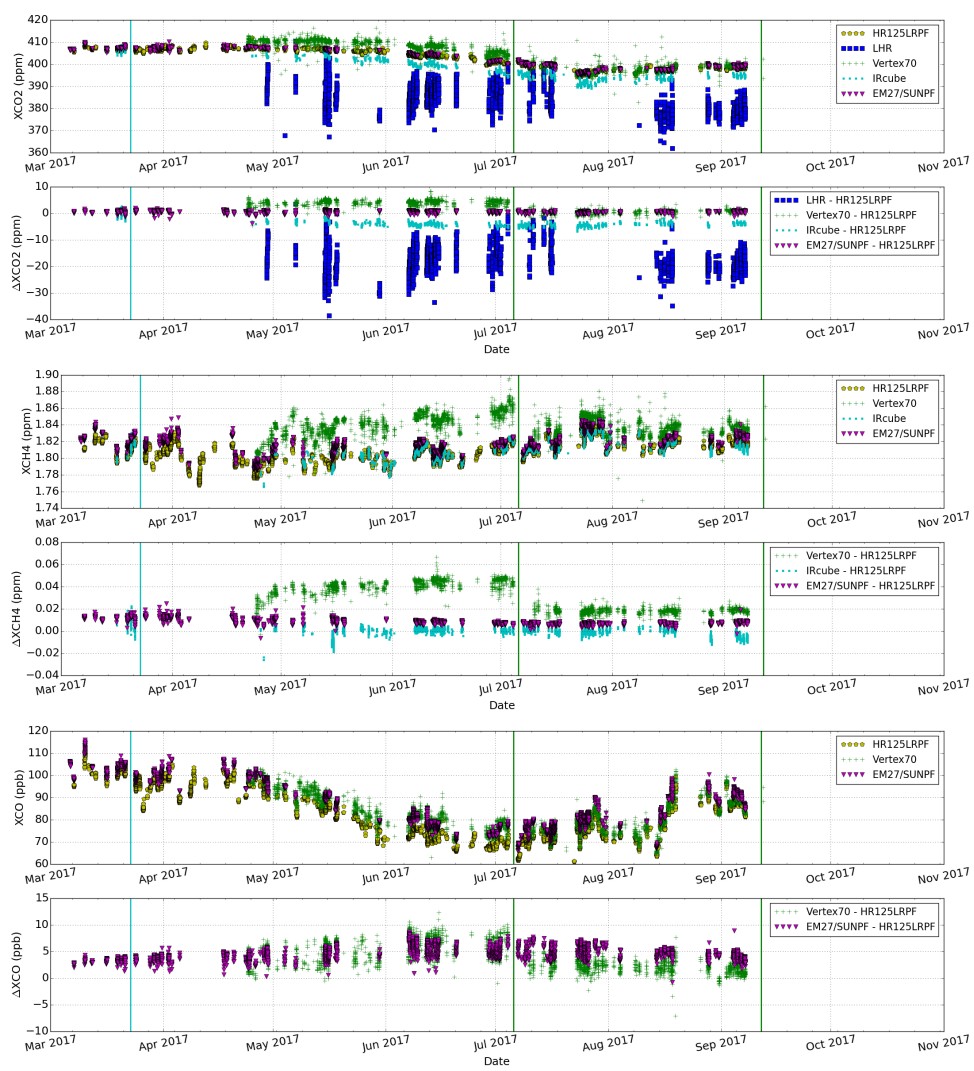

**Figure 6.** Timeseries of $XCO_2$ (first row panel), $XCH_4$ (third row panel) and XCO (fifth row panel) retrievals for HR125LR, LHR, Vertex70, IRcube and EM27/SUN using the standard procedure using TCCON a priori for measurements performed at Sodankylä in 2017. The difference of $XCO_2$ (second row panel), $XCH_4$ (fourth row panel) and XCO (sixth row panel) time series for the test instruments relative to the HR125LR results.



**Figure 7.** $XCO_2$ bias relative to TCCON for each instrument plotted w.r.t. the solar zenith angle: LHR (top left), Vertex70 (top right), IRcube (bottom left), EM27/SUN (bottom right). The colours represent the measurement performed during the different months of the year.

**Figure 8.** XCO$_2$ bias relative to HR125LR for each instrument plotted w.r.t. the solar zenith angle: LHR (top left), Vertex70 (top right), IRcube (bottom left), EM27/SUN (bottom right). The colours represent the measurement performed during the different months of the year.





**Figure 9.** XCH$_4$ bias relative to TCCON (left column) and relative to HR125LR (right column) for each instrument plotted w.r.t. the solar zenith angle: Vertex70 (top row), IRcube (middle row), EM27/SUN (bottom row). The colours represent the measurement performed during the different months of the year.



**Figure 10.** XCO bias relative to TCCON (left column) and relative to HR125LR (right column) for each instrument plotted w.r.t. the solar zenith angle: Vertex70 (top row) and EM27/SUN (bottom row). The colours represent the measurement performed during the different months of the year.





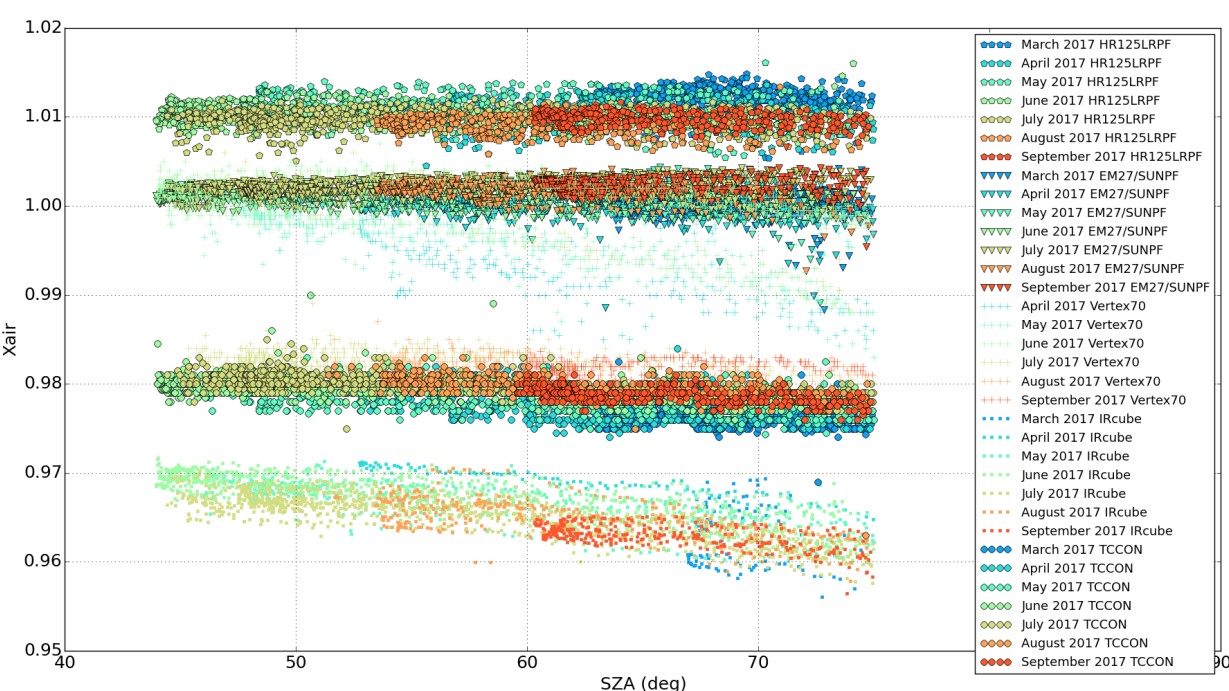

**Figure 11.** Xair plotted for HR125LR, EM27/SUN, Vertex70, IRcube and TCCON and w.r.t. the measurement solar zenith angle. The colours represent the measurement performed during the different months of the year.





**Figure 12.** $XCO_2$ bias relative to HR125LR for each instrument plotted w.r.t. the total column water vapour retrieved by the reference HR125LR measurements: LHR (top left), Vertex70 (top right), IRcube (bottom left), EM27/SUN (bottom right). The colours represent the measurement performed during the different months of the year.





**Figure 13.** XCH$_4$ bias relative to HR125LR for Vertex70 (top-left panel), for IRcube (top-right panel) and for EM27/SUN (middle-left panel) and XCO bias relative to HR125LR for Vertex70 (middle-right panel) and for EM27/SUN (bottom panel panel) plotted w.r.t. the total column water vapour retrieved by the reference HR125LR measurements. The colours represent the measurement performed during the different months of the year.

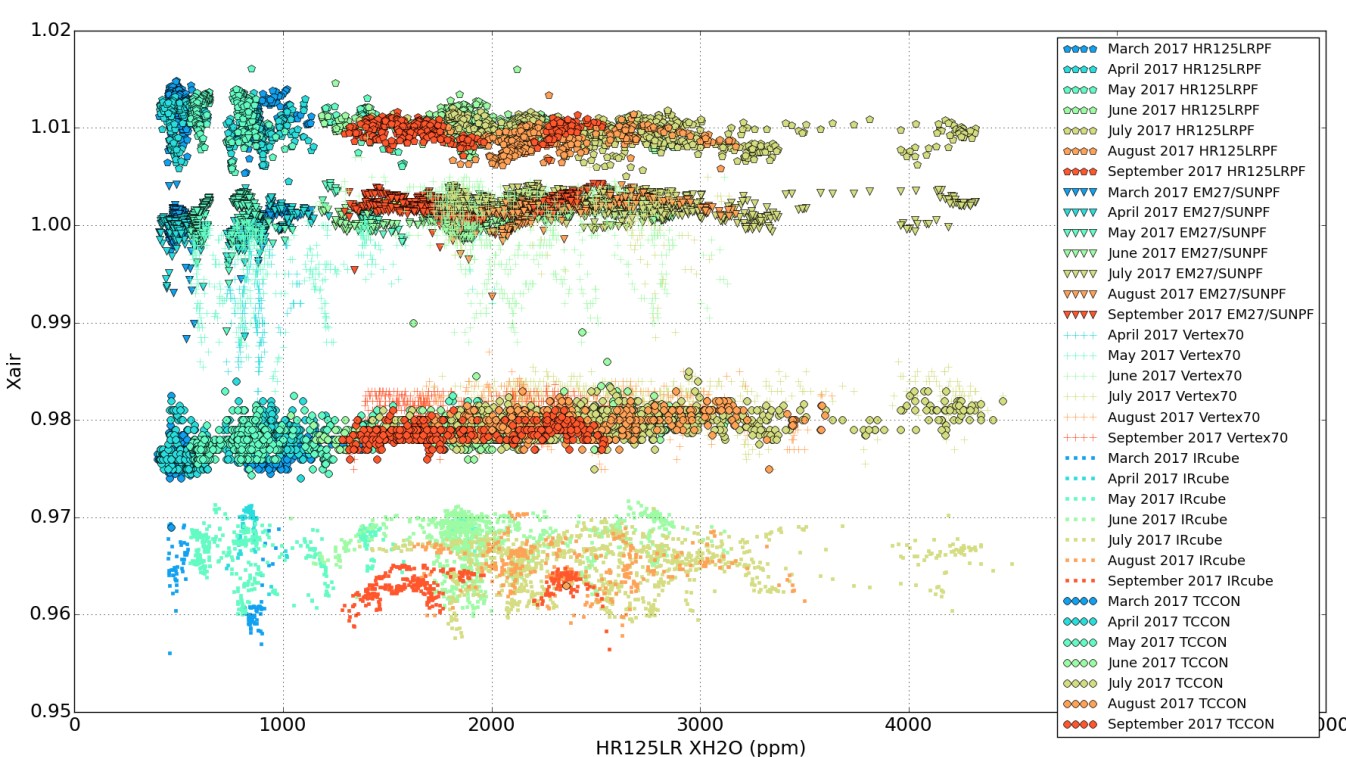

**Figure 14.** Xair plotted for HR125LR, EM27/SUN, Vertex70, IRcube and TCCON w.r.t. the total column water vapour retrieved by the reference HR125LR measurements. The colours represent the measurement performed during the different months of the year.





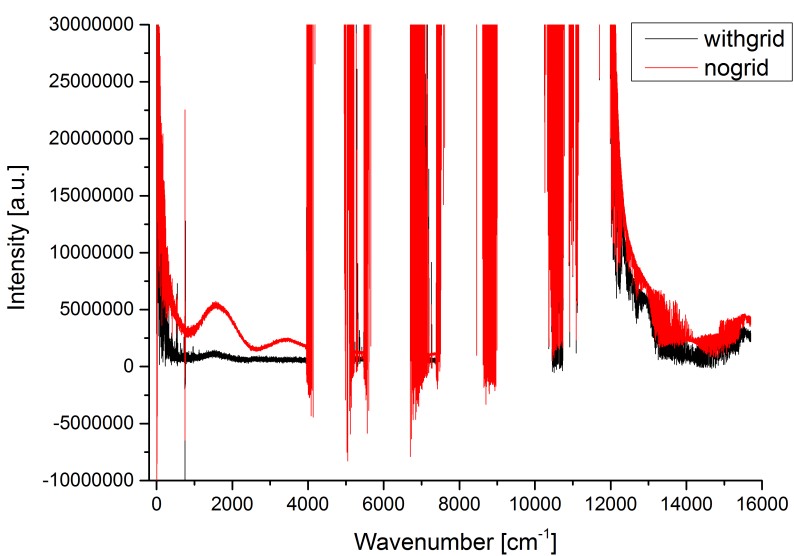

**Figure 15.** Standard spectrum (red) recorded with the Bruker IFS 125HR at Sodankylä TCCON facility. Spectrum (black) recorded with a grid placed in the parallel optical light path showing reduction of the non-linearity features in the out-of-band spectral regions. For comparison, the maximum intensity of both spectra (not visible in the plot) have been normalized to the same value.





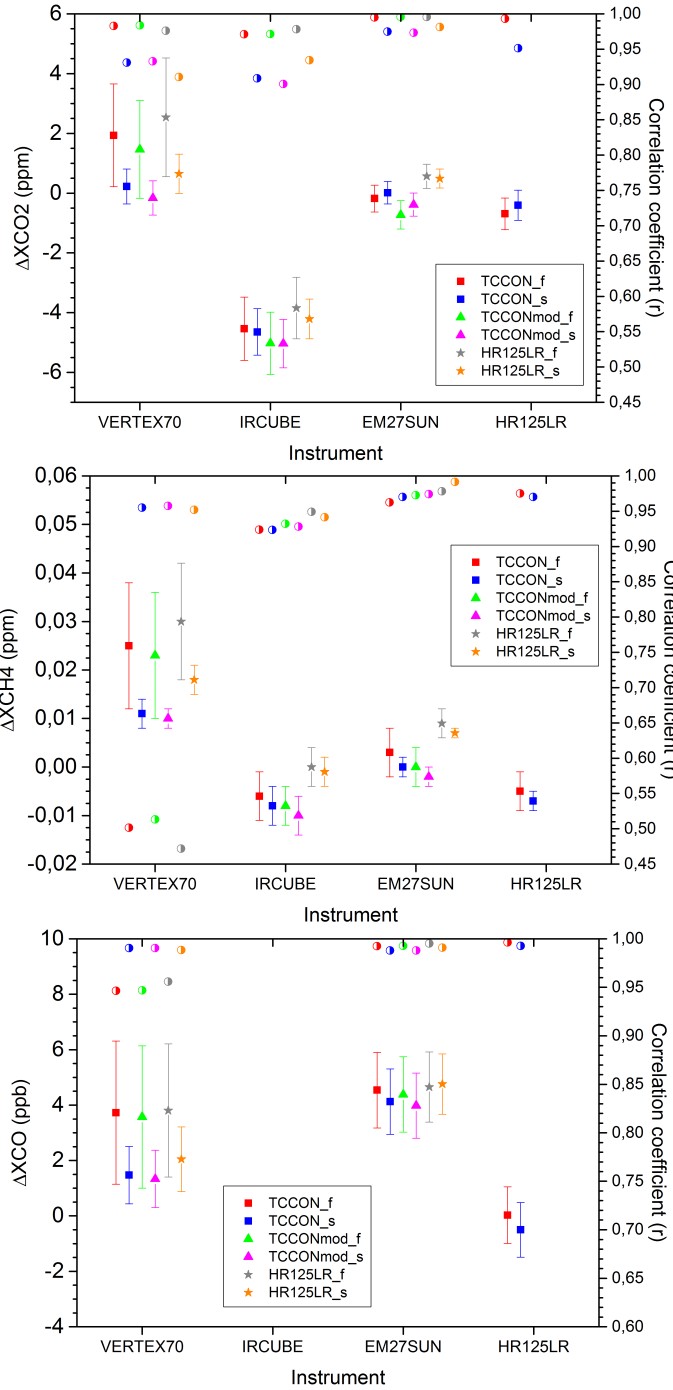

**Figure 16.** Top plot: $XCO_2$ bias plotted for each instrument relative to TCCON (full year–red box, short period–blue box), relative to non-linearity corrected TCCON (full year–green triangle, short period–magenta triangle) and relative to HR125LR (full year–grey star, short period–orange star). The correlation coefficient of the respective data set are plotted as half filled circles and correspond to the right side y-axis. The $XCH_4$ and XCO biases for each instrument are plotted in the middle and lower panel plots, respectively.





**Figure 17.** AirCore profile, GFIT map profile, AirCore extended profile and Tower mast measurements are plotted for $XCO_2$ (left-column), $XCH_4$ (middle-column) and XCO (right-column) as a function of altitude for 24 April 2017 with launch time at 15:13:39 and landing time at 16:13:10 UTC (top-row), 15 May 2017 with launch time at 09:33:22 UTC and landing time at 10:25:32 UTC (2nd-row) and 28 August 2017 with launch time at 09:13:15 UTC and landing time at 10:10:33 UTC (3rd-row), respectively. Bottom-row - left to right: Column averaging kernels of $CO_2$, $CH_4$ and CO retrievals for TCCON, plotted as a function of pressure and altitude. The different colours correspond to the varying solar zenith angle.

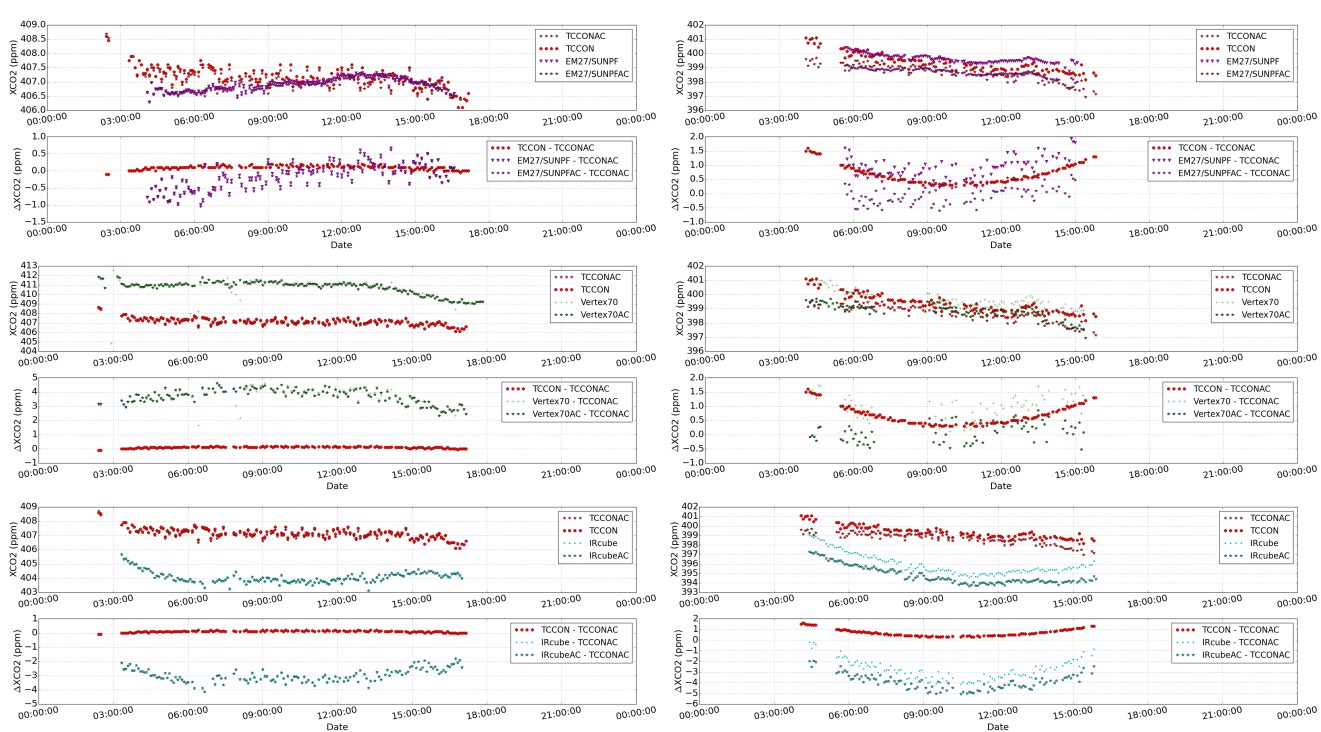

**Figure 18.** Top-left - upper panel: $XCO_2$ plotted for TCCON and EM27/SUN retrievals with the TCCON a priori and with a modified a priori (calculated using in-situ, AirCore and TCCON map files; labelled with AC in the end) for measurements performed on 15 May 2017 at Sodankylä. Top-left - lower panel: shows the difference between the two retrievals in absolute units. Middle-left and bottom-left figures show the same plots as mentioned above for Vertex70 and IRcube, respectively. Top-right, middle-right and bottom-right: show the above mentioned $XCO_2$ plots for measurements performed on 28 August 2017 at Sodankylä.



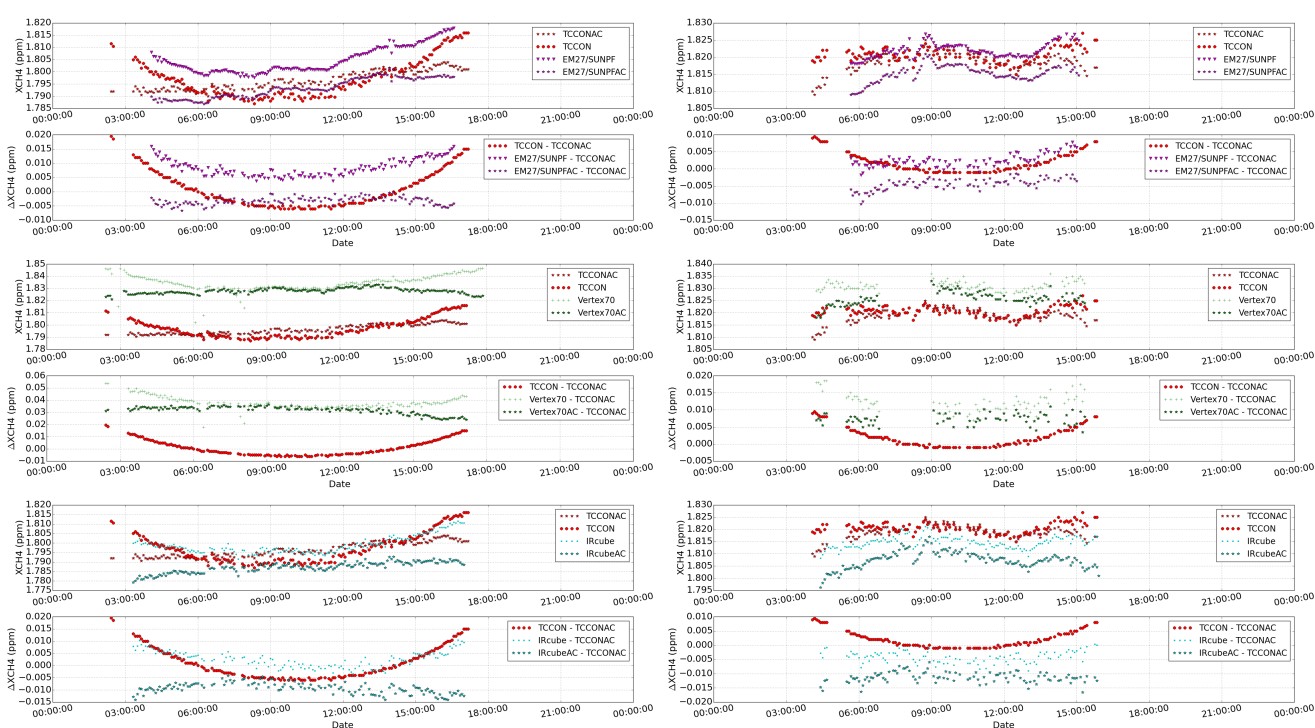

**Figure 19.** Top-left - upper panel: XCH$_4$ plotted for TCCON and EM27/SUN retrievals with the TCCON a priori and with a modified a priori (calculated using in-situ, AirCore and TCCON map files; labelled with AC in the end) for measurements performed on 15 May 2017 at Sodankylä. Top-left - lower panel: shows the difference between the two retrievals in absolute units. Middle-left and bottom-left figures show the same plots as mentioned above for Vertex70 and IRcube, respectively. Top-right, middle-right and bottom-right: show the above mentioned XCH$_4$ plots for measurements performed on 28 August 2017 at Sodankylä.




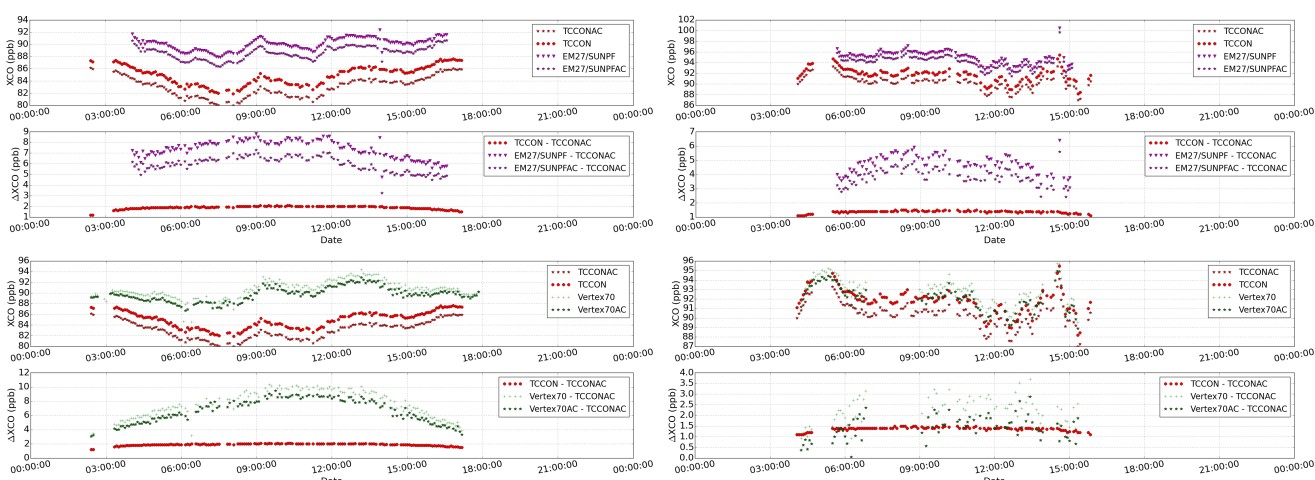

**Figure 20.** Top-left - upper panel: XCO plotted for TCCON and EM27/SUN retrievals with the TCCON a priori and with a modified a priori (calculated using in-situ, AirCore and TCCON map files; labelled with AC in the end) for measurements performed on 15 May 2017 at Sodankylä. Top-left - lower panel: show the difference between the two retrievals in absolute units. Bottom-left figures show the same plots as mentioned above for Vertex70. Top-right and bottom-right: show the above mentioned XCO plots for measurements performed on 28 August 2017 at Sodankylä.





**Figure 21.** Timeseries of XCH$_4$ retrievals using non-linearity corrected TCCON and AirCore measurements (top panel) and bias plot in absolute unit (middle panel) plotted for measurements performed in 2017 at SZA < 75°. The correlation of XCH$_4$ retrievals between non-linearity corrected TCCON and AirCore measurements are plotted in the lower panel. The colours represent the measurement performed during the different months of the year.





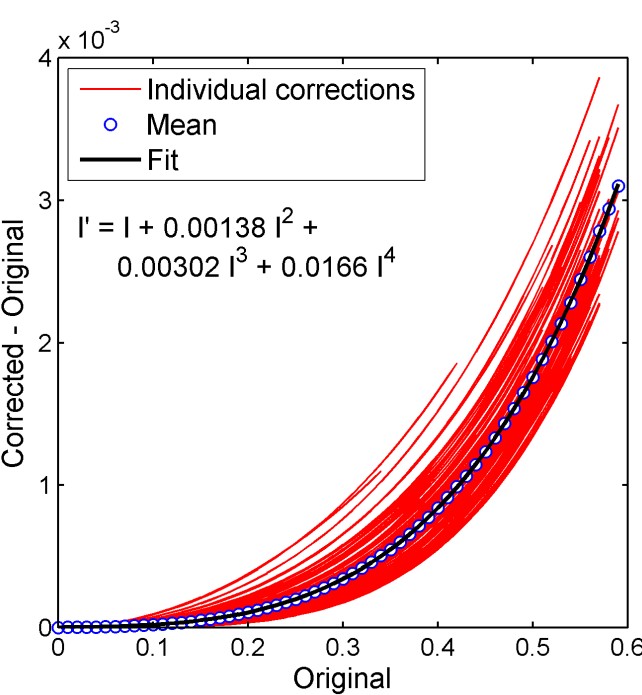

**Figure A1.** Plot showing the difference of the corrected interferograms - original interferograms vs the original interferograms. The individual corrections are plotted in red, the mean value is plotted as blue open circles and the black line is the fit.







**Figure A2.** Original (top left) and non-linearity corrected (bottom left) spectra; zoom of the out-of-band spectral region (100–3600 cm$^{-1}$) with the original spectra (top right) and non-linearity corrected (bottom right) spectra from the Bruker IFS 125HR at Sodankylä TCCON facility. The colour of the spectrum depends on the interferogram maximum signal at the center burst. The highest values corresponding to the dark red colour are recorded during the noon time when the signal is the highest.



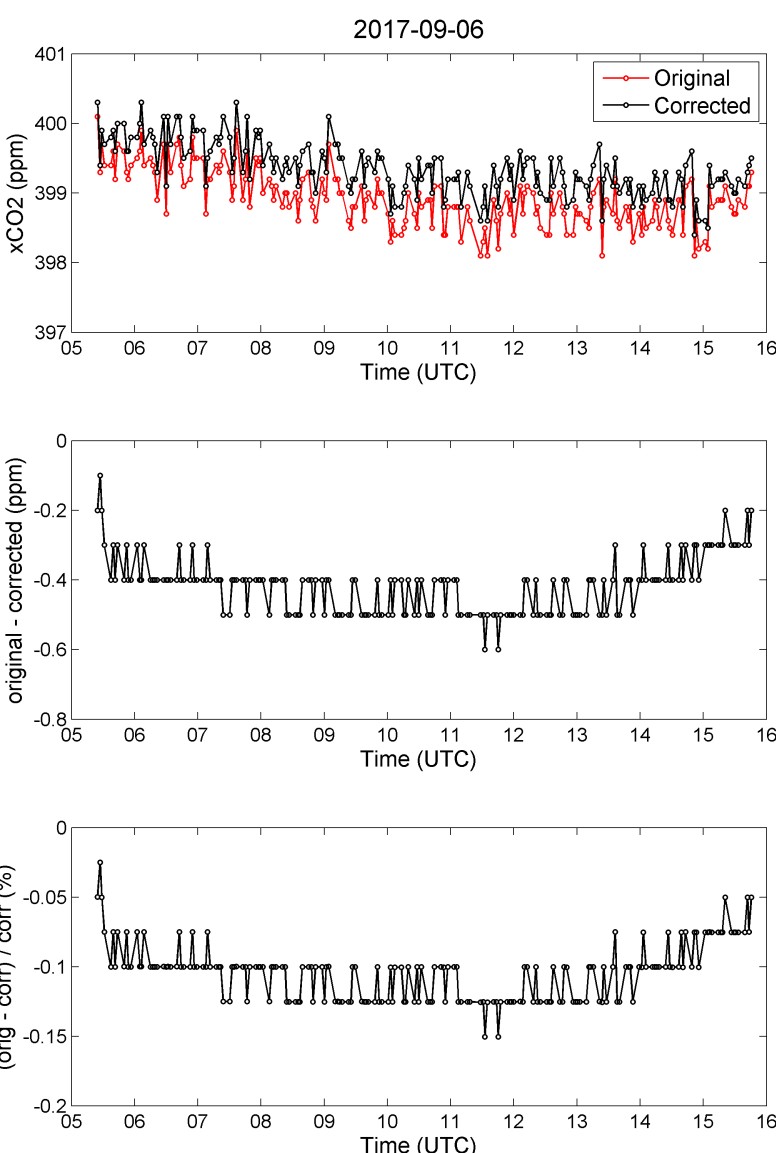

**Figure A3.** Top panel: plot showing the original (red) and non-linearity corrected (black) XCO2 values for one day of measurement performed on 06 September 2017 by the Bruker 125HR Sodankylä TCCON instrument. Middle-panel: shows the difference between the original and the corrected XCO2 values. Lower-panel: shows the relative difference (original - corrected)/corrected in percentage for the XCO2 values.





**Table 1.** List of instruments participating in the FRM4GHG campaign in 2017 and their properties.

| Item | Bruker IFS 125HR | Bruker Vertex70 | Bruker IRcube | Bruker EM27/SUN | LHR |
|---|---|---|---|---|---|
| Beamsplitter | CaF$_2$ | CaF$_2$ | Quartz | Quartz (single plate) | ZnSe |
| Entrance window | CaF$_2$ | CaF$_2$ | CaF$_2$ | Schott RG (IR transmitting filter glasses) | open |
| Aperture (mm) | 1 | 0.25 | 0.5 | 0.6 | 0.3 |
| Focal length (mm) | 418 | 100 | 69 | 127 | 75 |
| Scanner velocity (kHz) | 10 | 10 | 10 | 10 | N/A |
| Detectors | RT-Si Diode DC RT-InGaAs DC | RT-InGaAs DC LN2 cooled - InSb DC | RT-InGaAs DC | RT-InGaAs DC extended RT-InGaAs DC | Thermoelectrically cooled MCT |
| Acquisition mode | Single–sided forward–backward | Single–sided forward–backward | Single–sided forward–backward | Double–sided forward–backward | Sequential LO scanning |
| Dimension (cm$^3$) | | 80x50x30 | 29x31x23.5 | 35x40x27 | 40x40x20 |
| Weight (kg) | | 62 (without tracker) | 14 (without tracker) | 25 (with tracker) | ∼10 (without tracker) |
| Vacuum | yes | no | no | no | no |





**Table 2.** List of instruments, their measurement properties and retrieval strategy for the FRM4GHG campaign in 2017.

| Instrument | Institute | Spectral range (cm$^{-1}$) | Resolution (cm$^{-1}$) | Measurement time (approx., min) | Sample scans | Main species | Dataset | Retrieval code |
|---|---|---|---|---|---|---|---|---|
| Bruker IFS 125HR (TCCON) | FMI | 1800–15000 | 0.004 | 2.6 | 4 | XCO$_2$, XCH$_4$, XCO @ 0.02 cm$^{-1}$ | TCCON | GFIT 2014 |
| | | | | | | | TCCONNLC | Posterior non-linearity correction |
| Bruker Vertex70 | Uni Bremen & BIRA-IASB | 2500–15000 | 0.16 | 2.5 | 18 | XCO$_2$, XCH$_4$, XCO @ 0.2 cm$^{-1}$ | VERTEX70 | GFIT 2014 |
| Bruker IRcube | Uni Wollongong | 4500–15000 | 0.5 | 1.7 | 33 | XCO$_2$, XCH$_4$ | IRcube | GFIT 2014 |
| Bruker EM27/SUN (COCCON) | KIT | 4000–9000 | 0.5 | 1 | 10 | XCO$_2$, XCH$_4$, XCO | EM27/SUN | PROFFAST |
| Bruker IFS 125HR (HR125LR) | FMI & KIT | 1800–15000 | 0.004 | 1 | 10 | XCO$_2$, XCH$_4$, XCO @ 0.5 cm$^{-1}$ | HR125LR | PROFFAST |
| LHR | RAL | 952–955 | 0.002 and 0.02 | 0.5 | 1 | CO$_2$, H$_2$O @ 0.02 cm$^{-1}$ | LHR | own code, optimal estimation |
| AirCore | Uni Groningen & FMI | In-situ sampling | 13.4 mbar (Amb.P. > 232 mbar) 3.9 mbar (Amb.P. < 232 mbar) | | | CO$_2$, CH$_4$, CO vertical profiles calibrated to WMO standards | AirCore | |



**Table 3.** Instrumental line shape characteristics and modifications of the participating instruments.

| Instrument | max OPD (cm) | Modulation efficiency | Phase error (mrad) | Modification periods in 2017 | Modification comments |
|---|---|---|---|---|---|
| TCCON | 45 | <1.02 | ±2 | begin–end | no modifications |
| EM27/SUN | 1.8 | 1.02 | -3 to +1 | begin–end | no modifications |
| Vertex70 | | | | | |
| – before blind phase | 4.5 | -0.935 | -16 to -36 | begin–6 July | parallel beam diameter 40 mm |
| – after blind phase | | -0.973 | -13 | 6 July–12 September | reduced aperture with parallel beam diameter 20 mm |
| IRcube | 1.8 | -0.95 | -5 to +1.5 | begin–23 March | old fibre cable |
| | | | | 23 March–end | new fibre cable |





**Table 4.** Statistics of intercomparison results of LHR, Vertex70, IRcube and EM27/SUN vs TCCON for measurements performed in 2017 with SZA < 75°.

| Species | $XCO_2$ / ppm | $XCH_4$ / ppm | XCO / ppb | Xair |
|---|---|---|---|---|
| Bias (mean ± standard deviation) and correlation coefficient (r) | | | | |
| Measurement period: Full year in 2017 | | | | |
| LHR | -18.37±5.32 (0.502) | - | - | - |
| VERTEX70 | 1.93±1.72 (0.983) | 0.025±0.013 (0.502) | 3.73±2.58 (0.946) | 0.011±0.009 (-0.134) |
| IRCUBE | -4.54±1.06 (0.971) | -0.006±0.005 (0.924) | - | -0.013±0.003 (0.296) |
| EM27/SUN | -0.18±0.45 (0.995) | 0.003±0.005 (0.962) | 4.54±1.37 (0.993) | 0.023±0.002 (0.336) |
| Measurement period: 06 July–12 September 2017 | | | | |
| LHR | -18.62±4.43 (0.463) | - | - | - |
| VERTEX70 | 0.22±0.58 (0.931) | 0.011±0.003 (0.955) | 1.47±1.04 (0.991) | 0.002±0.002 (0.361) |
| IRCUBE | -4.65±0.78 (0.909) | -0.008±0.004 (0.924) | - | -0.015±0.002 (0.629) |
| EM27/SUN | 0.02±0.38 (0.975) | -0.000±0.002 (0.970) | 4.12±1.18 (0.988) | 0.022±0.002 (-0.158) |





**Table 5.** Statistics of the intercomparison results of HR125LR vs TCCON for measurements performed in 2017 with SZA < 75°.

| Species | $XCO_2$ / ppm | $XCH_4$ / ppm | XCO / ppb | Xair |
|---|---|---|---|---|
| Bias (mean ± standard deviation) and correlation coefficient (r) | | | | |
| Measurement period: Full year in 2017 | | | | |
| HR125LR | -0.69±0.53 (0.993) | -0.005±0.004 (0.975) | 0.03±1.02 (0.996) | 0.032±0.003 (-0.596) |
| Measurement period: 06 July–12 September 2017 | | | | |
| HR125LR | -0.40±0.50 (0.951) | -0.007±0.002 (0.970) | -0.50±0.99 (0.993) | 0.030±0.002 (-0.189) |

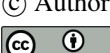



**Table 6.** Statistics of intercomparison results of LHR, Vertex70, IRcube and EM27/SUN vs HR125LR for measurements performed in 2017 with SZA < 75°.

| Species | $XCO_2$ / ppm | $XCH_4$ / ppm | XCO / ppb |
|---|---|---|---|
| Bias (mean ± standard deviation) and correlation coefficient (r) | | | |
| Measurement period: Full year in 2017 | | | |
| LHR | -17.52±5.27 (0.526) | - | - |
| VERTEX70 | 2.54±1.99 (0.976) | 0.030±0.012 (0.472) | 3.81±2.41 (0.956) |
| IRCUBE | -3.85±1.03 (0.978) | 0.000±0.004 (0.949) | - |
| EM27/SUN | 0.56±0.40 (0.996) | 0.009±0.003 (0.978) | 4.65±1.27 (0.995) |
| Measurement period: 06 July–12 September 2017 | | | |
| LHR | -18.00±4.61 (0.472) | - | - |
| VERTEX70 | 0.65±0.66 (0.911) | 0.018±0.003 (0.952) | 2.05±1.17 (0.988) |
| IRCUBE | -4.21±0.67 (0.934) | -0.001±0.003 (0.942) | - |
| EM27/SUN | 0.49±0.32 (0.981) | 0.007±0.001 (0.991) | 4.76±1.09 (0.991) |





**Table 7.** Statistics of intercomparison results of LHR, Vertex70, IRcube and EM27/SUN vs non-linearity corrected TCCON for measurements performed in 2017 with SZA < 75°.

| Species | $XCO_2$ / ppm | $XCH_4$ / ppm | XCO / ppb |
|---|---|---|---|
| Bias (mean ± standard deviation) and correlation coefficient (r) | | | |
| Measurement period: Full year in 2017 | | | |
| LHR | -18.89±5.34 (0.499) | - | - |
| VERTEX70 | 1.46±1.63 (0.984) | 0.023±0.013 (0.513) | 3.57±2.57 (0.947) |
| IRCUBE | -5.02±1.04 (0.971) | -0.008±0.004 (0.932) | - |
| EM27/SUN | -0.73±0.47 (0.996) | 0.000±0.004 (0.973) | 4.38±1.36 (0.993) |
| Measurement period: 06 July–12 September 2017 | | | |
| LHR | -19.02±4.44 (0.462) | - | - |
| VERTEX70 | -0.16±0.57 (0.933) | 0.010±0.002 (0.958) | 1.34±1.04 (0.991) |
| IRCUBE | -5.03±0.81 (0.901) | -0.010±0.004 (0.928) | - |
| EM27/SUN | -0.38±0.39 (0.973) | -0.002±0.002 (0.974) | 3.98±1.18 (0.988) |





Table 8. AirCore flight performed during the FRM4GHG campaign in 2017 at the Sodankylä TCCON site.

| Flights | Date | Start time of flight / UTC | End time of flight / UTC |
|---------|------|---------------------------|--------------------------|
| 1 | 21/04/2017 | 07:39:24 | 08:23:10 |
| 2 | 24/04/2017 | 15:13:39 | 16:13:10 |
| 3 | 26/04/2017 | 09:16:15 | 10:00:05 |
| 4 | 15/05/2017 | 09:33:22 | 10:25:32 |
| 5 | 28/08/2017 | 09:13:15 | 10:10:33 |
| 6 | 04/09/2017 | 09:15:58 | 10:04:15 |
| 7 | 05/09/2017 | 09:23:35 | 10:06:12 |
| 8 | 06/09/2017 | 09:10:20 | 09:49:10 |
| 9 | 07/09/2017 | 08:52:19 | 09:40:41 |
| 10 | 09/10/2017 | 09:49:48 | 10:50:14 |



**Table 9.** Statistics of intercomparison results of AirCore vs TCCON and non-linearity corrected TCCON data sets for measurements performed in 2017 with SZA < 75° for the TCCON measurements.

| Species | $XCO_2$ / ppm | $XCH_4$ / ppm | XCO / ppb |
|---|---|---|---|
| Bias (mean ± standard deviation) and correlation coefficient (r) | | | |
| TCCONvsAirCore | 0.47±0.66 (0.994) | -0.004±0.011 (0.959) | 6.40±1.88 (0.950) |
| TCCONmodvsAirCore | -0.03±0.71 (0.995) | -0.007±0.011 (0.969) | 6.25±1.88 (0.951) |