# Peer review of "Intercomparison of low and high resolution infrared spectrometers for ground-based solar remote sensing measurements of total column concentrations of CO2, CH4 and CO"

_Atmospheric Measurement Techniques, 2019_

## Referee Comment (RC1) · Ruediger Lang (Referee) · 16 Dec 2019

Accurate ground based observations of carbon dioxide and methane are becoming increasingly important in the context of continuous satellite remote sensing validation. While ground based in-situ and remote sensing networks measuring CO2 and CH4 concentrations in the atmosphere exist, and are instrumental in measuring the continuous increase in greenhouse gases over the past decades, it has been recognised that the existing network of stations is still lacking with respect to its expected role in future monitoring and verification system-of-systems (MVS) of greenhouse gas emissions.

Ground based remote sensing instruments like the Total Carbon Column Observing Network (TCONN) FTS instruments measuring total column dry air mole fractions of carbon dioxide and methane (XCO2 and XCH4) are key for the validation and calibration of future operational satellite based observing systems, targeting greenhouse gas emissions. However, the current distribution of stations and their representability for the validation of global satellite based measurements, as well as their maintenance and operability for providing a continuous flow of quality-monitored data are providing big challenges ahead for the network to become the much aimed for fiducial greenhouse-gas reference measurements within an operational MVS context. One of the key-challenges is achieving and maintaining the very high accuracies needed – significantly below 0.5 ppm for XCO2 – in order to become useful for the MVS and the Cal/Val of its satellite components.

The paper by Sha et al. is an important and significant contribution towards the latter aspect, by addressing the question on how to secure high accuracies across the network, e.g. through travelling instrument standards, while identifying, correcting and potentially even reducing instrument and measurement biases. The key finding of the paper is, from my point of view, the potential of at least one of the instruments taking part in the campaign exercise at Sodankyla, Finnland, functioning as a "travelling standard" in a potential future operational network of reference FTS instruments as currently operated under the TCCON umbrella. The paper also addresses important open questions with respect to remaining systematic biases and points at, maybe even more significant, remaining issues in measurement bias and precision. The paper is well written, although I think the paper could benefit from some restructuring of the results sections. I therefore recommend the paper for publishing in ACP, but would like to highlight in the following a couple of observations (apart from some additional minor comments), which the authors, from my point of view, should address.

1) Ari-core comparisons and non-linearity effects

The discussion of the non-linearity effect found in the measurements of the Bruker

IFS HR reference instruments (referred to as TCCON "reference", 125HR), as well as the comparison to the AirCore measurements, as taken during the campaign at Sodankyla - which are considered to represent the "true atmospheric" state - are both presented only at the end of the major results section 5. This is first of all confusing, since the section on non-linearity effects implies that the non-linearity correction "has been applied to TCCON" data as a whole. At this stage, the reader wonders if this therefore now applies to all previous results, but the answer is probably not, since the results and different labelling then implies that the main comparisons results are not (yet) non-linearity corrected.

Second, the performance of the "reference instruments" with respect to what is considered the "true state of the atmosphere" represented by the AirCore measurements is an important result against which also the results of the measurements from the other systems have to be evaluated and interpreted.

So both aspects have to be taken together. I would therefore recommend to present the results on the 125HR instrument non-linearity and the performance of the reference instruments against the "truth" (5.8 and 5.9) at the beginning of Section 5 (and ideally then present only the non-linearity corrected results for the TCONN reference – if this is considered a stable result – in the comparison against the other systems). This would help to interpret the results of the other systems when compared to the "reference" 125HR better with respect to the AirCore "truth".

In addition, establishing the TCCON 125HR as a "reference" and therefore then talking about a "bias" with respect to the reference for the other instruments (and not in terms of "differences"), requires, in my view, presenting the AirCore results first anyhow.

2) EM27 systematic biases with respect to the two references

The EM27 (COCCON) instruments show a convincing performance with respect to the "reference" TCCON (125HR), both with respect to the systematic biases and precision (for the latter see below). This is alas true especially with respect to the noncorrected TCCON results. It seems peculiar that this bias is even lower than -0.2 ppm for the non-corrected comparisons, but then gets worse for the TCCON non-linearity corrected ones (-0.73 ppm). At the same time, the low-resolution "reference" measurements (125LR) show a high bias to EM27 of roughly the same order of magnitude than the effect of the non-linearity correction. Considering that the low resolution 125HR are more comparable in terms of information content (and probably AKs, although those have unfortunately not been presented –see below), one wonders if the non-linearity correction should not also be applied to the 125LR measurements (or has this been done?), which then may lead to a consistency between EM27 and the 125LR. This would then also physically make a lot of sense considering the information content of both measurements. Also it would be important to rule out any link (e.g. in retrieval processing) between the standard 125HR measurements (not corrected) and the EM27 results, which potentially make them similar to the standard reference by default (and therefore more biased with respect to the corrected 125HR results and the AirCore "truth"). In this context, the processing algorithm ProFIT vs GFIT performances and their potentially relevant "peculiarities" are not much discussed at all. ProFIT is used both for the TCCON 125LR and the COCCON EM27s but not for the uncorrected TCCON "reference" it seems. Is the bias correction scheme used in ProFit maybe somehow related to TCCON measurements (or associated climatologies?).

I think it would be important to add a discussion of the (potential) relationship between the observed systematic biases between "reference" HR, LR, and non-linearity correction on one side, and the EM27 measurement results on the other side, which could shine some light on the underlying mechanisms.

3) Averaging kernels, low res versus high res

The differences in performance and measurement information content for the low-resolution measurements (125LR and EM27) are of high importance, especially with respect to the fact that some of the forthcoming remote sensing satellite based systems may be operated at lower spectral resolutions, i.e. being more similar to the LR

measurements in terms of spectroscopy. Also some of the differences observed between AirCore truth, high resolution and low resolution measurements, as discussed under 2), may be interpreted in this respect. The performance of these lower resolution systems are of importance also with respect to the knowledge of spectroscopy and potential future research needs there. But mainly the differences in performance between LR and HR (also with respect to AirCore "truth") may be better interpreted if the differences in their respective AK would be discussed and addressed - at least to some extend (and potentially also differences in GFIT and ProFIT a priori profiles if any – see also 2)). This could be done e.g., and ideally, in a corresponding figure to Fig. 17 (or adding such results to the latter).

4) Precision of TCCON versus COCCON

The striking difference in measurement noise (Fig. 18) between TCCON and COCCON EM27 is not much discussed it seems. Is this feature a question of retrieval algorithm performance (or/and applied constraints therein), spectral resolution (like the more stable LR with respect to HR measurements), or is it solely related to instrument noise?

Additional minor comments:

p. 2, l.36ff: "Carbon dioxide (CO2) and methane (CH4) are the two main components of the carbon cycle of the earth's atmosphere. They absorb and retain the heat in the atmosphere causing the greenhouse effect and global warming" – I would remove "greenhouse effect" here. Otherwise, water vapour would have to be mentioned in this context.

Section 5.3: I think it would be would be very helpful to introduce here the logic of the following sections (titles content) to guide the reader through what is coming, apart from potentially shifting the AirCore results and non-linearity sections first (see above).

Figure 1 to 4: It would be helpful to use the same axis limits in the scatter plots.

Figure 16: A horizontal dashed line or similar at zero would really help to interpret the results. Especially since the range of the y-axis has to be quite large.

Section 5.4.4. I think here it should be highlighted and maybe discussed in context of the discussion on biases between TCCON, TCCONmod, LR and EM27, why the EM27 is so much closer to one than TCCON (see also point 2) above).

――――――――――――――――――――

---

## Referee Comment (RC2) · Anonymous Referee #2 · 24 Dec 2019

Review of Sha et al.: Intercomparison of low and high resolution infrared spectrometers for ground-based solar remote sensing measurements of total column concentrations of CO2, CH4 and CO

This paper details the results from an intensive campaign wherein 4 portable spectrometers (Bruker EM27/SUN, Bruker IRcube, Bruker Vertex40, RAL LHR) were hosted near the Sodankylä TCCON station from March-September 2017. AirCore profiles were also flown throughout the campaign. The campaign resulted in some interesting data collected that helped reduce uncertainties in the Vertex40, pointed out where the

LHR needs improvement, and identified a problem unique to the Sodankylä TCCON station that was resolved.

This paper may be suitable for publication after the following concerns are satisfactorily addressed.

Major Comments

1. The nonlinearity problem at Sodankylä is unique to Sodanklyä, and this is not made clear in the paper. In general, TCCON stations have limited the light incident on the detectors by a combination of reducing the input field stop (on the aperture wheel) and placing an aperture stop after the input CaF2 window. From what I can glean from Section 5.8 (specifically, "The TCCONmod data set is a better representation of the true atmospheric signal. As TCCON is our primary data reference for the intercomparison study for this campaign, the non-linearity correction has been applied to the TCCON data."), this nonlinearity correction has been applied to all comparisons/figures/tables throughout the paper (except where the nonlinearity is discussed directly). If I have interpreted this correctly, I believe it is the correct approach, since it is more representative of the data a TCCON station would produce. That said, it should be stated up front (i.e., in section 2.2.1) that this correction has been applied, with a clear statement that this does not generally affect all TCCON data; just Sodankylä's. The details included in section 5.8 (and 5.9) ought to be relegated to Appendix A.

2. Your comparisons of Xair between the various instruments is misleading because it is not an apples-to-apples comparison. It is not only instrument effects that impact Xair: it is also spectroscopy. PROFFAST and GGG2014 do not include the same spectroscopy, line shapes, etc., and thus would be expected that they have different Xair values. It would be helpful to have a short (1-paragraph) discussion about the differences between the two retrieval software algorithms, as relevant to this issue. It is unclear to me why you would choose to use PROFFAST to analyse the EM27/SUN data and not also use the EGI wrapper for GGG2014 to maintain consistency with

the other retrievals. I suggest you add the EGI retrievals of the EM27/SUNs to your analysis. (https://bitbucket.org/em27gi/egi/wiki/Home)

3. It is unclear to me why you include the time period before the hardware upgrades of the Vertex and IRcube in your subsequent analysis. It makes more sense to me to show the significant improvement in their data after fixing the hardware in the time series plot, and then do not show the pre-improvement data in subsequent analyses, focusing only on the "good" data.

Minor comments

1. I found the authors' motivation for the need for the COCCON (or another complement to TCCON) misleading. I believe it is true that the TCCON could be usefully supplemented by LR portable spectrometers, but the atmospheric and surface conditions you list are generally already covered by TCCON stations. For example:

a. "A denser distribution of ground-based solar absorption measurements is needed to cover various atmospheric conditions (humid, dry, polluted, presence of aerosol), various surface conditions (high and low albedo) and a larger latitudinal distribution."

i. Humid –> Darwin

ii. Dry –> Armstrong, Eureka, Sodankylä, Pasadena

iii. Polluted –> Pasadena, Tsukuba, Wollongong (sometimes)

iv. Aerosol –> Pasadena, Tsukuba, sometimes Ascension

v. High albedo –> Armstrong

vi. Low albedo –> Park Falls, etc.

b. Where I do agree, is that if we want to cover geographic gaps in the locations of these stations, we need more stations, and the low-resolution instruments may be well suited to that.

2. It is unclear to me whether you use identical surface pressure values in your retrievals for each instrument. If they are different, this would also cause biases in Xair. For a fair comparison, they should be identical and calibrated to a meteorological standard. Please discuss and resolve if necessary.

3. It is unclear to me why the HR125LR has its own section. Why not include it in the comparisons with the other LR instruments wrt TCCON?

4. The water vapour dependence section is not conclusive, since Sodankylä is generally dry (XH2O <4500 ppm). This is therefore not an exhaustive test of XH2O dependence. I suggest moving this section to an appendix, noting in the main text that little XH2O dependence was found over the (relatively dry) conditions at Sodankylä.

Technical comments

1. P1L1: . . . the baseline *ground-based* network of instruments. . .

2. P1L5: Northern America –> North America and again in L72.

3. P2L25-26: This seems to imply that the nonlinearity is a problem throughout TCCON, which it is not. Please revise.

4. P2L41: increasing in ** recent years (no "the")

5. P2L43: You may want to mention the (important) role of VOCs in the production of CO.

6. P2L48: positive radiative forcing*, therefore it is* considered as an indirect. . .

7. P3 first paragraph: Generally unnecessary paragraph. It's unclear what you mean by "To ensure equal dependency on the input spectral data, . . ."

8. P3L74: Again, there are TCCON stations that span all those conditions. Rephrase.

9. P3L80. "However, there has been little characterization, intercomparison and harmonization of these new instruments in comparison to the standard instrument used in

TCCON." There is some literature on just this topic. Please cite:

a. Hedelius, J. K., C. Viatte, D. Wunch, C. M. Roehl, G. C. Toon, J. Chen, T. Jones, S. C. Wofsy, J. E. Franklin, H. Parker, M. K. Dubey, and P. O. Wennberg (2016), Assessment of errors and biases in retrievals of XCO2, XCH4, XCO, and XN2O from a 0.5 cm-1 resolution solar-viewing spectrometer, Atmos. Meas. Tech., 9(8), 3527–3546, doi:10.5194/amt-9-3527-2016.

b. Hedelius, J. K., H. Parker, D. Wunch, C. M. Roehl, C. Viatte, S. Newman, G. C. Toon, J. R. Podolske, P. W. Hillyard, L. T. Iraci, M. K. Dubey, and P. O. Wennberg (2017), Intercomparability of XCO2 and XCH4 from the United States TCCON sites, Atmos. Meas. Tech., 10(4), 1481–1493, doi:10.5194/amt-10-1481-2017.

10. While I understand that Sodankylä was chosen for (important) practical considerations, it is a challenging place for a ground-based campaign for a number of reasons, and these challenges ought to be clearly stated near the beginning of the paper with your thoughts on how those challenges may manifest:

a. High latitude means higher SZA

b. Lack of full seasonal cycle

c. Proximity to the polar vortex means increased likelihood of poor a priori profiles as GGG2014 does not handle vortex air

d. Dry atmosphere does not provide full range of XH2O seen in other locations

11. P3L85-85: Awkward sentence. Please clarify.

12. P4L88: Please cite the Karion paper when AirCore is introduced.

13. P5L124: What does "The number of usable detector positions differs for the three instruments." mean in practical terms for this work?

14. P4L136: Please cite: Keppel-Aleks, G., G. C. Toon, P. O. Wennberg, and

N. M. Deutscher (2007), Reducing the impact of source brightness fluctuations on spectra obtained by Fourier-transform spectrometry., Appl. Opt., 46(21), 4774–4779, doi:10.1364/AO.46.004774.

15. P4L138: Please state that this is the GGG2014 software that you are using.

16. P4L141: Are you recording just on the InGaAs detector alone? Are ghost corrections performed?

17. P7L187: build –> built

18. P8L232: atmospheric ** and local oscillator *beams* are mixed

19. P8L244: 13.4 mbar for ** ambient pressure*s between the surface and* 232 mbar, and 3.9 mbar for ** ambient pressure*s* lower than 232 mbar.

20. P9L260: Why are you listing this CO value in ppb?

21. P9L275: The sentence beginning with "The continuous operation" is awkward. Please revise.

22. P10L293: All teams performed *a* full functionality test. . .

23. P10: Change "highest OPD" to "maximum OPD".

24. P11L312: documentation (no 's')

25. P11L322: Is the surface pressure identical between instruments?

26. P11L328: What scaled ratio are you referring to? You refer to scaling several times throughout the paper and it is not clear what you are referring to. Please clarify.

27. P11L335: These two lines are repetitive with information stated earlier in the text.

28. P12L350: State your reasoning for the SZA cutoff earlier.

29. In general, I do not think that uncommon phrases, and/or phrases that are used only a few times in a manuscript should be given acronyms. I find these acronyms

detract from the readability of the paper. For example, I would appreciate it if you would type out the acronyms for:

a. LO

b. FO

c. NA

d. ME (though admittedly common in our field, but used only a few times)

e. PE (again, common in our field)

30. P13: consider reporting XCH4 differences in ppb instead of ppm.

31. In general, in these Xgas-specific subsections: I do not think the seasonal cycle needs to be described in words.

32. P14L430: "Any difference relative to the ideal case is an indicator for the instrument and retrieval code performance." And the spectroscopy, which can be distinct from the code itself. Please add.

33. P14L440: Again, the EM27/SUN data are processed using *an entirely different retrieval code with different spectroscopy* so it is not surprising (or an indication of instrument performance) that the EM27/SUN Xair shows the smallest airmass dependence. Please clarify.

34. P15L451: "However, no such instrument specific calibration factors were applied for the other instruments...". Why not? You know the biases wrt TCCON now.

35. P15L457: Why not just truncate the HR interferograms instead of recording special HR125LR interferograms?

36. P16L499: Again, without the LR AKs to compare with, we cannot assess the impact of errors in the prior on the total column comparisons.

37. P17S5.4.4: Is there a figure this section refers to?

38. P18L553: This paragraph is a non sequitur wrt the previous paragraph regarding XCO.

39. P19L576: "From the plots it can be seen that the SZA dependency... is related to the spectral resolution and the AK of the instruments." While I agree that it's probably true, it cannot be seen from the plots, since you have not shown the LR AKs.

40. P20L606: Any idea why this is occurring? This is a potentially interesting result.

41. P20L617: Again, these have different retrieval code and different spectroscopy. Please make that clear.

42. Section 5.7.1: Suggest moving to an appendix for the reasons stated above.

43. Section 5.8: Suggest putting brief sentence earlier in manuscript, and moving this section to Appendix A for the reasons stated above. Also, "These higher values could come from the spectral double passing of the signal within the interferometer." This sounds interesting but requires far more explanation. Where is the double-passing coming from? How can it be removed?

44. P22L687: "the annex 1" –> Appendix A.

45. P25L775-780: This discussion is confusing. Please clarify.

46. P26L817: This is not surprising, because the GGG2014 TCCON CO prior is a climatology. It will generally not capture pollution events.

47. Section 5.9.2: Move to Appendix.

48. P29L891: The airmass dependence of the retrievals is an effect of the software *and spectroscopy*.

49. P29L892: What is this "airmass correction factor" you're speaking of? Please make this language clearer throughout the paper, and define your terms.

50. P29L904: showed → provided

51. Figures. In general, there are far too many figures, the font size on the figures is far too small, the point sizes are too small, the point styles are indistinguishable, and there are too many points on the figures. This will be even more problematic after typesetting when the sizes of the figures are shrunk to fit into the AMT two-column format. Consider relegating some figures to appendices, using shading instead of individual points, removing whitespace between multi-panel figures, simplifying the content, etc.

52. Figure 6. Does the "PF" at the end of the HR125LRPF indicate "PROFFAST"? If so, please state clearly in the caption.

53. Figure 17. When plotting vertical profiles in the context of a total column retrieval, it is more natural to plot the vertical axis in linear pressure. This is because it better approximates the mass-weighting that the total column represents. Please change these plots to display them in linear pressure. Also, you show column averaging kernels for TCCON up to 85 SZA, but do not use them above 75 SZA. Why plot the other two SZAs? Furthermore, you do not show the HR125LR, EM27/SUN, Vertex, IRcube AKs, which is important to understand how the differences between the a priori profile and the true profile would affect the total column. Please add the AKs of the LR instruments (if they are basically the same, you can just add one representative set).

54. Figure 18. What exactly is being plotted? 10 minute averages or all individual points? If averages, please make this clear in the caption, and describe the number of points being averaged in each averaging period for each instrument.

55. Figure 21. If you are comparing data to a reference (in this case, AirCore is the reference), I believe it is customary to put the reference on the x-axis. You have plotted TCCON on the x-axis here.

56. Tables: In general, some tables could be combined and descriptive text should be in the caption, not embedded in the table (e.g., Table 4).

---

## Author Response (AR1)

mahesh.sha@aeronomie.be                                    18 March 2020

Letter to the Editor

for manuscript by **Sha et al.**, MS No. **amt-2019-371**; title: "**Intercomparison of low and high resolution infrared spectrometers for ground-based solar remote sensing measurements of total column concentrations of CO₂, CH4 and CO**".

Dear Editor,

We would like to thank you for acting as the Editor for our manuscript and providing positive comments during the start of the review process.

We have revised our manuscript carefully taking into account all of the comments made by the two referees.

Both referees suggested a restructuring of the paper and to put emphasis on the non-linearity corrected TCCON data. We found this suggestion useful. In the revised version of the paper, we therefore start our results section with the description of the non-linearity (section 5.2), followed by the discussion on the performance of the Bruker IFS 125HR (both standard and non-linearity corrected data sets) against the AirCore. We then discuss the intercomparison results of the low-resolution test instruments with respect to the non-linearity corrected TCCON data set. In addition, we have moved the discussions of the uncorrected reference data sets to the appendix. The sections related to the solar zenith angle (SZA) and humidity dependence of the bias have also been moved to the appendix following the suggestion of the referee. Keeping in line with the above-mentioned changes, all associated figures have been updated. The restructuring only changed the ordering of the sections, but our lessons learned and the conclusions drawn from the campaign remain the same.

We hope that the revised version of the manuscript now meets the high quality standards for publication in the journal "Atmospheric Measurement Techniques".

With kind regards,
Mahesh Kumar Sha (on behalf of all authors)

**Response to comments from Referee 1 (Dr. Ruediger Lang)**

Black: Referee's comments; Blue: Authors' answers

We thank referee #1, Dr. Ruediger Lang for the review and for providing useful feedback.

Referee:

Accurate ground based observations of carbon dioxide and methane are becoming increasingly important in the context of continuous satellite remote sensing validation. While ground based in-situ and remote sensing networks measuring $CO_2$ and CH4 concentrations in the atmosphere exist, and are instrumental in measuring the continuous increase in greenhouse gases over the past decades, it has been recognised that the existing network of stations is still lacking with respect to its expected role in future monitoring and verification system-of-systems (MVS) of greenhouse gas emissions. Ground based remote sensing instruments like the Total Carbon Column Observing Network (TCONN) FTS instruments measuring total column dry air mole fractions of carbon dioxide and methane ($XCO_2$ and XCH4) are key for the validation and calibration of future operational satellite based observing systems, targeting greenhouse gas emissions. However, the current distribution of stations and their representability for the validation of global satellite based measurements, as well as their maintenance and operability for providing a continuous flow of quality-monitored data are providing big challenges ahead for the network to become the much aimed for fiducial greenhouse-gas reference measurements within an operational MVS context. One of the key-challenges is achieving and maintaining the very high accuracies needed – significantly below 0.5 ppm for $XCO_2$ – in order to become useful for the MVS and the Cal/Val of its satellite components.

The paper by Sha et al. is an important and significant contribution towards the latter aspect, by addressing the question on how to secure high accuracies across the network, e.g. through travelling instrument standards, while identifying, correcting and potentially even reducing instrument and measurement biases. The key finding of the paper is, from my point of view, the potential of at least one of the instruments taking part in the campaign exercise at Sodankyla, Finnland, functioning as a "travelling standard" in a potential future operational network of reference FTS instruments as currently operated under the TCCON umbrella. The paper also addresses important open questions with respect to remaining systematic biases and points at, maybe even more significant, remaining issues in measurement bias and precision. The paper is well written, although I think the paper could benefit from some restructuring of the results sections. I therefore recommend the paper for publishing in ACP, but would like to highlight in the following a couple of observations (apart from some additional minor comments), which the authors, from my point of view, should address.

Authors' response:

Thanks for the positive comments. As the paper focuses on the measurements of atmospheric components using ground-based instruments, we think it is better suited for publication in AMT rather than ACP and would like to keep it here.

Referee:

1) Ari-core comparisons and non-linearity effects

The discussion of the non-linearity effect found in the measurements of the Bruker IFS HR reference instruments (referred to as TCCON "reference", 125HR), as well as the comparison to the AirCore measurements, as taken during the campaign at Sodankyla - which are considered to represent the "true atmospheric" state - are both presented only at the end of the major results

section 5. This is first of all confusing, since the section on non-linearity effects implies that the non-linearity correction "has been applied to TCCON" data as a whole. At this stage, the reader wonders if this therefore now applies to all previous results, but the answer is probably not, since the results and different labelling then implies that the main comparisons results are not (yet) non-linearity corrected.

Second, the performance of the "reference instruments" with respect to what is considered the "true state of the atmosphere" represented by the AirCore measurements is an important result against which also the results of the measurements from the other systems have to be evaluated and interpreted. So both aspects have to be taken together. I would therefore recommend to present the results on the 125HR instrument non-linearity and the performance of the reference instruments against the "truth" (5.8 and 5.9) at the beginning of Section 5 (and ideally then present only the non-linearity corrected results for the TCONN reference – if this is considered a stable result – in the comparison against the other systems). This would help to interpret the results of the other systems when compared to the "reference" 125HR better with respect to the AirCore "truth". In addition, establishing the TCCON 125HR as a "reference" and therefore then talking about a "bias" with respect to the reference for the other instruments (and not in terms of "differences"), requires, in my view, presenting the AirCore results first anyhow.

Authors' response:
Thanks for this suggestion.
Our original idea was to present the comparison results first with the official TCCON data and then show the results of the non-linearity corrected TCCON data, which was part of the lessons learned during the campaign. We agree to the referee's viewpoint and have moved the (former) sections 5.8 and 5.9 to the beginning of Section 5. In addition, we have moved the other intercomparison results with the official TCCON data and the HR125LR data to the appendix. These results are useful as a reference comparison to the official TCCON data. They also provide useful analysis on the resolution dependent effects on the Xgas retrievals of the target gases.

Referee:
2) EM27 systematic biases with respect to the two references The EM27 (COCCON) instruments show a convincing performance with respect to the "reference" TCCON (125HR), both with respect to the systematic biases and precision (for the latter see below). This is alas true especially with respect to the non- corrected TCCON results. It seems peculiar that this bias is even lower than -0.2 ppm for the non-corrected comparisons, but then gets worse for the TCCON non-linearity corrected ones (-0.73 ppm). At the same time, the low-resolution "reference" measurements (125LR) show a high bias to EM27 of roughly the same order of magnitude than the effect of the non-linearity correction. Considering that the low resolution 125HR are more comparable in terms of information content (and probably AKs, although those have unfortunately not been presented –see below), one wonders if the non-linearity correction should not also be applied to the 125LR measurements (or has this been done?), which then may lead to a consistency between EM27 and the 125LR. This would then also physically make a lot of sense considering the information content of both measurements. Also it would be important to rule out any link (e.g. in retrieval processing) between the standard 125HR measurements (not corrected) and the EM27 results, which potentially make them similar to the standard reference

by default (and therefore more biased with respect to the corrected 125HR results and the AirCore "truth"). In this context, the processing algorithm ProFIT vs GFIT performances and their potentially relevant "peculiarities" are not much discussed at all. ProFIT is used both for the TCCON 125LR and the COCCON EM27s but not for the uncorrected TCCON "reference" it seems. Is the bias correction scheme used in ProFIT maybe somehow related to TCCON measurements (or associated climatologies?).

I think it would be important to add a discussion of the (potential) relationship between the observed systematic biases between "reference" HR, LR, and non-linearity correction on one side, and the EM27 measurement results on the other side, which could shine some light on the underlying mechanisms.

Authors' response:
We have added a description of the PROFFAST code in the paper and the underlying differences with respect to the GFIT code. See lines 173 – 188: "PROFFAST is a code for retrieving trace gas amounts from low-resolution solar absorption spectra. It has been developed on behalf of ESA, in order to provide a source-open and freely available code (without any licensing restrictions) as required by the growing COCCON user community, e.g. for TROPOMI validation work. It is a least-squares fitting algorithm, which adjusts the trace gas amounts by scaling atmospheric a priori profiles. The retrievals are performed on spectra generated with the included PREPROCESS tool. This tool produces spectra out of the measured DC-coupled EM27/SUN interferograms. It includes a DC correction of the interferogram, a dedicated phase correction scheme for double-sided interferograms and several quality control tests (e.g. testing for the presence of out-of-band artefacts). The lookup table for cross-sections used by PROFFAST is created on the basis of HITRAN spectroscopic line lists: For H2O, CH4, N2O, HITRAN 2008 line lists are used (in case of H2O including some minor empirical adjustments), for CO2 and CO HITRAN 2012 line lists are used. PROFFAST uses the solar line list compiled by Geoff Toon, JPL, for GGG2014. In contrast to the TCCON GGG2014 processing, the empirical airmass-independent and airmass-dependent post-calibrations are applied species-wise including molecular oxygen. Thereby, the Xair equivalent provided by PROFFAST is on average normalised to unity, while it remains an un-calibrated intermediate result in GGG2014, which calibrates only the Xgas results. The PROFFAST approach of calibrating Xair is transparent for users, as the calibration factors can be directly related to deviations of the spectroscopic band intensities, and gives the user a more sensitive diagnostic tool at hand, as airmass-dependent artefacts in the reported quantity are also reduced."

The Xgas values, which are calculated using GFIT are scaled to the WMO standards using calibration factors (airmass dependent and independent). The calibration factors were derived from dedicated campaigns at different TCCON sites. However, the $XCO_2$ and $XCH_4$ products from the EM27/SUN are bias-corrected based on the scaling factors calculated from the extensive COCCON development. The residual bias between the non-linearity corrected TCCON data and the EM27/SUN might be due to several reasons, such as (1) effect of non-linearity correction of the TCCON data on the retrieved Xgas values, (2) bias of the EM27/SUN due to the imperfect instrument specific scaling factor used which has been determined independently prior to this study from long-term intercomparison measurements performed at the KIT TCCON site.  (3) bias of the Sodankylä TCCON station which may come from the imperfect use of the airmass

independent calibration factor derived for the global TCCON. As more comparisons of the COCCON spectrometers with respect to the TCCON stations takes place in the future we need to verify the site-to-site bias of the TCCON with respect to the COCCON spectrometer. As mentioned by the referee as well, the use of the low-resolution spectrometer as a travelling standard will be very useful in this respect.

The HR125LR measurements (low-resolution measurements with the Bruker IFS 125HR) are also affected by the non-linearity of the InGaAs detector. As this is not our standard data set, no non-linearity correction has been done. It is used to check the resolution dependent effects of the measurements as discussed in the paper. For example, the intercomparison results help us to verify the bias in XCH4 during the springtime (as high as 0.01 ppm) which is due to the large difference between the a prior profile to the true atmospheric state and the influence of the different AKs between the low- and high-resolution spectrometers. The $XCO_2$ bias between the EM27SUN vs the HR125LR is 0.56 ppm. If we apply similar non-linearity correction to the HR125LR data, then the expected $XCO_2$ bias is about 0.06 ppm, XCH4 is about 0.006 ppm and XCO is about 4.5 ppb.

3) Averaging kernels, low res versus high res

The differences in performance and measurement information content for the low-resolution measurements (125LR and EM27) are of high importance, especially with respect to the fact that some of the forthcoming remote sensing satellite based systems may be operated at lower spectral resolutions, i.e. being more similar to the LR measurements in terms of spectroscopy. Also some of the differences observed between AirCore truth, high resolution and low resolution measurements, as discussed under 2), may be interpreted in this respect. The performance of these lower resolution systems are of importance also with respect to the knowledge of spectroscopy and potential future research needs there. But mainly the differences in performance between LR and HR (also with respect to AirCore "truth") may be better interpreted if the differences in their respective AK would be discussed and addressed - at least to some extend (and potentially also differences in GFIT and ProFIT a priori profiles if any – see also 2)). This could be done e.g., and ideally, in a corresponding figure to Fig. 17 (or adding such results to the latter).

Authors' response:
We completely agree to the referee's point of view. It is very important to identify differences caused by the resolution differences of the spectrometers and their related effects. As a result we have kept the HR125LR comparison results in the appendix. The AKs of the TCCON and the EM27/SUN for all SZA are shown in Figure 6 of Hedelius et al. (2016). The retrieval of the low and high-resolution measurement data sets were done using the TCCON a priori as the common a priori. As discussed in point 2 above, other differences between the PROFFAST and GFIT codes have been added to the paper.

Referee:
4) Precision of TCCON versus COCCON

The striking difference in measurement noise (Fig. 18) between TCCON and COCCON EM27 is not much discussed it seems. Is this feature a question of retrieval algorithm performance (or/and applied constraints therein), spectral resolution (like the more stable LR with respect to HR measurements), or is it solely related to instrument noise?

Authors' response:

The EM27/SUN shows a lower scatter as compared to the TCCONmod due to the low noise resulting from the averaging of the individual measurements. The individual measurement time for each instrument is provided in Table 2 in the paper. Within the period of five minutes, it is possible to average five measurements for the EM27/SUN data set. This is equivalent to 50 scans in total in 5 min with 10 scans for each measurement. Whereas a maximum of only two measurements are possible for the TCCONmod data set. This is equivalent to a maximum of 8 scans in 5 min with 4 scans for each measurement.

The scatter in the IRcube and Vertex70 is comparable to the TCCONmod due to the averaging of the similar number of measurements within the five minutes time interval.

Additional minor comments:

Referee:

p. 2, l.36ff: "Carbon dioxide (CO2) and methane (CH4) are the two main components of the carbon cycle of the earth's atmosphere. They absorb and retain the heat in the atmosphere causing the greenhouse effect and global warming" – I would remove "greenhouse effect" here. Otherwise, water vapour would have to be mentioned in this context.

Authors' response:

Done

Referee:

Section 5.3: I think it would be would be very helpful to introduce here the logic of the following sections (titles content) to guide the reader through what is coming, apart from potentially shifting the AirCore results and non-linearity sections first (see above).

Authors' response:

We agree. We now begin the results section with a description of the non-linearity issue with our reference measurement "TCCON". This is followed by a comparison of the standard and non-linearity corrected TCCON results with respect to the AirCore.

Referee:

Figure 1 to 4: It would be helpful to use the same axis limits in the scatter plots.

Authors' response:

Done. Note that these are now Figures 7 – 10 which are made for the TCCONmod case with the same axis for the scatter plots for each instrument except for the $XCO_2$ plot for the LHR instrument where the scatter is significantly higher as compared to the other instruments.

Referee:

Figure 16: A horizontal dashed line or similar at zero would really help to interpret the results. Especially since the range of the y-axis has to be quite large.

Authors' response:
Done

Referee:
Section 5.4.4. I think here it should be highlighted and maybe discussed in context of the discussion on biases between TCCON, TCCONmod, LR and EM27, why the EM27 is so much closer to one than TCCON (see also point 2) above).

Authors' response:
We have added an explanation in response to point 2 above and we included it in the last paragraph of section 5.5, see lines 631 – 642: "The Xgas biases between the low-resolution test instruments and the TCCONmod data sets as reference may be due to effects such as different responses to a priori profiles, interfering species in the retrieval windows or different averaging kernels. Furthermore it is important to note that, TCCON uses a network-wide constant scaling factor to scale its Xgas values to the WMO standards. The scaling factors specific to each gas for TCCON had been determined from several measurement campaigns where vertically distributed measurements of the gases were performed from airborne platforms using WMO calibrated instruments. The EM27/SUN uses species dependent scaling factors for XCO2 and XCH4 which had been calculated from long-term intercomparison measurements performed at the KIT TCCON site. However, no such instrument specific calibration factors were applied for the other instruments and also for the XCO results from the EM27/SUN measurements. This also contributes to the residual bias which is observed in this intercomparison result. The biases which are purely due to resolution differences are addressed by performing low-resolution measurements with the same TCCON instrument. These data are then used for an intercomparison relative to the TCCON as well as for the intercomparison with other low-resolution test instruments. Further details of the intercomparison results are given in appendix C and D, respectively."

**Response to comments from Referee 2**

Black: Referee's comments; Blue: Authors' answers

We thank referee #2 for the review and for providing useful feedback.

Review of Sha et al.: Intercomparison of low and high resolution infrared spectrometers for ground-based solar remote sensing measurements of total column concentrations of $CO_2$, CH4 and CO

This paper details the results from an intensive campaign wherein 4 portable spectrometers (Bruker EM27/SUN, Bruker IRcube, Bruker Vertex40, RAL LHR) were hosted near the Sodankylä TCCON station from March-September 2017. AirCore profiles were also flown throughout the campaign. The campaign resulted in some interesting data collected that helped reduce uncertainties in the Vertex40, pointed out where the LHR needs improvement, and identified a problem unique to the Sodankylä TCCON station that was resolved.

This paper may be suitable for publication after the following concerns are satisfactorily addressed.

Major Comments

Referee:
1. The non-linearity problem at Sodankylä is unique to Sodanklyä, and this is not made clear in the paper. In general, TCCON stations have limited the light incident on the detectors by a combination of reducing the input field stop (on the aperture wheel) and placing an aperture stop after the input CaF2 window. From what I can glean from Section 5.8 (specifically, "The TCCONmod data set is a better representation of the true atmospheric signal. As TCCON is our primary data reference for the intercomparison study for this campaign, the non-linearity correction has been applied to the TCCON data."), this non-linearity correction has been applied to all comparisons/figures/tables throughout the paper (except where the non-linearity is discussed directly). If I have interpreted this correctly, I believe it is the correct approach, since it is more representative of the data a TCCON station would produce. That said, it should be stated up front (i.e., in section 2.2.1) that this correction has been applied, with a clear statement that this does not generally affect all TCCON data; just Sodankylä's. The details included in section 5.8 (and 5.9) ought to be relegated to Appendix A.

Authors' response:
Although the referee is correct in pointing out that non-linearity is not a problem that affects all TCCON stations, it is as well unjustified claiming that it is a special problem affecting only the single TCCON site at Sodankylä. The Sodankylä TCCON station is a well-maintained and carefully operated site performing measurements since 2009. The input field stop on the aperture wheel was set to 1 mm and the aperture stop after the $CaF_2$ window was set to 32 mm following the standard TCCON recommendations. However, it was only during this campaign, while comparing with the EM27/SUN spectrometer, that we discovered the issue with the non-linearity and its associated influences on the trace gas results. This is a very important finding of this campaign. The detection of non-linearity should be incorporated in the TCCON processing chain (as it is done for COCCON), flagging spectra with non-linearity issues before the trace gas analysis is attempted. As this has not been done in the TCCON data quality management, it may well be possible that other sites might also be affected by non-linearity and remain unnoticed during a longer period of time.

In our paper, the non-linearity corrected TCCON data are labelled as "TCCONmod" and the standard TCCON data are labelled as "TCCON". The standard TCCON data is also what is made publicly available via the TCCON data repository. The TCCONmod data is a product of this campaign and not yet submitted to the TCCON database. Following the suggestion of both referees, we have moved the section of the discussion with the non-linearity corrected TCCON data to the beginning of the results section and the discussions with the uncorrected data to the appendix.

Referee:
2. Your comparisons of Xair between the various instruments is misleading because it is not an apples-to-apples comparison. It is not only instrument effects that impact Xair: it is also

spectroscopy. PROFFAST and GGG2014 do not include the same spectroscopy, line shapes, etc., and thus would be expected that they have different Xair values. It would be helpful to have a short (1-paragraph) discussion about the differences between the two retrieval software algorithms, as relevant to this issue. It is unclear to me why you would choose to use PROFFAST to analyse the EM27/SUN data and not also use the EGI wrapper for GGG2014 to maintain consistency with the other retrievals. I suggest you add the EGI retrievals of the EM27/SUNs to your analysis. (https://bitbucket.org/em27gi/egi/wiki/Home)

Authors' response:

The PROFFAST software has been used for evaluating the performance of the EM27/SUN, because this is the official code to be used for the COCCON analysis. From this perspective, we demonstrate an end-to-end evaluation of instrument hardware and data processing, as targeted by the FRM4GHG project. The PROFFAST code has been developed on behalf of ESA in order to provide a source-open and freely available code (without any licensing restrictions) as required by the growing COCCON user community. Note that the use of an independent code does not imply generating "oranges" instead of "apples" as both processors intend to provide an optimal reconstruction of the true atmospheric state. Instead, it allows us to uncover the discrepancies introduced by the various design considerations of the pre-processing, the retrieval codes and generated outputs. We agree that a future alignment of line lists between GFIT and PROFFAST is desirable for revealing those changes introduced by more subtle aspects of the setups (e.g. treatment of background continuum). It is therefore planned to upgrade PROFFAST to the line lists used by the upcoming GFIT version in the framework of a follow-up project. We also have added a paragraph describing some details of the PROFFAST setup.

See lines 173 – 188: "PROFFAST is a code for retrieving trace gas amounts from low-resolution solar absorption spectra. It has been developed on behalf of ESA, in order to provide a source-open and freely available code (without any licensing restrictions) as required by the growing COCCON user community, e.g. for TROPOMI validation work. It is a least-squares fitting algorithm, which adjusts the trace gas amounts by scaling atmospheric a priori profiles. The retrievals are performed on spectra generated with the included PREPROCESS tool. This tool produces spectra out of the measured DC-coupled EM27/SUN interferograms. It includes a DC correction of the interferogram, a dedicated phase correction scheme for double-sided interferograms and several quality control tests (e.g. testing for the presence of out-of-band artefacts). The lookup table for cross-sections used by PROFFAST is created on the basis of HITRAN spectroscopic line lists: For $H_2O$, $CH_4$, $N_2O$, HITRAN 2008 line lists are used (in case of $H_2O$ including some minor empirical adjustments), for $CO_2$ and CO HITRAN 2012 line lists are used. PROFFAST uses the solar line list compiled by Geoff Toon, JPL, for GGG2014. In contrast to the TCCON GGG2014 processing, the empirical airmass-independent and airmass-dependent post-calibrations are applied species-wise including molecular oxygen. Thereby, the Xair equivalent provided by PROFFAST is on average normalised to unity, while it remains an un-calibrated intermediate result in GGG2014, which calibrates only the Xgas results. The PROFFAST approach of calibrating Xair is transparent for users, as the calibration factors can be directly related to deviations of the spectroscopic band intensities, and gives the user a more sensitive diagnostic tool at hand, as airmass-dependent artefacts in the reported quantity are also reduced."

Referee:

3. It is unclear to me why you include the time period before the hardware upgrades of the Vertex and IRcube in your subsequent analysis. It makes more sense to me to show the significant improvement in their data after fixing the hardware in the time series plot, and then do not show the pre-improvement data in subsequent analyses, focusing only on the "good" data.

Authors' response:

The campaign began with an initial blind intercomparison phase where the instruments were operated with the optimised setting best known to their PIs to get good SNR comparable to the TCCON. However, it was found that the settings were not optimal for Vertex70 and the improvements were significant after the instrument modification. The change of the optical fibre also affected the IRcube results. Therefore, we find it relevant to report these data. The other instruments did not undergo any modification and therefore the data for the whole period is considered as "good" data. In the paper, we present two sets of results for each intercomparison section where the results from the full year are relevant for the instruments, which did not undergo any modification, and the results for the shorter period, which are focusing on the comparison of the Vertex70 instrument results relative to the other instruments for the same period.

Minor comments

Referee:

1. I found the authors' motivation for the need for the COCCON (or another complement to TCCON) misleading. I believe it is true that the TCCON could be usefully supplemented by LR portable spectrometers, but the atmospheric and surface conditions you list are generally already covered by TCCON stations. For example:

a. "A denser distribution of ground-based solar absorption measurements is needed to cover various atmospheric conditions (humid, dry, polluted, presence of aerosol), various surface conditions (high and low albedo) and a larger latitudinal distribution."
i. Humid –> Darwin
ii. Dry –> Armstrong, Eureka, Sodankylä, Pasadena
iii. Polluted –> Pasadena, Tsukuba, Wollongong (sometimes)
iv. Aerosol –> Pasadena, Tsukuba, sometimes Ascension
v. High albedo –> Armstrong
vi. Low albedo –> Park Falls, etc.

b. Where I do agree, is that if we want to cover geographic gaps in the locations of these stations, we need more stations, and the low-resolution instruments may be well suited to that.

Authors' response:

Taking the referee's comments into account we have added further details in the paper (see lines 4 – 9) to clarify our statement:

"The number of stations in the network (currently about 25) is limited and has a very uneven geographical coverage: the stations in the Northern hemisphere are distributed mostly in North America, Europe, Japan and only 20% of the stations are located in the Southern hemisphere

leaving gaps in the global coverage. A denser distribution of ground-based solar absorption measurements is needed to improve the representativeness of the measurement data for various atmospheric conditions (humid, dry, polluted, presence of aerosol), various surface conditions such as high albedo (> 0.4) and very low albedo and a larger latitudinal distribution."

Referee:
2. It is unclear to me whether you use identical surface pressure values in your retrievals for each instrument. If they are different, this would also cause biases in Xair. For a fair comparison, they should be identical and calibrated to a meteorological standard. Please discuss and resolve if necessary.

Authors' response:
Yes, all spectrometers used the identical set of ground-pressure data collected at the Sodankylä site. We have added this information to the paper (see line 363): "The spectrometers used an identical set of ground-pressure data collected at the Sodankylä site for the retrieval."

Referee:
3. It is unclear to me why the HR125LR has its own section. Why not include it in the comparisons with the other LR instruments wrt TCCON?

Authors' response:
The HR125LR data set provides interesting results when compared to the reference data set. It is a unique data set as both HR125LR (low-resolution) and the reference high-resolution measurements were performed using the same instrument. In order to highlight these features we have provided a separate section for the discussion of the results.

Referee:
4. The water vapour dependence section is not conclusive, since Sodankylä is generally dry (XH2O <4500 ppm). This is therefore not an exhaustive test of XH2O dependence. I suggest moving this section to an appendix, noting in the main text that little XH2O dependence was found over the (relatively dry) conditions at Sodankylä.

Authors' response:
Sodankylä is not the most humid TCCON site. The maximum XH2O measured by the TCCON is < 6000 ppm during the summer period (see Figure 1 below). In comparison, the TCCON site at Darwin, which is a relatively humid site, show a maximum measured XH2O of < 10000 ppm during the summer period. The year 2017 was relatively dry where the range of XH2O measured at the Sodankylä site was between 500 and 4500 ppm. Following the suggestion of the referee, we have moved the detailed discussion to the appendix F and introduced a small paragraph as section 5.6 in the main text focusing on the main results.

[Figure]

Figure 1: Timeseries of the XH2O measured at the Sodankylä TCCON site.

Technical comments

Referee:

1. P1L1: … the baseline *ground-based* network of instruments …

Authors' response:

Done

Referee:

2. P1L5: Northern America –> North America and again in L72.

Authors' response:

Done

Referee:

3. P2L25-26: This seems to imply that the non-linearity is a problem throughout TCCON, which it is not. Please revise.

Authors' response:

We have added further explanation to emphasise that the non-linearity results are for the Sodankylä campaign. See lines 21 – 25: "The reference measurements performed with the Bruker IFS 125HR were found to be affected by non-linearity of the Indium Gallium Arsenide (InGaAs) detector. Therefore, a non-linearity correction of the 125HR data was performed for the whole campaign period and compared with the test instruments and AirCore. The non-linearity corrected data (TCCONmod data set) show a better match with the test instruments and AirCore data as compared to the non-corrected reference data."

Referee:

4. P2L41: increasing in ** recent years (no "the")

Authors' response:

Done

Referee:

5. P2L43: You may want to mention the (important) role of VOCs in the production of CO.

Authors' response:

Done

Referee:

6. P2L48: positive radiative forcing*, therefore it is* considered as an indirect …

Authors' response:

Done

Referee:

7. P3 first paragraph: Generally unnecessary paragraph. It's unclear what you mean by "To ensure equal dependency on the input spectral data, …"

Authors' response:

This paragraph provides the rationale for the ground-based total column measurements and its usefulness for satellite validation. Following the suggestion of the referee, we have added further explanations in the paragraph to make our statements clearer for the reader.  See lines 63 – 65: "To ensure equal dependency on the measurement parameters, the best validation method for the satellite data is to use the total column amounts of the trace gases calculated from the solar absorption measurements performed from the surface and the satellite in the same spectral region."

Referee:

8. P3L74: Again, there are TCCON stations that span all those conditions. Rephrase.

Authors' response:

We have modified the sentence, see lines 76 – 79: Furthermore, for the complete validation of the satellite data set, a denser distribution of ground-based solar absorption measurements is needed to cover geographical gaps and to improve the representativeness of the measurement data for various surface and atmospheric conditions (e.g., high and very low surface albedo, pollution, aerosol presence, humid, dry).

Referee:

9. P3L80. "However, there has been little characterization, intercomparison and harmonization of these new instruments in comparison to the standard instrument used in TCCON." There is some literature on just this topic. Please cite:

a. Hedelius, J. K., C. Viatte, D. Wunch, C. M. Roehl, G. C. Toon, J. Chen, T. Jones, S. C. Wofsy, J. E. Franklin, H. Parker, M. K. Dubey, and P. O. Wennberg (2016), Assessment of errors and biases in retrievals of XCO2, XCH4, XCO, and XN2O from a 0.5 cm-1 resolution solar-viewing spectrometer, Atmos. Meas. Tech., 9(8), 3527–3546, doi:10.5194/amt-9-3527-2016.
b. Hedelius, J. K., H. Parker, D. Wunch, C. M. Roehl, C. Viatte, S. Newman, G. C. Toon, J. R. Podolske, P. W. Hillyard, L. T. Iraci, M. K. Dubey, and P. O. Wennberg (2017), Intercomparability of XCO2 and XCH4 from the United States TCCON sites, Atmos. Meas. Tech., 10(4), 1481–1493, doi:10.5194/amt-10-1481-2017.

Authors' response:

Done

Referee:

10. While I understand that Sodankylä was chosen for (important) practical considerations, it is a challenging place for a ground-based campaign for a number of reasons, and these challenges ought to be clearly stated near the beginning of the paper with your thoughts on how those challenges may manifest:

a. High latitude means higher SZA

b. Lack of full seasonal cycle

c. Proximity to the polar vortex means increased likelihood of poor a priori profiles as GGG2014 does not handle vortex air

d. Dry atmosphere does not provide full range of XH2O seen in other locations

Authors' response:

a. We have added an explanation in the text. See lines 110 – 113: "Due to the location of the site at a high latitude, measurements are possible for solar zenith angle (SZA) range between > 43° and < 90°. The coverage of high SZAs is important to check the dependence of the airmass on the retrieval results. The airmass dependent correction factor applied to the remote sensing data is relevant for measurements at higher SZA."

b. Yes, the Sodankylä TCCON site lacks measurements during Nov – Jan. However, it has the benefit of having high variability of signal (e.g. for $CO_2$) during the year.

c. The same a priori was used for all instruments. The low-resolution instruments are aiming to complement TCCON type measurements. It is therefore very interesting to compare them also in conditions where TCCON stations exists.

In addition, validation of the XCH4 product from the TROPOMI instrument on board the Sentinel-5 Precursor (S5P) with respect to the TCCON station at Sodankylä shows a high bias during the spring period as compared to the rest of the year (see Figure 2 below). It is therefore very interesting to identify the cause of the bias. Measurements performed with the low-resolution ground-based instruments during this campaign will help us to understand some of the causes of the bias.

d. The year 2017 was a dry year in general. Typically, XH2O values for Sodankylä cover higher range and reach up to 6000 ppm. However, we agree that comparing the low-resolution instruments for conditions with very high humidity would be useful in a future campaign.

As mentioned in the paper, after carefully analysing the available site options and the requirements to host simultaneously all instruments and the availability of AirCore facilities, we have selected the Sodankylä TCCON site for our campaign.

[Figure]

Figure 2: Relative difference of the XCH4 from the (S5P-TCCON)/TCCON data sets for the Sodankylä site. The coincidence criteria for the validation are given in the presentation by Sha et al. 2019.

Referee:
11. P3L85-85: Awkward sentence. Please clarify.
Authors' response:
Indeed, it was a long complex sentence. Therefore, we replaced it with two sentences to convey our message. See lines 88 – 92: "For this reason in 2017, the European Space Agency (ESA) initiated an intercomparison campaign within the project Fiducial Reference Measurements for Ground-Based Infrared Greenhouse Gas observation (FRM4GHG). The campaign was performed in Sodankylä (Finland) with the aim to assess the performance of different spectrometric instruments for remote sensing of atmospheric trace gases and to quantify their performances regarding precise measurements of column-averaged dry-air volume mole fractions of $CO_2$, CH4 and CO."

Referee:
12. P4L88: Please cite the Karion paper when AirCore is introduced.
Authors' response:
Done

Referee:
13. P5L124: What does "The number of usable detector positions differs for the three instruments." mean in practical terms for this work?
Authors' response:

We have elaborated this point by providing further details on the number and type of detectors for each instrument in the paper. This is mentioned in Table1. See lines 131 – 135: "The number of usable detector positions differs for the three instruments. The EM27/SUN can accommodate two room temperature (RT) Indium Gallium Arsenide (InGaAs) detectors covering different frequency ranges. Also the Vertex70 can accommodate two detectors, one InGaAs and a second channel with either a liquid nitrogen (LN2) cooled Indium Antimonide (InSb) or an RT InGaAs detector. The IRcube can only accommodate one InGaAs detector and has no room for a second detector."

Referee:
14. P4L136: Please cite: Keppel-Aleks, G., G. C. Toon, P. O. Wennberg, and N. M. Deutscher (2007), Reducing the impact of source brightness fluctuations on spectra obtained by Fourier-transform spectrometry., Appl. Opt., 46(21), 4774–4779, doi:10.1364/AO.46.004774.
Authors' response:
Done

Referee:
15. P4L138: Please state that this is the GGG2014 software that you are using.
Authors' response:
Done

Referee:
16. P4L141: Are you recording just on the InGaAs detector alone? Are ghost corrections performed?
Authors' response:
Yes, the double-sided DC coupled interferograms at 0.5 cm-1 are recorded using the InGaAs detector. The laser sampling error (LSE) caused by any asymmetry is minimized by collecting data employing the interpolated sampling option provided by Bruker. Gisi, 2014 showed that no ghost were found when the option of interpolation was enabled with the M16 electronics. In our measured spectra, we did not find any LSE ghost.

Referee:
17. P7L187: build –> built
Authors' response:
Done

Referee:
18. P8L232: atmospheric ** and local oscillator *beams* are mixed
Authors' response:
Done

Referee:
19. P8L244: 13.4 mbar for ** ambient pressure*s between the surface and* 232 mbar, and 3.9 mbar for ** ambient pressure*s* lower than 232 mbar.

Authors' response:
Done

Referee:
20. P9L260: Why are you listing this CO value in ppb?
Authors' response:
We reported it now in ppm.

Referee:
21. P9L275: The sentence beginning with "The continuous operation" is awkward. Please revise.
Authors' response:
Done. See lines 302 – 303: "The measurements preformed helped to observe diurnal variation of the target gases."

Referee:
22. P10L293: All teams performed *a* full functionality test …
Authors' response:
Done

Referee:
23. P10: Change "highest OPD" to "maximum OPD".
Authors' response:
Done

Referee:
24. P11L312: documentation (no 's')
Authors' response:
Done

Referee:
25. P11L322: Is the surface pressure identical between instruments?
Authors' response:
Yes, all spectrometers used the identical set of ground-pressure data collected at the Sodankylä site.

Referee:
26. P11L328: What scaled ratio are you referring to? You refer to scaling several times throughout the paper and it is not clear what you are referring to. Please clarify.
Authors' response:
We have modified the text giving further explanation on this and added the definition of Xair following TCCON publication.
TCCON uses airmass dependent and airmass independent calibration factors to scale the Xgas values to the WMO standards. The results of the low-resolution instruments analysed with GFIT

were scaled in the same way as TCCON. The scaling factor used for the EM27/SUN results are discussed in Frey et al. 2015.

Referee:
27. P11L335: These two lines are repetitive with information stated earlier in the text.
Authors' response:
We have removed both lines as suggested by the referee and added this information in Section 3.1.

Referee:
28. P12L350: State your reasoning for the SZA cutoff earlier.
Authors' response:
We think that the reasoning for the filtering of SZA cutoff is better suited here. Therefore, we have modified the text in the first paragraph of section 5.1 such that we do not need to mention the filtering criterion there. See lines 348 – 349: "However, these measurements performed were recorded with SZA > 75°."

Referee:
29. In general, I do not think that uncommon phrases, and/or phrases that are used only a few times in a manuscript should be given acronyms. I find these acronyms detract from the readability of the paper. For example, I would appreciate it if you would type out the acronyms for:
a. LO
b. FO
c. NA
d. ME (though admittedly common in our field, but used only a few times)
e. PE (again, common in our field)
Authors' response:
Done

Referee:
30. P13: consider reporting XCH4 differences in ppb instead of ppm.
Authors' response:
The TCCON XCH4 are reported in ppm. As TCCON is our reference, we also stick to reporting XCH4 values and differences in ppm.

Referee:
31. In general, in these Xgas-specific subsections: I do not think the seasonal cycle needs to be described in words.
Authors' response:
We prefer to keep the description of the seasonal cycle as it emphasises the variability of the signal for the Sodankylä site. Also for the comparison of the LHR $XCO_2$ values, we could observe that it was able to capture the annual summer drawdown. Having the description of the seasonal cycle helps to point to this kind of aspects.

Referee:

32. P14L430: "Any difference relative to the ideal case is an indicator for the instrument and retrieval code performance." And the spectroscopy, which can be distinct from the code itself. Please add.

Authors' response:

Done

Referee:

33. P14L440: Again, the EM27/SUN data are processed using *an entirely different retrieval code with different spectroscopy* so it is not surprising (or an indication of instrument performance) that the EM27/SUN Xair shows the smallest airmass dependence. Please clarify.

Authors' response:

Done, please see our reply to point no. 2 of the major comments.

Referee:

34. P15L451: "However, no such instrument specific calibration factors were applied for the other instruments …". Why not? You know the biases wrt TCCON now.

Authors' response:

The EM27/SUN spectrometer participating in this campaign is part of the COCCON network. As discussed in Frey et al., 2015, a species dependent calibration factor is defined for each EM27/SUN spectrometer. The calibration factor has been checked at the Karlsruhe TCCON site before the instrument was shipped for the campaign. The EM27/SUN retrieval results we scaled using these calibration factors. However, the other instruments were deployed for the first time in the configuration discussed in the instrument description section in the paper. Therefore, no such calibration measurements were performed for the other low-resolution measurements before this campaign. As a result, we want to report the absolute bias seen by the individual instrument. We need to investigate further if this bias remains constant over longer period. This is currently under study as part of the extension of the campaign.

Referee:

35. P15L457: Why not just truncate the HR interferograms instead of recording special HR125LR interferograms?

Authors' response:

The recording of low-resolution double-sided interferograms (HR125LR) has added benefits as compared to truncating the high-resolution single-sided interferograms.

1. The acquisition time of the HR125LR interferograms is much shorter as compared to that of HR interferograms. Valid low-resolution measurements can be acquired under weather conditions that are too poor for high-resolution measurements; therefore, there is the possibility of adding more observations with the HR125LR type observations.

2. HR125LR are double-sided interferograms that make the phase correction easier as compared to the HR interferograms that are single sided and truncating it might lead to artifacts. The centre burst of the interferogram is near the ramp-up section of the forward scan.

3. The signal to noise of the truncated HR interferograms will not be comparable to the LR interferograms since most of the observation time is omitted.

Referee:
36. P16L499: Again, without the LR AKs to compare with, we cannot assess the impact of errors in the prior on the total column comparisons.
Authors' response:
The LR and HR AKs are shown in the Figure 6 of Hedelius et al., 2016, this reference is already stated by the referee in point no. 9 of the technical comments.

Referee:
37. P17S5.4.4: Is there a figure this section refers to?
Authors' response:
Yes, this section referred to Figure 5. A description of this has been given in the introduction part of section 5.4 in the original version of the paper.

Referee:
38. P18L553: This paragraph is a non sequitur wrt the previous paragraph regarding XCO.
Authors' response:
Done

Referee:
39. P19L576: "From the plots it can be seen that the SZA dependency … is related to the spectral resolution and the AK of the instruments." While I agree that it's probably true, it cannot be seen from the plots, since you have not shown the LR AKs.
Authors' response:
Please see our reply to point no. 36 of the technical comments.

Referee:
40. P20L606: Any idea why this is occurring? This is a potentially interesting result.
Authors' response:
This is potentially due to the difference in the retrieval methods between the GFIT and PROFFIT. The data from other low-resolution instrument were analysed using GFIT and therefore show similar dependency and lower difference when compared to TCCON.

Referee:
41. P20L617: Again, these have different retrieval code and different spectroscopy. Please make that clear.
Authors' response:
Done, see lines: 930 – 933: "The EM27/SUN and the HR125LR results retrieved with PROFFAST do not show SZA dependence for species where an airmass correction factor, which was previously determined, was applied except for carbon monoxide where no correction was applied. The other instruments show a SZA dependence to some degree. In order to minimise the effect of the SZA, measurements with an SZA<75° should be used for the instruments."

Referee:
42. Section 5.7.1: Suggest moving to an appendix for the reasons stated above.
Authors' response:
Done

Referee:
43. Section 5.8: Suggest putting brief sentence earlier in manuscript, and moving this section to Appendix A for the reasons stated above. Also, "These higher values could come from the spectral double passing of the signal within the interferometer." This sounds interesting but requires far more explanation. Where is the double-passing coming from? How can it be removed?
Authors' response:
We have moved the discussion with the intercomparison results using the uncorrected data to the appendix. This is in line with what has been suggested by the referee. We have added explanation about double-passing in the paper, see lines 389 – 391: "These higher values can be explained by the presence of unintended double passing of the infrared beam in the interferometer that occurs if some radiation is reflected back from the detector system."
If present, then this is a feature of the detector. However, this is not relevant as the signal is in the out-of-band spectral region and not affecting the retrieval of trace gases.

Referee:
44. P22L687: "the annex 1" –> Appendix A.
Authors' response:
Done

Referee:
45. P25L775-780: This discussion is confusing. Please clarify.
Authors' response:
We have added further explanation, see lines 479 – 486: "For example the AK values for $CO_2$ for lower altitudes are >1 for measurements performed at higher SZA, which means that the retrieval will overcompensate any over- or underestimation of the a priori: If the a priori is underestimating the lower partial column values in comparison to the true atmospheric state, then these will be overestimated by the retrieval in the total column amount; and vice versa if the a priori overestimates the lower partial columns then the retrieval will underestimate their contribution in the total column amount. Similar reasoning is applicable to the case where the AK<1 for lower SZA measurements typically at local noon. From Fig. 2 we can see that the TCCON a priori is underestimating during the summer months and therefore the SZA dependence in the bias (TCCONmod - TCCONmodAC) in Fig. 4 can be explained from the shape of the AK and it is higher for the 28 August case as compared to the 15 May measurements.

Referee:
46. P26L817: This is not surprising, because the GGG2014 TCCON CO prior is a climatology. It will generally not capture pollution events.
Authors' response:

We have added this information in the paper. See lines 521 – 523: "The AirCore profile measured on 28 August captured a large signal in the troposphere but it is not seen in the TCCON a priori. The TCCON CO prior is a representation of the climatology and so it will generally not capture pollution events."

Referee:
47. Section 5.9.2: Move to Appendix.
Authors' response:
Following the suggestion of the first referee, we have moved this section to the start of the results section and the discussions with the uncorrected data to the appendix.

Referee:
48. P29L891: The airmass dependence of the retrievals is an effect of the software *and spectroscopy*.
Authors' response:
Done

Referee:
49. P29L892: What is this "airmass correction factor" you're speaking of? Please make this language clearer throughout the paper, and define your terms.
Authors' response:
Done, please see our reply to point no. 2 of the major comments.

Referee:
50. P29L904: showed ! provided
Authors' response:
Done

Referee:
51. Figures. In general, there are far too many figures, the font size on the figures is far too small, the point sizes are too small, the point styles are indistinguishable, and there are too many points on the figures. This will be even more problematic after typesetting when the sizes of the figures are shrunk to fit into the AMT two-column format. Consider relegating some figures to appendices, using shading instead of individual points, removing whitespace between multi-panel figures, simplifying the content, etc.
Authors' response:
Following the referee's suggestion, we have moved several figures to the appendix. We now have only 11 figures (from before 21) in the main part of the paper. The font size for several figures was increased.

Referee:
52. Figure 6. Does the "PF" at the end of the HR125LRPF indicate "PROFFAST"? If so, please state clearly in the caption.
Authors' response:

Done

Referee:
53. Figure 17. When plotting vertical profiles in the context of a total column retrieval, it is more natural to plot the vertical axis in linear pressure. This is because it better approximates the mass-weighting that the total column represents. Please change these plots to display them in linear pressure. Also, you show column averaging kernels for TCCON up to 85 SZA, but do not use them above 75 SZA. Why plot the other two SZAs? Furthermore, you do not show the HR125LR, EM27/SUN, Vertex, IRcube AKs, which is important to understand how the differences between the a priori profile and the true profile would affect the total column. Please add the AKs of the LR instruments (if they are basically the same, you can just add one representative set).

Authors' response:
Done, we have replaced the vertical profiles plots plotted against altitude and added the plots plotted against pressure. For the rest of the comments please see our reply to point no. 36 of the technical comments.

Referee:
54. Figure 18. What exactly is being plotted? 10 minute averages or all individual points? If averages, please make this clear in the caption, and describe the number of points being averaged in each averaging period for each instrument.

Authors' response:
These are 5-minute averages. We have added further explanation in the text, see lines 462 – 465: "In order to make the intercomparison, data from each instrument were sorted and all data within a time interval of a 5 min sequence were averaged and associated to the respective start time of the bin. The timestamp of the reference data set (e.g., TCCONmod) was matched with the same timestamp as the other instruments to find the coincident data pairs, which were used for the difference and the correlation calculation."

We have also added further explanation in the text regarding the number of points being averaged, see lines 488 – 493: "Within the period of five minutes, it is possible to average five measurements for the EM27/SUN data set whereas a maximum of only two measurements is possible for the TCCONmod data set. The Vertex70 measurements on 15 May were performed before the instrument modifications. As a result, a high bias relative to the TCCONmod was seen. This bias is not present for the measurements performed after the instrument modification on 28 August. The scatter in the IRcube and Vertex70 is comparable to the TCCONmod due to the averaging of the similar number of measurements within the five minutes time interval.

Referee:
55. Figure 21. If you are comparing data to a reference (in this case, AirCore is the reference), I believe it is customary to put the reference on the x-axis. You have plotted TCCON on the x-axis here.

Authors' response:
We have removed this plot to reduce the number of figures in the paper.

Referee:

56. Tables: In general, some tables could be combined and descriptive text should be in the caption, not embedded in the table (e.g., Table 4).

Authors' response:

We have removed the descriptive text embedded in the tables 4,5,6,7 and 9 and included it in the caption of the table.

[revised manuscript text omitted]

For this reason in 2017, the European Space Agency (ESA) initiated an intercomparison campaign within the project Fiducial Reference Measurements for Ground-Based Infrared Greenhouse Gas observation (FRM4GHG). The campaign was performed in Sodankylä (Finland) with the aim to assess the performance of different spectrometric instruments for remote sensing of atmospheric trace gases and to quantify their performances regarding precise measurements of column-averaged dry-air volume mole fractions of $CO_2$, $CH_4$ and CO. The instruments were deployed at the meteorological observatory Sodankylä where measurements took place between March and October 2017. The remote sensing measurements were complemented by regular AirCore (Karion et al., 2010) launches from the same site. AirCore measurements provide vertical profiles of the target gas concentrations as auxiliary reference data for the column measurements. The performances of the instruments were compared between themselves and to a reference TCCON instrument. The goal of this campaign was the characterisation of less expensive and more portable FTSs to complement TCCON for the establishment of a wider and denser network.

This paper is organised as follows: Section 2 gives a description of the campaign site, lists the details of the instruments taking part in the campaign and their evolution. Section 3 gives a description of the measurement strategy that was used to ensure comparable observations. Section 4 gives the description of the data and its availability. Section 5 gives the campaign results showing the intercomparison results between the TCCON, non-linearity corrected TCCON (TCCONmod) and AirCore data and results using the AirCore profile as a priori for the FTS retrievals. It also gives the intercomparison results between the test instruments with respect to the reference TCCONmod. Section 6 concludes the paper by giving a summary of the results.

**2 Measurements at Sodankylä and campaign instrumentation**

**2.1 Description of the campaign site**

[revised manuscript text omitted]

180    work of the COCCON-PROCEEDS project funded by the European Space Agency (ESA). Column abundances of $CO_2$, $CH_4$, CO, $H_2O$, and $O_2$ were retrieved from the resulting spectra using the PROFFAST retrieval code.  PROFFAST is a code for retrieving trace gas amounts from low-resolution solar absorption spectra. It has been developed on behalf of ESA, in order to provide a source-open and freely available code (without any licensing restrictions) as required by the growing COCCON user community, e.g. for TROPOMI validation work. It is a least-squares fitting algorithm, which adjusts the trace

185    gas amounts by scaling atmospheric a priori profiles. The retrievals are performed on spectra generated with the included PREPROCESS tool. This tool produces spectra out of the measured DC-coupled EM27/SUN interferograms. It includes a

DC correction of the interferogram, a dedicated phase correction scheme for double-sided interferograms and several quality control tests (e.g. testing for the presence of out-of-band artefacts). The lookup table for cross-sections used by PROFFAST is created on the basis of HITRAN spectroscopic line lists: For $H_2O$, $CH_4$, $N_2O$, HITRAN 2008 line lists are used (in case of $H_2O$ including some minor empirical adjustments), for $CO_2$ and CO HITRAN 2012 line lists are used. PROFFAST uses the solar line list compiled by Geoff Toon, JPL, for GGG2014. In contrast to the TCCON GGG2014 processing, the empirical airmass-independent and airmass-dependent post-calibrations are applied species-wise including molecular oxygen. Thereby, the Xair equivalent provided by PROFFAST is on average normalised to unity, while it remains an un-calibrated intermediate result in GGG2014, which calibrates only the Xgas results. The PROFFAST approach of calibrating Xair is transparent for users, as the calibration factors can be directly related to deviations of the spectroscopic band intensities, and gives the user a more sensitive diagnostic tool at hand, as airmass-dependent artefacts in the reported quantity are also reduced. 
[revised manuscript text omitted]
 an optical fibre which was broken on 23 March 2017 was replaced in April 2017 and the measurements resumed again as of 25 April 2017. The first fibre-optic used for the IRcube was an ultra-low OH silica optical fibre from Polymicron Technologies, part FIA8008801100 with a numerical aperture of 0.22 and a core diameter of 800 $\mu$m. Due to a long delivery time of this fibre-optic, a replacement fibre-optic as discussed in section 2.2.4 was ordered and used since the end of April 2017.

The EM27/SUN was operated without any modifications during the whole campaign period. The exact dates of all performed modifications are shown in Table 3.

A total of 10 AirCore launches were performed during the campaign and these were used as an in-situ reference data set to better understand the intercomparison of the remote sensing data. Further details are discussed in section 5.2.1 and section 5.3.

**3.2 Instrument characterisation**

All teams performed a full functionality test of their respective instruments and accessories before shipping and upon arrival at the campaign site in Sodankylä. The functionality test included quality checks as well as performing ILS measurements of the instruments. These measurements serve as reference to check the effects (if any) of transport on the instrumental properties and to ensure nominal operation in case of new set ups. During the campaign all teams performed ILS measurements when possible to monitor the long-term stability of the participating instruments. The modulation efficiency of the TCCON instrument at the maximum OPD was <1.02 with a phase error in the range of $\pm 2$ mrad throughout the year. The modulation efficiency of the EM27/SUN at the maximum OPD was about 1.02 with a phase error in the range between -3 mrad and 1 mrad throughout the year. The modulation efficiency of the Vertex70 before shipping and upon arrival at the Sodankylä site was about 0.935 at 4.5 cm OPD and the phase error was changing between -16 and -36 mrad. The modulation efficiency improved significantly from 0.935 to about 0.973 and the phase error improved to about -13 mrad after the modification of the Vertex70 with the introduction of the additional aperture. The IRcube has a modulation efficiency of about 0.95 with the phase error in the range between -5 and +1.5 mrad. A summary of the ILS properties of the FTS is given in Table 3. The ILS of the LHR was determined by the radio frequency (RF) filter characteristics used to limit the detector bandwidth and hence the spectral resolution of the instrument and is therefore an inherent property of the instrument. A detailed description of the ILS validation of the LHR with $C_2H_4$ gas cell measurements can be found in a technical document by Hoffmann et al. (2017). None of the instruments showed any sign of degradation of the instrumental properties during the whole campaign.

**4 Data description**

The raw measurements (level 0 data) from all participating remote sensing instruments are made publicly available with the DOI https://doi.org/10.18758/71021040 (Sha et al., 2018). The atmospheric concentration of the trace gases (level 2 data) together with the auxiliary data are made publicly available with the DOI https://doi.org/10.18758/71021048 (Sha et al., 2019). All data sets and the  documentation are also made publicly available via the project webpage (http://frm4ghg.aeronomie.be) as well as via the ESA Atmospheric Validation Data Centre (EVDC).

**5 Campaign results**

**5.1 Intercomparison data**

Sodankylä is located within the Arctic Circle therefore solar measurements with sufficiently low SZA are only possible from the beginning of March to the end of October. During the month of September and October we had mostly overcast sky. Only three days of measurements were possible with the TCCON instrument during  this period. However, these measurements were recorded with SZA > 75°.

Based on the measurement capabilities by the individual instruments, the groups were asked to provide some or preferably all of the following parameters: Measurement day/time; ground pressure; total column amounts of $O_2$, $H_2O$, $CO_2$, $CH_4$, CO; and column averaged dry air mole fraction of the gas (Xgas) values for $XCO_2$, $XCH_4$, XCO. Xgas is defined by the following equation:

$$Xgas = \frac{gas_{column,dry}}{O_{2,column,dry}} \times 0.2095 \tag{1}$$

where 0.2095 is the dry air $O_2$ mole fraction.

For the FTIR instruments also  the column averaged dry air mole fraction of dry air (Xair) was submitted. Xair is dependent on the total column amounts of measured oxygen, surface pressure and water vapour column. It is calculated following equation 3 described in Wunch et al. (2015). Xair is a measure of the instrument's performance  and is used by TCCON to examine station-to-station biases. Ideally, the Xair values should be 1  for measurements of total column amounts of oxygen with accurate spectroscopy, surface pressure and water vapour retrievals. Typical Xair values for TCCON measurements are 0.98 which is because of a 2% bias in the $O_2$ spectroscopy. A summary of the data sets and the corresponding retrieval methods is provided in Table 2. The spectrometers used an identical set of ground-pressure data collected at the Sodankylä site for the retrieval. The Xgas values which were calculated using GFIT were scaled to the WMO standards using the calibration factors used by TCCON and as discussed in Wunch et al. (2015). The recent values of the correction factors (airmass dependent correction factor (ADCF) and airmass independent correction factor (AICF)) for the respective gases were taken from Table 4

in Wunch et al. (2015). The scaling factors for the Xgas values which were calculated using PROFFAST for the EM27/SUN are discussed in detail in Frey et al. (2015).

 All interventions performed on the respective instruments and as discussed in section 3.1 are marked in the time-series plots with vertical lines and the colours corresponding to the respective instrument. The dates are given in Table 3. In the following sections the intercomparison results will be shown, the long-term stability will be discussed and cases where clear deviations of the retrieval results from the participating instruments w.r.t. the reference data set are observed will be explained.

**5.2 Detector non-linearity effects**

The reference measurements performed with the Bruker IFS 125HR during the campaign in 2017  are found to be affected by the non-linearity of the InGaAs detector. The non-linearity was identified towards the very end of the campaign in 2017  while checking the interferogram signal measured by the TCCON and comparing it to the  EM27/SUN. The detector non-linearity is dependent on the photon load incident on the detector and influences the Xgas values dependent on the signal strength of the measurements. The non-linearity being a signal dependent function, can be avoided by keeping the signal level within the linear domain of the detector. To test the non-linearity, a metal grid was placed in the parallel light beam at the entrance port to reduce the signal by about 20%. Figure 1 shows two spectra measured with the standard TCCON configuration with no grid (red) and with a grid (black) placed in the parallel light beam. These spectra cover the complete spectral regions measured by the detector and are zoomed in to highlight the signal of the out-of-band spectral regions. The non-linearity effect leads to out-of-band artefacts in the spectrum falsely indicating the presence of energy where the detector is insensitive. The signal between $0 \, \text{cm}^{-1}$ and the lower cutoff of the detector at $4000 \, \text{cm}^{-1}$ as well as the signal between the upper cutoff at about $12000 \, \text{cm}^{-1}$ and the end of the detector bandpass at about $16000 \, \text{cm}^{-1}$ show non-zero values for the no grid case indicating that the measurements performed were affected by the detector non-linearity. However the measurements performed with the reduced intensity by introducing the grid in the parallel beam do not show such high out-of-band intensities. The lower wavenumber out-of-band region shows only noise values and the higher wavenumber region close to the detector bandpass shows values which are higher than the noise but much lower than the signal of the standard measurements. These higher values can be explained by the presence of unintended double passing of the infrared beam in the interferometer that occurs if some radiation is reflected back from the detector system. The presence of the signal, as a result of this double passing, is superimposed on to the non-linearity artifact of the detector in this wavenumber region which makes this spectral region unusable for the determination of non-linearity. The high signal in the out-of-band spectral regions confirms that the TCCON

measurements performed during 2017 are affected by the detector non-linearity. A correction method has been developed based on the method described in Hase (2000, chap. 5), it has been tested and applied to the TCCON data. The results of this are shown in the appendix A. The non-linearity corrected TCCON data are henceforth referred to as TCCONmod in this paper. The AirCore measurements performed during the campaign were used to compare with the TCCON and TCCONmod data sets. These results are discussed further in the next section.

[revised manuscript text omitted]

by the TCCON is < 6000 ppm during the summer period. In comparison, the TCCON site at Darwin, which is a relatively humid site, show maximum measured XH$_2$O of < 10000 ppm during the summer period. The year 2017 was relatively dry where the range of XH$_2$O measured at the Sodankylä site was between 500 and  4500 ppm. A detailed discussion of the bias dependence on the humidity present along the measurement line-of-sight is

685 presented in section F. The results show that the Xgas values derived from the low-resolution instruments during the campaign period showed no dependencies on the humidity along the measurement line-of-sight.

**6   Summary and outlook**

The FRM4GHG campaign was successfully executed by comparing four portable remote sensing instruments against the reference TCCON instrument at the Sodankylä site during the year 2017. The EM27/SUN was set up every day at the

690 ambient temperature and pressure and was operated without configuration changes during the whole campaign. The other low-resolution FTIR and the LHR were operated from inside a dedicated temperature controlled container. The instruments needed optimisation and behaved better with a low bias and a high correlation relative to the TCCON instrument afterwards.

In the course of the campaign not only the Vertex70, IRcube and LHR instruments have been improved but also the TCCON instrument by detecting and correcting non-linearity of the detector response. Detecting this issue by comparison with the

695 EM27/SUN shows the potential of this instrument as a traveling standard for TCCON.

The intercomparison results using AirCore profiles as a priori provided interesting insights to the FTS retrievals, its sensitivity to the resolution and the averaging kernels. The AirCore profiles also showed the differences relative to the TCCON a prioris and the resulting biases in the retrievals of the target species. The Xgas calculated from AirCore and compared to the TCCON and the non-linearity corrected TCCON (TCCONmod) data sets show that the latter data set is a better representation of the

700 true atmospheric state.

The EM27/SUN Xgas biases relative to the TCCONmod data were low for the target species except for the high XCH$_4$ bias during the March–May period which is due to the difference in the sensitivity of the high and low-resolution instruments and the a prioris not matching well with the actual profile shape. The EM27/SUN results include a instrument specific bias correction for XCO$_2$ and XCH$_4$ using scaling factors which has been determined independently prior to this study from

705 long-term intercomparison measurements performed at the Karlsruhe TCCON site. It may be that the scaling factor is not optimal for the current location and is also contributing to the bias. This needs to be verified for comparison measurements performed at other TCCON locations. The EM27/SUN Xgas values show high precision and good correlation relative to the reference data sets.

[revised manuscript text omitted]

Inoue, M., Morino, I., Uchino, O., Nakatsuru, T., Yoshida, Y., Yokota, T., Wunch, D., Wennberg, P. O., Roehl, C. M., Griffith, D. W. T., Velazco, V. A., Deutscher, N. M., Warneke, T., Notholt, J., Robinson, J., Sherlock, V., Hase, F., Blumenstock, T., Rettinger, M., Sussmann, R., Kyrö, E., Kivi, R., Shiomi, K., Kawakami, S., De Mazière, M., Arnold, S. G., Feist, D. G., Barrow, E. A., Barney, J., Dubey, M., Schneider, M., Iraci, L. T., Podolske, J. R., Hillyard, P. W., Machida, T., Sawa, Y., Tsuboi, K., Matsueda, H., Sweeney, C., Tans, P. P., Andrews, A. E., Biraud, S. C., Fukuyama, Y., Pittman, J. V., Kort, E. A., and Tanaka, T.: Bias corrections of GOSAT SWIR $XCO_2$ and $XCH_4$ with TCCON data and their evaluation using aircraft measurement data, Atmospheric Measurement Techniques, 9, 3491–3512, https://doi.org/10.5194/amt-9-3491-2016, https://www.atmos-meas-tech.net/9/3491/2016/, 2016.

Jing, Y., Wang, T., Zhang, P., Chen, L., Xu, N., and Ma, Y.: Global Atmospheric CO2 Concentrations Simulated by GEOS-Chem: Comparison with GOSAT, Carbon Tracker and Ground-Based Measurements, Atmosphere, 9, https://doi.org/10.3390/atmos9050175, https://www.mdpi.com/2073-4433/9/5/175, 2018.

Karion, A., Sweeney, C., Tans, P., and Newberger, T.: AirCore: An Innovative Atmospheric Sampling System, Journal of Atmospheric and Oceanic Technology, 27, 1839–1853, https://doi.org/10.1175/2010JTECHA1448.1, https://doi.org/10.1175/2010JTECHA1448.1, 2010.

Keppel-Aleks, G., Toon, G. C., Wennberg, P. O., and Deutscher, N. M.: Reducing the impact of source brightness fluctuations on spectra obtained by Fourier-transform spectrometry, Appl. Opt., 46, 4774–4779, https://doi.org/10.1364/AO.46.004774, 
[revised manuscript text omitted]
 figure shows the same plots  as mentioned above for  measurements  performed on 28 August 2017 at Sodankylä.

[Figure]

**Figure 5.** Top figure - upper panel: XCH$_4$ plotted for TCCONmod and EM27/SUN retrievals with the TCCON a priori and with a modified a priori (calculated using in-situ, AirCore and TCCON map files; labelled with AC in the end) for measurements performed on 15 May 2017 at Sodankylä. Top figure - lower panel: shows the difference between the two retrievals in absolute units. Bottom figure shows the same plots as mentioned above for measurements performed on 28 August 2017 at Sodankylä.

[Figure]

**Figure 6.** Top figure - upper panel: XCO plotted for TCCONmod and EM27/SUN retrievals with the TCCON a priori and with a modified a priori (calculated using in-situ, AirCore and TCCON map files; labelled with AC in the end) for measurements performed on 15 May 2017 at Sodankylä. Top figure - lower panel: shows the difference between the two retrievals in absolute units. Bottom figure shows the same plots as mentioned above for measurements performed on 28 August 2017 at Sodankylä.

[Figure]

**Figure 7.** Timeseries of XCO$_2$ retrievals for TCCONmod, LHR, Vertex70, IRcube and EM27/SUN using the standard procedure using TCCON a priori for measurements performed at Sodankylä in 2017 (top row panel), the difference of XCO$_2$ time series for each instrument relative to the reference TCCONmod results (second row panel). The correlation plots of XCO$_2$ from LHR, Vertex70, IRcube and EM27/SUN instruments vs TCCONmod for all measurements with SZA < 75° : Third row-left panel: LHR vs TCCONmod; third row-right panel: Vertex70 vs TCCONmod; bottom row-left panel: IRcube vs TCCONmod; bottom row-right panel: EM27/SUN vs TCCONmod measurements. The colours represent the measurements performed during the different months of the year.

[Figure]

**Figure 8.** Timeseries of XCH$_4$ retrievals for TCCONmod, Vertex70, IRcube and EM27/SUN using the standard procedure using TCCON a priori for measurements performed at Sodankylä in 2017 (top row panel) and the difference of XCH$_4$ time series for each instrument relative to the reference TCCONmod results (middle row panel). The correlation plots of XCH$_4$ from Vertex70, IRcube and EM27/SUN instruments vs TCCONmod for all measurements with SZA < 75° : Bottom row-left panel: Vertex70 vs TCCONmod; bottom row-middle panel: IRcube vs TCCONmod; bottom row-right panel: EM27/SUN vs TCCONmod measurements. The colours represent the measurements performed during the different months of the year.

[Figure]

**Figure 9.** Timeseries of XCO retrievals for TCCONmod, Vertex70 and EM27/SUN using the standard procedure using TCCON a priori for measurements performed at Sodankylä in 2017 (top row panel) and the difference of XCO time series for each instrument relative to the reference TCCONmod results (middle row panel). The correlation plots of XCO from Vertex70 and EM27/SUN instruments vs TCCONmod for all measurements with SZA < 75° : Bottom row-left panel: Vertex70 vs TCCONmod; bottom row-right panel: EM27/SUN vs TCCONmod measurements. The colours represent the measurements performed during the different months of the year.

[Figure]

**Figure 10.** Timeseries of Xair for TCCONmod, Vertex70, IRcube and EM27/SUN using the standard procedure using TCCON a priori for measurements performed at Sodankylä in 2017 (top row panel) and the difference of Xair time series for each instrument relative to the reference TCCONmod results (middle row panel). The correlation plots of Xair from Vertex70, IRcube and EM27/SUN instruments vs TCCONmod for all measurements with SZA < 75° : Bottom row-left panel: Vertex70 vs TCCONmod; bottom row-middle panel: IRCUBE vs TCCONmod; bottom row-right panel: EM27/SUN vs TCCONmod measurements. The colours represent the measurements performed during the different months of the year.

[Figure]

**Figure 11.** Top plot: $XCO_2$ bias plotted for each instrument relative to non-linearity corrected TCCON (full year–green triangle, short period–magenta triangle), relative to TCCON (full year–red box, short period–blue box) and relative to HR125LR (full year–grey star, short period–orange star). The correlation coefficient of the respective data set are plotted as half filled circles and correspond to the right side y-axis.The $XCH_4$ and XCO biases for each instrument are plotted in the middle and lower panel plots, respectively. A horizontal dashed line at zero is overlayed on each plot to help in the interpretation of the results.

[Figure]

**Figure A1.** Plot showing the difference of the corrected interferograms - original interferograms vs the original interferograms. The individual corrections are plotted in red, the mean value is plotted as blue open circles and the black line is the fit.

[Figure]

**Figure A2.** Original (top left) and non-linearity corrected (bottom left) spectra; zoom of the out-of-band spectral region (100–3600 cm$^{-1}$) with the original spectra (top right) and non-linearity corrected (bottom right) spectra from the Bruker IFS 125HR at Sodankylä TCCON facility. The colour of the spectrum depends on the interferogram maximum signal at the center burst. The highest values corresponding to the dark red colour are recorded during the noon time when the signal is the highest.

[Figure]

**Figure A3.** Top panel: plot showing the original (red) and non-linearity corrected (black) XCO2 values for one day of measurement performed on 06 September 2017 by the Bruker 125HR Sodankylä TCCON instrument. Middle-panel: shows the difference between the original and the corrected XCO2 values. Lower-panel: shows the relative difference (original - corrected)/corrected in percentage for the XCO2 values.

[Figure]

**Figure A4.** XCO$_2$ plotted for TCCONmod and Vertex70 retrievals with the TCCON a priori and with a modified a priori (calculated using in-situ, AirCore and TCCON map files; labelled with AC in the end) are shown in top row panel and their difference in second row panel for measurements performed on 15 May 2017 at Sodankylä. Third and fourth panel plots show the same plots for 28 August 2017. The same plots for TCCONmod and IRcube retrievals for measurements performed on 15 May 2017 are shown in fifth and sixth row panels and that for the 28 August 2017 are shown in seventh and eight panel plots.

[Figure]

**Figure A5.** XCH$_4$ plotted for TCCONmod and Vertex70 retrievals with the TCCON a priori and with a modified a priori (calculated using in-situ, AirCore and TCCON map files; labelled with AC in the end) are shown in top row panel and their difference in second row panel for measurements performed on 15 May 2017 at Sodankylä. Third and fourth panel plots show the same plots for 28 August 2017. The same plots for TCCONmod and IRcube retrievals for measurements performed on 15 May 2017 are shown in fifth and sixth row panels and that for the 28 August 2017 are shown in seventh and eight panel plots.

[Figure]

**Figure A6.** XCO plotted for TCCONmod and Vertex70 retrievals with the TCCON a priori and with a modified a priori (calculated using in-situ, AirCore and TCCON map files; labelled with AC in the end) are shown in top row panel and their difference in second row panel for measurements performed on 15 May 2017 at Sodankylä. Third and fourth panel plots show the same plots for 28 August 2017.

[revised manuscript text omitted]

Statistics of intercomparison results of LHR, Vertex70, IRcube and EM27/SUN vs non-linearity corrected TCCON for measurements performed in 2017 with SZA < 75°. Species $XCO_2$ / ppm $XCH_4$ / ppm XCO / ppbLHR -18.89±5.34 (0.499) - -VERTEX70 1.46±1.63 (0.984) 0.023±0.013 (0.513) 3.57±2.57 (0.947)IRCUBE -5.02±1.04 (0.971) -0.008±0.004 (0.932) -EM27/SUN -0.73±0.47 (0.996) 0.000±0.004 (0.973) 4.38±1.36 (0.993)LHR -19.02±4.44 (0.462) - -VERTEX70 -0.16±0.57 (0.933) 0.010±0.002 (0.958) 1.34±1.04 (0.991)IRCUBE -5.03±0.81 (0.901) -0.010±0.004 (0.928) -EM27/SUN -0.38±0.39 (0.973) -0.002±0.002 (0.974) 3.98±1.18 (0.988)

AirCore flight performed during the FRM4GHG campaign in 2017 at the SodankyläTCCON site. Flights Date Start time of flight / UTC End time of flight / UTC 1 21/04/2017 07:39:24 08:23:10 2 24/04/2017 15:13:39 16:13:10 3 26/04/2017 09:16:15 10:00:05 4 15/05/2017 09:33:22 10:25:32 5 28/08/2017 09:13:15 10:10:33 6 04/09/2017 09:15:58 10:04:15 7 05/09/2017 09:23:35 10:06:12 8 06/09/2017 09:10:20 09:49:10 9 07/09/2017 08:52:19 09:40:41 10 09/10/2017 09:49:48 10:50:14

Statistics of intercomparison results of AirCore vs TCCON and non-linearity corrected TCCON data sets for measurements performed in 2017 with SZA < 75° for the TCCON measurements. Species $XCO_2$ / ppm $XCH_4$ / ppm XCO / ppbTCCONvsAirCore 0.47±0.66 (0.994) -0.004±0.011 (0.959) 6.40±1.88 (0.950)TCCONmodvsAirCore -0.03±0.71 (0.995) -0.007±0.011 (0.969) 6.25±1.88 (0.951)

---

## Referee Report (RR1)

Second review of Sha et al., Intercomparison of low and high resolution infrared spectrometers for ground-based solar remote sensing measurements of total column concentrations of CO2, CH4 and CO

I would first like to thank the authors for adding a substantial new piece of work in section 5.7. It strengthens the paper to show comparisons with PROFFAST and GGG. However, I have a few concerns about the way in which the results are presented:

1. Several other groups successfully process the CO channel in the EM27/SUNs using the EGI/GGG software. It is unclear what the problem is here, but the authors could reach out to other EGI/GGG users for help in processing the CO channel.

2. You state that "The intraday scatter generated by GFIT is noticeably higher than that achieved with PROFFAST." You cite a private communication with Niki Jacobs to confirm this observation. However, from what I understand, the PROFFAST software typically averages forward and reverse spectra, thus decreasing the scatter. Has this forward/reverse averaging been done here in the PROFFAST retrievals in this comparison? GGG does not typically perform this averaging.

3. You show the xair diurnal variability for the PROFFAST and GFIT retrievals in Fig 13. The GFIT retrievals show what you call a bifurcation. Is this bifurcation between consecutive forward and reverse spectra, or does xair vary smoothly by time of day? If the latter, this could indicate that the ZPD times are listed incorrectly in the GGG runlog. Please check and fix the error. Also in this figure, are forward and reverse scans averaged for the PROFFAST results? It would be appropriate to show (both in Figure 13 and 12) results that have been filtered identically and averaged identically.

---

## Author Response (AR3)

mahesh.sha@aeronomie.be                                          10 July 2020

Letter to the Editor

for manuscript by **Sha et al.**, MS No. **amt-2019-371**; title: "**Intercomparison of low and high resolution infrared spectrometers for ground-based solar remote sensing measurements of total column concentrations of CO₂, CH4 and CO**".

Dear Editor,

We would like to thank you for your positive comments.

Our response to comments from Referee 2 is below.

We hope that our answers are satisfactory and the manuscript can now proceed for publication in the journal "Atmospheric Measurement Techniques".

With kind regards,
Mahesh Kumar Sha (on behalf of all authors)

**Response to comments from Referee 2**

Black: Referee's comments; Blue: Authors' answers

We thank referee #2 for the review and for providing useful feedback.

Referee:
Second review of Sha et al., Intercomparison of low and high resolution infrared spectrometers for ground-based solar remote sensing measurements of total column concentrations of CO2, CH4 and CO

I would first like to thank the authors for adding a substantial new piece of work in section 5.7. It strengthens the paper to show comparisons with PROFFAST and GGG. However, I have a few concerns about the way in which the results are presented:

1. Several other groups successfully process the CO channel in the EM27/SUNs using the EGI/GGG software. It is unclear what the problem is here, but the authors could reach out to other EGI/GGG users for help in processing the CO channel.

Authors' response:
The EM27/SUN has a dual-detector and stores two separated interferograms for each measurement, which are then transformed to two separated spectra covering different

wavenumber regions. University of Bremen did the necessary adaptation in the GFIT setup for performing the CO retrievals from the second detector. The results are included in section 5.7 of the manuscript. We have also included the reference to the EGI/GGG software in the manuscript.

2. You state that "The intraday scatter generated by GFIT is noticeably higher than that achieved with PROFFAST." You cite a private communication with Niki Jacobs to confirm this observation. However, from what I understand, the PROFFAST software typically averages forward and reverse spectra, thus decreasing the scatter. Has this forward/reverse averaging been done here in the PROFFAST retrievals in this comparison? GGG does not typically perform this averaging.

Authors' response:
Yes, the forward / reverse averaging have been done for both cases.
In order to make the intercomparison, data from each setup were sorted and all data within a time interval of a 5 min sequence were averaged and associated to the respective start time of the bin. The timestamp of the reference data set (e.g. TCCONmod) was matched with the same timestamp as the other data set to find the coincident data pairs, which were used for the difference and the correlation calculation.

3. You show the xair diurnal variability for the PROFFAST and GFIT retrievals in Fig 13. The GFIT retrievals show what you call a bifurcation. Is this bifurcation between consecutive forward and reverse spectra, or does xair vary smoothly by time of day? If the latter, this could indicate that the ZPD times are listed incorrectly in the GGG runlog. Please check and fix the error. Also in this figure, are forward and reverse scans averaged for the PROFFAST results? It would be appropriate to show (both in Figure 13 and 12) results that have been filtered identically and averaged identically.

Authors' response:
We have performed a thorough investigation of our GFIT setup and analysis for the EM27/SUN measurements. We did not find the reason for the bifurcation due to the split between forward and reverse spectra. We also checked the ZPD times and did not find any inconsistency. University of Bremen found that the I2S recognised the EM27/SUN interferograms as asymmetrical and not symmetrical during the initial processing. For this reason, the processed spectra had double amount of data points than the processing for symmetrical case. This was further verified by generating a spectrum using OPUS with comparable parameters for performing a Fourier transformation of the same interferogram. The figure below shows the zoom of the three spectrum. The spectrum in red corresponds to the old I2S processing, the spectrum in green corresponds to the new I2S processing and the spectrum in blue corresponds to the spectrum generated with OPUS.

[Figure]

Figure 1: Red: Zoom of spectrum generated from old I2S processing, green: zoom of spectrum generated from new I2S processing and blue: zoom of spectrum generated with OPUS.

The complete time series of the EM27/SUN (EM27SUNGFIT) was reprocessed and the results are updated accordingly in section 5.7, figures 12, 13 and 14 and table 7. We do not see the bifurcation of the Xair values in the newly processed data. The extra points in the old processed spectra is probably the reason for the bifurcation of the Xair values in the old EM27SUNGFIT results.
As mentioned in point 2, the data filtering and averaging for the EM27SUNPF and EM27SUNGFIT is done in an identical way.

[revised manuscript text omitted]

75   only five stations in the southern hemisphere. The lack of stations close to important source areas and the limited number of stations in general is unable to resolve global GHG gradients. Furthermore, for the complete validation of the satellite data set, a denser distribution of ground-based solar absorption measurements is needed to cover geographical gaps and to improve the representativeness of the measurement data for various surface and atmospheric conditions (e.g., high and very low surface albedo, pollution, aerosol presence, humid, dry).

80   An extension of the TCCON network is limited by high start-up, maintenance and operational costs, as well as difficulties of campaign based transportability. The maintenance of the instrument requires skill and experience. All these factors resulted in the development of a number of cheap and easily deployable instruments for remote sensing measurements of greenhouse gases, mainly driven by scientific research institutes in collaboration with industrial partners. Some of these instruments have been in operation for several years. However, there has been little characterisation, intercomparison and harmonisation of these

85   new instruments in comparison to the standard instrument used in TCCON, except for the EM27/SUN were some previous characterisation works are done (Gisi et al., 2012; Frey et al., 2015; Hedelius et al., 2016, 2017; Frey et al., 2019). These

comparisons however are mandatory for using these individual data sets independently for science. The EM27/SUN deployed for this campaign is part of the COllaborative Carbon Column Observing Network (COCCON).

For this reason in 2017, the European Space Agency (ESA) initiated an intercomparison campaign within the project Fiducial Reference Measurements for Ground-Based Infrared Greenhouse Gas observation (FRM4GHG). The campaign was performed in Sodankylä (Finland) with the aim to assess the performance of different spectrometric instruments for remote sensing of atmospheric trace gases and to quantify their performances regarding precise measurements of column-averaged dry-air volume mole fractions of $CO_2$, $CH_4$ and CO. The instruments were deployed at the meteorological observatory Sodankylä where measurements took place between March and October 2017. The remote sensing measurements were complemented by regular AirCore (Karion et al., 2010) launches from the same site. AirCore measurements provide vertical profiles of the target gas concentrations as auxiliary reference data for the column measurements. The performances of the instruments were compared between themselves and to a reference TCCON instrument. The goal of this campaign was the characterisation of less expensive and more portable FTSs to complement TCCON for the establishment of a wider and denser network.

This paper is organised as follows: Section 2 gives a description of the campaign site, lists the details of the instruments taking part in the campaign and their evolution. Section 3 gives a description of the measurement strategy that was used to ensure comparable observations. Section 4 gives the description of the data and its availability. Section 5 gives the campaign results showing the intercomparison results between the TCCON, non-linearity corrected TCCON (TCCONmod) and AirCore data and results using the AirCore profile as a priori for the FTS retrievals. It also gives the intercomparison results between the test instruments with respect to the reference TCCONmod. This section concludes with a presentation of the intercomparison results of EM27/SUN data processed with PROFFAST (COCCON processing chain) and GFIT (TCCON processing suite) highlighting the code dependent biases. Section 6 concludes the paper by giving a summary of the results.

**2 Measurements at Sodankylä and campaign instrumentation**

**2.1 Description of the campaign site**

[revised manuscript text omitted]

**2.2.2 Bruker EM27/SUN**

The EM27/SUN spectrometer has been developed by Karlsruhe Institute of Technology (KIT) in cooperation with Bruker starting in 2011 (Gisi et al., 2012). The spectrometer is available as commercial item from Bruker since 2014, an additional channel for CO detection has been assigned in 2016 (Hase et al., 2016). Today already more than 40 units are operated by working groups around the globe (Frey et al., 2019). The EM27/SUN used during the campaign was provided by KIT. The EM27/SUN records double-sided DC coupled interferograms making an average of 10 scans in about 58 sec at a spectral resolution of 0.5 cm$^{-1}$. A double-sided recording of the interferograms largely reduces the sensitivity to residual phase error. The measurements were performed using a RT InGaAs detector (5500–11000 cm$^{-1}$) and a DC-coupled wavelength extended RT InGaAs detector (4000–5500 cm$^{-1}$) (Hase et al., 2016). In this extended configuration, the EM27/SUN covers the same spectral region as TCCON and encompasses the spectral section as observed by TROPOspheric Monitoring Instrument (TROPOMI) (TRO, a, b). Spectra were generated from raw interferograms using the pre-processor tool developed by KIT in the framework of the COCCON-PROCEEDS project funded by the European Space Agency (ESA). Column abundances of $CO_2$, $CH_4$, CO, $H_2O$, and $O_2$ were retrieved from the resulting spectra using the PROFFAST retrieval code. PROFFAST is a code for retrieving trace gas amounts from low-resolution solar absorption spectra. It has been developed on behalf of ESA, in order to provide a source-open and freely available code (without any licensing restrictions) as required by the growing COCCON user community, e.g. for TROPOMI validation work. It is a least-squares fitting algorithm, which adjusts the trace gas amounts by scaling atmospheric a priori profiles. The retrievals are performed on spectra generated with the included PREPROCESS tool. This tool produces spectra out of the measured DC-coupled EM27/SUN interferograms. It includes a DC correction of the interferogram, a dedicated phase correction scheme for double-sided interferograms and several quality control tests (e.g. testing for the presence of out-of-band artefacts). The lookup table for cross-sections used by PROFFAST is created on the basis of HITRAN spectroscopic line lists: For $H_2O$, $CH_4$, $N_2O$, HITRAN 2008 line lists are used (in case of $H_2O$ including some minor empirical adjustments), for $CO_2$ and CO HITRAN 2012 line lists are used. PROFFAST uses the solar line list compiled by Geoff Toon, JPL, for GGG2014. In contrast to the TCCON GGG2014 processing, the empirical

airmass-independent and airmass-dependent post-calibrations are applied species-wise including molecular oxygen. Thereby, the Xair equivalent provided by PROFFAST is on average normalised to unity, while it remains an un-calibrated intermediate result in GGG2014, which calibrates only the Xgas results. The PROFFAST approach of calibrating Xair is transparent for users, as the calibration factors can be directly related to deviations of the spectroscopic band intensities, and gives the user a more sensitive diagnostic tool at hand, as airmass-dependent artefacts in the reported quantity are also reduced. 
[revised manuscript text omitted]
 an optical fibre which was broken on 23 March 2017 was replaced in April 2017 and the measurements resumed again as of 25 April 2017. The first fibre-optic used for the IRcube was an ultra-low OH silica optical fibre from Polymicron Technologies, part FIA8008801100 with a numerical aperture of

0.22 and a core diameter of 800 $\mu$m. Due to a long delivery time of this fibre-optic, a replacement fibre-optic as discussed in

320   section 2.2.4 was ordered and used since the end of April 2017.

The EM27/SUN was operated without any modifications during the whole campaign period. The exact dates of all performed modifications are shown in Table 3.

A total of 10 AirCore launches were performed during the campaign and these were used as an in-situ reference data set to better understand the intercomparison of the remote sensing data. Further details are discussed in section 5.2.1 and section 5.3.

325   ## 3.2   Instrument characterisation

All teams performed a full functionality test of their respective instruments and accessories before shipping and upon arrival at the campaign site in Sodankylä. The functionality test included quality checks as well as performing ILS measurements of the instruments. These measurements serve as reference to check the effects (if any) of transport on the instrumental properties and to ensure nominal operation in case of new set ups. During the campaign all teams performed ILS measurements when possible

330   to monitor the long-term stability of the participating instruments. The modulation efficiency of the TCCON instrument at the maximum OPD was <1.02 with a phase error in the range of $\pm2$ mrad throughout the year. The modulation efficiency of the EM27/SUN at the maximum OPD was about 1.02 with a phase error in the range between -3 mrad and 1 mrad throughout the year. The modulation efficiency of the Vertex70 before shipping and upon arrival at the Sodankylä site was about 0.935 at 4.5 cm OPD and the phase error was changing between -16 and -36 mrad. The modulation efficiency improved significantly

335   from 0.935 to about 0.973 and the phase error improved to about -13 mrad after the modification of the Vertex70 with the introduction of the additional aperture. The IRcube has a modulation efficiency of about 0.95 with the phase error in the range between -5 and +1.5 mrad. A summary of the ILS properties of the FTS is given in Table 3. The ILS of the LHR was determined by the radio frequency (RF) filter characteristics used to limit the detector bandwidth and hence the spectral resolution of the instrument and is therefore an inherent property of the instrument. A detailed description of the ILS validation of the LHR with

340   $C_2H_4$ gas cell measurements can be found in a technical document by Hoffmann et al. (2017). None of the instruments showed any sign of degradation of the instrumental properties during the whole campaign.

**4   Data description**

The raw measurements (level 0 data) from all participating remote sensing instruments are made publicly available with the DOI https://doi.org/10.18758/71021040 (Sha et al., 2018). The atmospheric concentration of the trace gases (level 2 data)

345   together with the auxiliary data are made publicly available with the DOI https://doi.org/10.18758/71021048 (Sha et al., 2019). All data sets and the documentation are also made publicly available via the project webpage (http://frm4ghg.aeronomie.be) as well as via the ESA Atmospheric Validation Data Centre (EVDC).

**5 Campaign results**

**5.1 Intercomparison data**

350 Sodankylä is located within the Arctic Circle therefore solar measurements with sufficiently low SZA are only possible from the beginning of March to the end of October. During the month of September and October we had mostly overcast sky. Only three days of measurements were possible with the TCCON instrument during this period. However, these measurements were recorded with SZA > 75°.

Based on the measurement capabilities by the individual instruments, the groups were asked to provide some or preferably 355 all of the following parameters: Measurement day/time; ground pressure; total column amounts of $O_2$, $H_2O$, $CO_2$, $CH_4$, CO; and column averaged dry air mole fraction of the gas (Xgas) values for $XCO_2$, $XCH_4$, XCO. Xgas is defined by the following equation:

$$\text{Xgas} = \frac{\text{gas}_{\text{column,dry}}}{O_{2,\text{column,dry}}} \times 0.2095 \qquad (1)$$

where 0.2095 is the dry air $O_2$ mole fraction.

360 For the FTIR instruments also the column averaged dry air mole fraction of dry air (Xair) was submitted. Xair is dependent on the total column amounts of measured oxygen, surface pressure and water vapour column. It is calculated following equation 3 described in Wunch et al. (2015). Xair is a measure of the instrument's performance and is used by TCCON to examine station-to-station biases. Ideally, the Xair values should be 1 for measurements of total column amounts of oxygen with accurate spectroscopy, surface pressure and water vapour retrievals. Typical Xair values for TCCON measurements are 0.98 which is 365 because of a 2% bias in the $O_2$ spectroscopy. A summary of the data sets and the corresponding retrieval methods is provided in Table 2. The spectrometers used an identical set of ground-pressure data collected at the Sodankylä site for the retrieval. The Xgas values which were calculated using GFIT were scaled to the WMO standards using the calibration factors used by TCCON and as discussed in Wunch et al. (2015). The recent values of the correction factors (airmass dependent correction factor (ADCF) and airmass independent correction factor (AICF)) for the respective gases were taken from Table 4 in Wunch 370 et al. (2015). The scaling factors for the Xgas values which were calculated using PROFFAST for the EM27/SUN are discussed in detail in Frey et al. (2015).

All interventions performed on the respective instruments and as discussed in section 3.1 are marked in the time-series plots with vertical lines and the colours corresponding to the respective instrument. The dates are given in Table 3. In the following sections the intercomparison results will be shown, the long-term stability will be discussed and cases where clear deviations 375 of the retrieval results from the participating instruments w.r.t. the reference data set are observed will be explained.

**5.2 Detector non-linearity effects**

The reference measurements performed with the Bruker IFS 125HR during the campaign in 2017 are found to be affected by the non-linearity of the InGaAs detector. The non-linearity was identified towards the very end of the campaign in 2017 while checking the interferogram signal measured by the TCCON and comparing it to the EM27/SUN. The detector non-linearity

380  is dependent on the photon load incident on the detector and influences the Xgas values dependent on the signal strength of the measurements. The non-linearity being a signal dependent function, can be avoided by keeping the signal level within the linear domain of the detector. To test the non-linearity, a metal grid was placed in the parallel light beam at the entrance port to reduce the signal by about 20%. Figure 1 shows two spectra measured with the standard TCCON configuration with no grid (red) and with a grid (black) placed in the parallel light beam. These spectra cover the complete spectral regions measured

385  by the detector and are zoomed in to highlight the signal of the out-of-band spectral regions. The non-linearity effect leads to out-of-band artefacts in the spectrum falsely indicating the presence of energy where the detector is insensitive. The signal between 0 cm$^{-1}$ and the lower cutoff of the detector at 4000 cm$^{-1}$ as well as the signal between the upper cutoff at about 12000 cm$^{-1}$ and the end of the detector bandpass at about 16000 cm$^{-1}$ show non-zero values for the no grid case indicating that the measurements performed were affected by the detector non-linearity. However the measurements performed with

390  the reduced intensity by introducing the grid in the parallel beam do not show such high out-of-band intensities. The lower wavenumber out-of-band region shows only noise values and the higher wavenumber region close to the detector bandpass shows values which are higher than the noise but much lower than the signal of the standard measurements. These higher values can be explained by the presence of unintended double passing of the infrared beam in the interferometer that occurs if some radiation is reflected back from the detector system. The presence of the signal, as a result of this double passing,

395  is superimposed on to the non-linearity artifact of the detector in this wavenumber region which makes this spectral region unusable for the determination of non-linearity. The high signal in the out-of-band spectral regions confirms that the TCCON measurements performed during 2017 are affected by the detector non-linearity. A correction method has been developed based on the method described in Hase (2000, chap. 5), it has been tested and applied to the TCCON data. The results of this are shown in the appendix A. The non-linearity corrected TCCON data are henceforth referred to as TCCONmod in this paper.

400  The AirCore measurements performed during the campaign were used to compare with the TCCON and TCCONmod data sets. These results are discussed further in the next section.

[revised manuscript text omitted]
. We will therefore use the TCCONmod data set as our reference data set for further intercomparison studies in the main section of our paper. However, in the appendix B we also show the intercomparison results of the low-resolution measurements relative to the standard TCCON product which is not yet non-linearity corrected.

460 **5.3   Intercomparison results using AirCore as a priori profile**

[revised manuscript text omitted]

are then used for an intercomparison relative to the TCCON as well as for the intercomparison with other low-resolution test
645  instruments. Further details of the intercomparison results are given in appendix C and D, respectively.

**5.6 Humidity dependencies of bias**

The presence of water vapour lines in the retrieval windows can lead to errors in the determination of the Xgas values unless
they are fitted well in the forward model. It is therefore necessary to check the influence of the water vapour lines for retrievals
performed with the low-resolution instruments. Sodankylä is not the most humid TCCON site. The maximum $XH_2O$ measured
650  by the TCCON is $< 6000$ ppm during the summer period. In comparison, the TCCON site at Darwin, which is a relatively humid
site, show maximum measured $XH_2O$ of $< 10000$ ppm during the summer period. The year 2017 was relatively dry where the
range of $XH_2O$ measured at the Sodankylä site was between 500 and 4500 ppm. A detailed discussion of the bias dependence
on the humidity present along the measurement line-of-sight is presented in section F. The results show that the Xgas values
derived from the low-resolution instruments during the campaign period showed no dependencies on the humidity along the
655  measurement line-of-sight.

**5.7 Intercomparison of EM27/SUN data processed with PROFFAST and GFIT**

So far, the EM27/SUN (COCCON unit) tested in the framework of the campaign has been investigated using the procedures as
recommended by the COCCON network, including the consideration of the individual instrumental line shape (ILS) charac-
terisation and the use of the PREPROCESS and PROFFAST processing chain. This seems appropriate, because otherwise the
660  steps of the established and previously tested procedure for operating the EM27/SUN within COCCON would be skipped.

On the other hand, the separation of instrumental from processing effects provides additional insights and allows to es-
timate the performance of the spectrometer and the processing chain independently. For this purpose, a short comparison
between PREPROCESS & PROFFAST versus the GFIT processing suite as used and validated for TCCON is provided
in this section.
665
 The EM27/SUN GGG interferogram processing suite version
2014 developed by Hedelius and Wennberg (2017) was used for processing the EM27/SUN
data.

The timeseries of $XCO_2$, $XCH_4$ and XCO processed following the COCCON recommendations (labelled: EM27SUNPF)
670  and using GFIT (labelled: EM27SUNGFIT) and their respective differences and correlation are shown in Fig. 12 & 13,
respectively. The biases between the two approaches (EM27SUNPF - EM27SUNGFIT) are listed in the first row of Table 7.
The COCCON (EM27SUNPF) $XCO_2$ is biased low with respect to GFIT by about and the COCCON 0.29 ppm, the
$XCH_4$ is biased low by
18 ppb and the
675  XCO is biased high by about 1.3 ppb. On most days the intraday random variability (or scatter) is similar for both analysis but
the GFIT analysis includes a larger number of outliers from the daily means (Figure 12). No consistent reduction of calibration

biases with respect to TCCON is achieved by applying GFIT instead of PROFFAST. A detailed study of the code differences is needed to understand the differences and is beyond the scope of this paper. Apart from this, the bias in the correlation is very stable between both codes without e.g. noticeable airmass-dependent artefacts or inter-annual drifts. The a-priori profile shapes recommended by TCCON are also applied by COCCON, so the smoothing error should largely cancel out, as both codes predict very similar column sensitivities.

The airmass dependency of Xair retrieved with either code is shown in Fig. 14. The Xair data product is not directly comparable, as PROFFAST applies both an airmass-independent and an airmass-dependent correction on Xair. The Xair values for EM27SUNPF are therefore around one. This is done for exploiting this important diagnostic tool in an optimal manner (while GFIT only calibrates the Xgas products for the target gases) - excursions due to instrumental issues can obviously be detected easier in a calibrated Xair product. Moreover, the definition of Xair in PROFFAST differs from GFIT, as the spectroscopically derived airmass is in the nominator and the pressure derived from the in-situ measurement is in the denominator, which is the opposite of the convention used in GFIT. Therefore, an excursion towards elevated values in PROFFAST is equivalent to a depression in the value reported by GFIT. The comparison between both codes looks plausible, as we find as expected a larger bias and a stronger airmass dependency in the uncalibrated GFIT Xair. The calibration chosen in PROFFAST seems accurate with a slight high bias in the order of 0.2%.

Table 7 presents the biases between the low resolution results achieved with either PROFFAST or GFIT and the TCCON reference (rows 2 and 3). GFIT applied to the EM27/SUN provides a smaller bias in $XCO_2$ (PROFFAST is biased low by about 0.7 ppm, GFIT is biased low by about 0.4 ppm), while it shows a higher bias in $XCH_4$ (GFIT is biased high by 19 ppb, while no detectable bias is found in the PROFFAST data) and smaller bias in XCO (PROFFAST is biased high by about 4.4 ppb, GFIT is biased high by about 3 ppb).

In summary, the code comparison suggests an excellent performance of the COCCON processing chain. We believe that remaining biases with respect to TCCON can be reduced by further careful adjustment of the calibration factors used in PROFFAST. This work of tying COCCON to TCCON has already been taken up and will be based on several COCCON instruments operated near different TCCON stations in order to minimize the impact of residual instrument or station specific biases. We also plan realization of a COCCON travel standard in this context. Based on our results, we recommend using the COCCON workflow for the processing of raw data collected with the EM27/SUN spectrometer.

**6 Summary and outlook**

The FRM4GHG campaign was successfully executed by comparing four portable remote sensing instruments against the reference TCCON instrument at the Sodankylä site during the year 2017. The EM27/SUN was set up every day at the ambient temperature and pressure and was operated without configuration changes during the whole campaign. The other low-resolution FTIR and the LHR were operated from inside a dedicated temperature controlled container. The instruments needed optimisation and behaved better with a low bias and a high correlation relative to the TCCON instrument afterwards.

710  In the course of the campaign not only the Vertex70, IRcube and LHR instruments have been improved but also the TCCON instrument by detecting and correcting non-linearity of the detector response. Detecting this issue by comparison with the EM27/SUN shows the potential of this instrument as a traveling standard for TCCON.

The intercomparison results using AirCore profiles as a priori provided interesting insights to the FTS retrievals, its sensitivity to the resolution and the averaging kernels. The AirCore profiles also showed the differences relative to the TCCON

715  a prioris and the resulting biases in the retrievals of the target species. The Xgas calculated from AirCore and compared to the TCCON and the non-linearity corrected TCCON (TCCONmod) data sets show that the latter data set is a better representation of the true atmospheric state.

The EM27/SUN Xgas biases relative to the TCCONmod data were low for the target species except for the high $XCH_4$ bias during the March–May period which is due to the difference in the sensitivity of the high and low-resolution instruments

720  and the a prioris not matching well with the actual profile shape. The EM27/SUN results include a instrument specific bias correction for $XCO_2$ and $XCH_4$ using scaling factors which has been determined independently prior to this study from long-term intercomparison measurements performed at the Karlsruhe TCCON site. It may be that the scaling factor is not optimal for the current location and is also contributing to the bias. This needs to be verified for comparison measurements performed at other TCCON locations. The EM27/SUN Xgas values show high precision and good correlation relative to the reference

725  data sets.

The IRcube Xgas values show relatively high biases which are related to the possible dependence of the signal level on the extended InGaAs detector known to have non-linearity characteristics. The ILS of the IRcube is also less ideal compared to other larger instruments due to the compact short focal length optics. The impact of the ILS on the biases is being further investigated. However the comparison shows low scatter and a good correlation relative to the TCCONmod data.

730  The Vertex70 was equipped with an extended InGaAs detector which led to identifiable non-linearity effects. The optical path was modified by introducing an aperture stop to avoid saturation and operate in the linear region of the detector to improve the ILS. The bias of the Xgas values, the standard deviation of the difference and the correlation of the modified Vertex70 instrument relative to the TCCONmod data were significantly lower after this instrument modification and comparable to the EM27/SUN results relative to the TCCONmod data.

735  The LHR was a new instrument deployed under test during this campaign. It showed large scatter and large biases with a strong diurnal variation relative to the TCCONmod and other FTS instruments. The LHR data for the 2017 campaign are not yet able to provide meaningful geophysical information. However this comparison has proven to be invaluable to characterise and understand the instrumental biases and possibly the retrieval biases. Both aspects are currently under investigation and improvements are being developed.

740  The intercomparison results showed that the non-linearity corrected TCCON data gave a better match with the low-resolution instruments. The standard deviation of the bias and the correlation coefficient was similar for the target species for the non-linearity corrected TCCON data relative to the standard TCCON data.

The intercomparison results of the EM27/SUN data processed with COCCON processing chain showed excellent performance in comparison to the retrieval results from the GFIT processing suite. We recommend the COCCON workflow for the processing of raw data collected with the EM27/SUN spectrometer.

[revised manuscript text omitted]

Inoue, M., Morino, I., Uchino, O., Nakatsuru, T., Yoshida, Y., Yokota, T., Wunch, D., Wennberg, P. O., Roehl, C. M., Griffith, D. W. T., Velazco, V. A., Deutscher, N. M., Warneke, T., Notholt, J., Robinson, J., Sherlock, V., Hase, F., Blumenstock, T., Rettinger, M., Sussmann, R., Kyrö, E., Kivi, R., Shiomi, K., Kawakami, S., De Mazière, M., Arnold, S. G., Feist, D. G., Barrow, E. A., Barney, J., Dubey, M., Schneider, M., Iraci, L. T., Podolske, J. R., Hillyard, P. W., Machida, T., Sawa, Y., Tsuboi, K., Matsueda, H., Sweeney, C., Tans, P. P., Andrews, A. E., Biraud, S. C., Fukuyama, Y., Pittman, J. V., Kort, E. A., and Tanaka, T.: Bias corrections of GOSAT SWIR $XCO_2$ and $XCH_4$ with TCCON data and their evaluation using aircraft measurement data, Atmospheric Measurement Techniques, 9, 3491–3512, https://doi.org/10.5194/amt-9-3491-2016, https://www.atmos-meas-tech.net/9/3491/2016/, 2016.

Jing, Y., Wang, T., Zhang, P., Chen, L., Xu, N., and Ma, Y.: Global Atmospheric CO2 Concentrations Simulated by GEOS-Chem: Comparison with GOSAT, Carbon Tracker and Ground-Based Measurements, Atmosphere, 9, https://doi.org/10.3390/atmos9050175, https://www.mdpi.com/2073-4433/9/5/175, 2018.

Karion, A., Sweeney, C., Tans, P., and Newberger, T.: AirCore: An Innovative Atmospheric Sampling System, Journal of Atmospheric and Oceanic Technology, 27, 1839–1853, https://doi.org/10.1175/2010JTECHA1448.1, https://doi.org/10.1175/2010JTECHA1448.1, 2010.

Keppel-Aleks, G., Toon, G. C., Wennberg, P. O., and Deutscher, N. M.: Reducing the impact of source brightness fluctuations on spectra obtained by Fourier-transform spectrometry, Appl. Opt., 46, 4774–4779, https://doi.org/10.1364/AO.46.004774, 
[revised manuscript text omitted]
 figure shows the same plots as mentioned above for measurements performed on 28 August 2017 at Sodankylä.

[Figure]

**Figure 5.** Top figure - upper panel: XCH$_4$ plotted for TCCONmod and EM27/SUN retrievals with the TCCON a priori and with a modified a priori (calculated using in-situ, AirCore and TCCON map files; labelled with AC in the end) for measurements performed on 15 May 2017 at Sodankylä. Top figure - lower panel: shows the difference between the two retrievals in absolute units. Bottom figure shows the same plots as mentioned above for measurements performed on 28 August 2017 at Sodankylä.

[Figure]

**Figure 6.** Top figure - upper panel: XCO plotted for TCCONmod and EM27/SUN retrievals with the TCCON a priori and with a modified a priori (calculated using in-situ, AirCore and TCCON map files; labelled with AC in the end) for measurements performed on 15 May 2017 at Sodankylä. Top figure - lower panel: shows the difference between the two retrievals in absolute units. Bottom figure shows the same plots as mentioned above for measurements performed on 28 August 2017 at Sodankylä.

[Figure]

**Figure 7.** Timeseries of $XCO_2$ retrievals for TCCONmod, LHR, Vertex70, IRcube and EM27/SUN using the standard procedure using TCCON a priori for measurements performed at Sodankylä in 2017 (top row panel), the difference of $XCO_2$ time series for each instrument relative to the reference TCCONmod results (second row panel). The correlation plots of $XCO_2$ from LHR, Vertex70, IRcube and EM27/SUN instruments vs TCCONmod for all measurements with SZA < 75° : Third row-left panel: LHR vs TCCONmod; third row-right panel: Vertex70 vs TCCONmod; bottom row-left panel: IRcube vs TCCONmod; bottom row-right panel: EM27/SUN vs TCCONmod measurements. The colours represent the measurements performed during the different months of the year.

[Figure]

**Figure 8.** Timeseries of XCH$_4$ retrievals for TCCONmod, Vertex70, IRcube and EM27/SUN using the standard procedure using TCCON a priori for measurements performed at Sodankylä in 2017 (top row panel) and the difference of XCH$_4$ time series for each instrument relative to the reference TCCONmod results (middle row panel). The correlation plots of XCH$_4$ from Vertex70, IRcube and EM27/SUN instruments vs TCCONmod for all measurements with SZA < 75° : Bottom row-left panel: Vertex70 vs TCCONmod; bottom row-middle panel: IRcube vs TCCONmod; bottom row-right panel: EM27/SUN vs TCCONmod measurements. The colours represent the measurements performed during the different months of the year.

[Figure]

**Figure 9.** Timeseries of XCO retrievals for TCCONmod, Vertex70 and EM27/SUN using the standard procedure using TCCON a priori for measurements performed at Sodankylä in 2017 (top row panel) and the difference of XCO time series for each instrument relative to the reference TCCONmod results (middle row panel). The correlation plots of XCO from Vertex70 and EM27/SUN instruments vs TCCONmod for all measurements with SZA < 75° : Bottom row-left panel: Vertex70 vs TCCONmod; bottom row-right panel: EM27/SUN vs TCCONmod measurements. The colours represent the measurements performed during the different months of the year.

[Figure]

**Figure 10.** Timeseries of Xair for TCCONmod, Vertex70, IRcube and EM27/SUN using the standard procedure using TCCON a priori for measurements performed at Sodankylä in 2017 (top row panel) and the difference of Xair time series for each instrument relative to the reference TCCONmod results (middle row panel). The correlation plots of Xair from Vertex70, IRcube and EM27/SUN instruments vs TCCONmod for all measurements with SZA < 75° : Bottom row-left panel: Vertex70 vs TCCONmod; bottom row-middle panel: IRCUBE vs TCCONmod; bottom row-right panel: EM27/SUN vs TCCONmod measurements. The colours represent the measurements performed during the different months of the year.

[Figure]

**Figure 11.** Top plot: $XCO_2$ bias plotted for each instrument relative to non-linearity corrected TCCON (full year–green triangle, short period–magenta triangle), relative to TCCON (full year–red box, short period–blue box) and relative to HR125LR (full year–grey star, short period–orange star). The correlation coefficient of the respective data set are plotted as half filled circles and correspond to the right side y-axis. The $XCH_4$ and XCO biases for each instrument are plotted in the middle and lower panel plots, respectively. A horizontal dashed line at zero is overlayed on each plot to help in the interpretation of the results.

[Figure]

**Figure 12.** Timeseries of $XCO_2$ (first row panel), $XCH_4$ (third row panel) and XCO (fifth row panel) retrieved from EM27/SUN measurements processed following the COCCON recommendations (EM27SUNPF) and using GFIT (EM27SUNGFIT) for measurements performed at Sodankylä in 2017. The difference of $XCO_2$ (second row panel), $XCH_4$ (fourth row panel)  and XCO (sixth row panel) time series for the EM27SUNPF relative to the EM27SUNGFIT analysis as reference.

[Figure]

**Figure 13.** The correlation plots $XCO_2$ (top-left panel), $XCH_4$ (top-right panel) and XCO (bottom panel) retrieved from EM27/SUN measurements processed following the COCCON recommendations (EM27SUNPF) and using GFIT (EM27SUNGFIT) for measurements performed at Sodankylä in 2017.

[Figure]

**Figure 14.** Xair plotted w.r.t the measurement solar zenith angle for retrievals performed with EM27/SUN data following COCCON recommendations (EM27SUNPF) and using GFIT (EM27SUNGFIT) for measurements performed at Sodankylä in 2017.

[Figure]

**Figure A1.** Plot showing the difference of the corrected interferograms - original interferograms vs the original interferograms. The individual corrections are plotted in red, the mean value is plotted as blue open circles and the black line is the fit.

[Figure]

**Figure A2.** Original (top left) and non-linearity corrected (bottom left) spectra; zoom of the out-of-band spectral region (100–3600 cm$^{-1}$) with the original spectra (top right) and non-linearity corrected (bottom right) spectra from the Bruker IFS 125HR at Sodankylä TCCON facility. The colour of the spectrum depends on the interferogram maximum signal at the center burst. The highest values corresponding to the dark red colour are recorded during the noon time when the signal is the highest.

[Figure]

**Figure A3.** Top panel: plot showing the original (red) and non-linearity corrected (black) XCO2 values for one day of measurement performed on 06 September 2017 by the Bruker 125HR Sodankylä TCCON instrument. Middle-panel: shows the difference between the original and the corrected XCO2 values. Lower-panel: shows the relative difference (original - corrected)/corrected in percentage for the XCO2 values.

[Figure]

**Figure A4.** XCO$_2$ plotted for TCCONmod and Vertex70 retrievals with the TCCON a priori and with a modified a priori (calculated using in-situ, AirCore and TCCON map files; labelled with AC in the end) are shown in top row panel and their difference in second row panel for measurements performed on 15 May 2017 at Sodankylä. Third and fourth panel plots show the same plots for 28 August 2017. The same plots for TCCONmod and IRcube retrievals for measurements performed on 15 May 2017 are shown in fifth and sixth row panels and that for the 28 August 2017 are shown in seventh and eight panel plots.

[Figure]

**Figure A5.** XCH$_4$ plotted for TCCONmod and Vertex70 retrievals with the TCCON a priori and with a modified a priori (calculated using in-situ, AirCore and TCCON map files; labelled with AC in the end) are shown in top row panel and their difference in second row panel for measurements performed on 15 May 2017 at Sodankylä. Third and fourth panel plots show the same plots for 28 August 2017. The same plots for TCCONmod and IRcube retrievals for measurements performed on 15 May 2017 are shown in fifth and sixth row panels and that for the 28 August 2017 are shown in seventh and eight panel plots.

[Figure]

**Figure A6.** XCO plotted for TCCONmod and Vertex70 retrievals with the TCCON a priori and with a modified a priori (calculated using in-situ, AirCore and TCCON map files; labelled with AC in the end) are shown in top row panel and their difference in second row panel for measurements performed on 15 May 2017 at Sodankylä. Third and fourth panel plots show the same plots for 28 August 2017.

[revised manuscript text omitted]

75 only five stations in the southern hemisphere. The lack of stations close to important source areas and the limited number of stations in general is unable to resolve global GHG gradients. Furthermore, for the complete validation of the satellite data set, a denser distribution of ground-based solar absorption measurements is needed to cover geographical gaps and to improve the representativeness of the measurement data for various surface and atmospheric conditions (e.g., high and very low surface albedo, pollution, aerosol presence, humid, dry).

80 An extension of the TCCON network is limited by high start-up, maintenance and operational costs, as well as difficulties of campaign based transportability. The maintenance of the instrument requires skill and experience. All these factors resulted in the development of a number of cheap and easily deployable instruments for remote sensing measurements of greenhouse gases, mainly driven by scientific research institutes in collaboration with industrial partners. Some of these instruments have been in operation for several years. However, there has been little characterisation, intercomparison and harmonisation of these

85 new instruments in comparison to the standard instrument used in TCCON, except for the EM27/SUN were some previous characterisation works are done (Gisi et al., 2012; Frey et al., 2015; Hedelius et al., 2016, 2017; Frey et al., 2019). These

comparisons however are mandatory for using these individual data sets independently for science. The EM27/SUN deployed for this campaign is part of the COllaborative Carbon Column Observing Network (COCCON).

For this reason in 2017, the European Space Agency (ESA) initiated an intercomparison campaign within the project Fiducial Reference Measurements for Ground-Based Infrared Greenhouse Gas observation (FRM4GHG). The campaign was performed in Sodankylä (Finland) with the aim to assess the performance of different spectrometric instruments for remote sensing of atmospheric trace gases and to quantify their performances regarding precise measurements of column-averaged dry-air volume mole fractions of $CO_2$, $CH_4$ and CO. The instruments were deployed at the meteorological observatory Sodankylä where measurements took place between March and October 2017. The remote sensing measurements were complemented by regular AirCore (Karion et al., 2010) launches from the same site. AirCore measurements provide vertical profiles of the target gas concentrations as auxiliary reference data for the column measurements. The performances of the instruments were compared between themselves and to a reference TCCON instrument. The goal of this campaign was the characterisation of less expensive and more portable FTSs to complement TCCON for the establishment of a wider and denser network.

This paper is organised as follows: Section 2 gives a description of the campaign site, lists the details of the instruments taking part in the campaign and their evolution. Section 3 gives a description of the measurement strategy that was used to ensure comparable observations. Section 4 gives the description of the data and its availability. Section 5 gives the campaign results showing the intercomparison results between the TCCON, non-linearity corrected TCCON (TCCONmod) and AirCore data and results using the AirCore profile as a priori for the FTS retrievals. It also gives the intercomparison results between the test instruments with respect to the reference TCCONmod. This section concludes with a presentation of the intercomparison results of EM27/SUN data processed with PROFFAST (COCCON processing chain) and GFIT (TCCON processing suite) highlighting the code dependent biases. Section 6 concludes the paper by giving a summary of the results.

**2 Measurements at Sodankylä and campaign instrumentation**

**2.1 Description of the campaign site**

[revised manuscript text omitted]

**2.2.2 Bruker EM27/SUN**

The EM27/SUN spectrometer has been developed by Karlsruhe Institute of Technology (KIT) in cooperation with Bruker starting in 2011 (Gisi et al., 2012). The spectrometer is available as commercial item from Bruker since 2014, an additional channel for CO detection has been assigned in 2016 (Hase et al., 2016). Today already more than 40 units are operated by working groups around the globe (Frey et al., 2019). The EM27/SUN used during the campaign was provided by KIT. The EM27/SUN records double-sided DC coupled interferograms making an average of 10 scans in about 58 sec at a spectral resolution of 0.5 cm$^{-1}$. A double-sided recording of the interferograms largely reduces the sensitivity to residual phase error. The measurements were performed using a RT InGaAs detector (5500–11000 cm$^{-1}$) and a DC-coupled wavelength extended RT InGaAs detector (4000–5500 cm$^{-1}$) (Hase et al., 2016). In this extended configuration, the EM27/SUN covers the same spectral region as TCCON and encompasses the spectral section as observed by TROPOspheric Monitoring Instrument (TROPOMI) (TRO, a, b). Spectra were generated from raw interferograms using the pre-processor tool developed by KIT in the framework of the COCCON-PROCEEDS project funded by the European Space Agency (ESA). Column abundances of $CO_2$, $CH_4$, CO, $H_2O$, and $O_2$ were retrieved from the resulting spectra using the PROFFAST retrieval code. PROFFAST is a code for retrieving trace gas amounts from low-resolution solar absorption spectra. It has been developed on behalf of ESA, in order to provide a source-open and freely available code (without any licensing restrictions) as required by the growing COCCON user community, e.g. for TROPOMI validation work. It is a least-squares fitting algorithm, which adjusts the trace gas amounts by scaling atmospheric a priori profiles. The retrievals are performed on spectra generated with the included PREPROCESS tool. This tool produces spectra out of the measured DC-coupled EM27/SUN interferograms. It includes a DC correction of the interferogram, a dedicated phase correction scheme for double-sided interferograms and several quality control tests (e.g. testing for the presence of out-of-band artefacts). The lookup table for cross-sections used by PROFFAST is created on the basis of HITRAN spectroscopic line lists: For $H_2O$, $CH_4$, $N_2O$, HITRAN 2008 line lists are used (in case of $H_2O$ including some minor empirical adjustments), for $CO_2$ and CO HITRAN 2012 line lists are used. PROFFAST uses the solar line list compiled by Geoff Toon, JPL, for GGG2014. In contrast to the TCCON GGG2014 processing, the empirical

airmass-independent and airmass-dependent post-calibrations are applied species-wise including molecular oxygen. Thereby, the Xair equivalent provided by PROFFAST is on average normalised to unity, while it remains an un-calibrated intermediate result in GGG2014, which calibrates only the Xgas results. The PROFFAST approach of calibrating Xair is transparent for users, as the calibration factors can be directly related to deviations of the spectroscopic band intensities, and gives the user a more sensitive diagnostic tool at hand, as airmass-dependent artefacts in the reported quantity are also reduced. 
[revised manuscript text omitted]
 an optical fibre which was broken on 23 March 2017 was replaced in April 2017 and the measurements resumed again as of 25 April 2017. The first fibre-optic used for the IRcube was an ultra-low OH silica optical fibre from Polymicron Technologies, part FIA8008801100 with a numerical aperture of

0.22 and a core diameter of 800 $\mu$m. Due to a long delivery time of this fibre-optic, a replacement fibre-optic as discussed in
320 section 2.2.4 was ordered and used since the end of April 2017.

The EM27/SUN was operated without any modifications during the whole campaign period. The exact dates of all performed modifications are shown in Table 3.

A total of 10 AirCore launches were performed during the campaign and these were used as an in-situ reference data set to better understand the intercomparison of the remote sensing data. Further details are discussed in section 5.2.1 and section 5.3.

325 ## 3.2 Instrument characterisation

All teams performed a full functionality test of their respective instruments and accessories before shipping and upon arrival at the campaign site in Sodankylä. The functionality test included quality checks as well as performing ILS measurements of the instruments. These measurements serve as reference to check the effects (if any) of transport on the instrumental properties and to ensure nominal operation in case of new set ups. During the campaign all teams performed ILS measurements when possible
330 to monitor the long-term stability of the participating instruments. The modulation efficiency of the TCCON instrument at the maximum OPD was <1.02 with a phase error in the range of $\pm 2$ mrad throughout the year. The modulation efficiency of the EM27/SUN at the maximum OPD was about 1.02 with a phase error in the range between -3 mrad and 1 mrad throughout the year. The modulation efficiency of the Vertex70 before shipping and upon arrival at the Sodankylä site was about 0.935 at 4.5 cm OPD and the phase error was changing between -16 and -36 mrad. The modulation efficiency improved significantly
335 from 0.935 to about 0.973 and the phase error improved to about -13 mrad after the modification of the Vertex70 with the introduction of the additional aperture. The IRcube has a modulation efficiency of about 0.95 with the phase error in the range between -5 and +1.5 mrad. A summary of the ILS properties of the FTS is given in Table 3. The ILS of the LHR was determined by the radio frequency (RF) filter characteristics used to limit the detector bandwidth and hence the spectral resolution of the instrument and is therefore an inherent property of the instrument. A detailed description of the ILS validation of the LHR with
340 $C_2H_4$ gas cell measurements can be found in a technical document by Hoffmann et al. (2017). None of the instruments showed any sign of degradation of the instrumental properties during the whole campaign.

**4 Data description**

The raw measurements (level 0 data) from all participating remote sensing instruments are made publicly available with the DOI https://doi.org/10.18758/71021040 (Sha et al., 2018). The atmospheric concentration of the trace gases (level 2 data)
345 together with the auxiliary data are made publicly available with the DOI https://doi.org/10.18758/71021048 (Sha et al., 2019). All data sets and the documentation are also made publicly available via the project webpage (http://frm4ghg.aeronomie.be) as well as via the ESA Atmospheric Validation Data Centre (EVDC).

**5 Campaign results**

**5.1 Intercomparison data**

350 Sodankylä is located within the Arctic Circle therefore solar measurements with sufficiently low SZA are only possible from the beginning of March to the end of October. During the month of September and October we had mostly overcast sky. Only three days of measurements were possible with the TCCON instrument during this period. However, these measurements were recorded with SZA > 75°.

Based on the measurement capabilities by the individual instruments, the groups were asked to provide some or preferably 355 all of the following parameters: Measurement day/time; ground pressure; total column amounts of $O_2$, $H_2O$, $CO_2$, $CH_4$, CO; and column averaged dry air mole fraction of the gas (Xgas) values for $XCO_2$, $XCH_4$, XCO. Xgas is defined by the following equation:

$$Xgas = \frac{gas_{column,dry}}{O_{2,column,dry}} \times 0.2095 \tag{1}$$

where 0.2095 is the dry air $O_2$ mole fraction.

360 For the FTIR instruments also the column averaged dry air mole fraction of dry air (Xair) was submitted. Xair is dependent on the total column amounts of measured oxygen, surface pressure and water vapour column. It is calculated following equation 3 described in Wunch et al. (2015). Xair is a measure of the instrument's performance and is used by TCCON to examine station-to-station biases. Ideally, the Xair values should be 1 for measurements of total column amounts of oxygen with accurate spectroscopy, surface pressure and water vapour retrievals. Typical Xair values for TCCON measurements are 0.98 which is 365 because of a 2% bias in the $O_2$ spectroscopy. A summary of the data sets and the corresponding retrieval methods is provided in Table 2. The spectrometers used an identical set of ground-pressure data collected at the Sodankylä site for the retrieval. The Xgas values which were calculated using GFIT were scaled to the WMO standards using the calibration factors used by TCCON and as discussed in Wunch et al. (2015). The recent values of the correction factors (airmass dependent correction factor (ADCF) and airmass independent correction factor (AICF)) for the respective gases were taken from Table 4 in Wunch 370 et al. (2015). The scaling factors for the Xgas values which were calculated using PROFFAST for the EM27/SUN are discussed in detail in Frey et al. (2015).

All interventions performed on the respective instruments and as discussed in section 3.1 are marked in the time-series plots with vertical lines and the colours corresponding to the respective instrument. The dates are given in Table 3. In the following sections the intercomparison results will be shown, the long-term stability will be discussed and cases where clear deviations 375 of the retrieval results from the participating instruments w.r.t. the reference data set are observed will be explained.

**5.2 Detector non-linearity effects**

The reference measurements performed with the Bruker IFS 125HR during the campaign in 2017 are found to be affected by the non-linearity of the InGaAs detector. The non-linearity was identified towards the very end of the campaign in 2017 while checking the interferogram signal measured by the TCCON and comparing it to the EM27/SUN. The detector non-linearity

380 is dependent on the photon load incident on the detector and influences the Xgas values dependent on the signal strength of the measurements. The non-linearity being a signal dependent function, can be avoided by keeping the signal level within the linear domain of the detector. To test the non-linearity, a metal grid was placed in the parallel light beam at the entrance port to reduce the signal by about 20%. Figure 1 shows two spectra measured with the standard TCCON configuration with no grid (red) and with a grid (black) placed in the parallel light beam. These spectra cover the complete spectral regions measured

385 by the detector and are zoomed in to highlight the signal of the out-of-band spectral regions. The non-linearity effect leads to out-of-band artefacts in the spectrum falsely indicating the presence of energy where the detector is insensitive. The signal between 0 cm$^{-1}$ and the lower cutoff of the detector at 4000 cm$^{-1}$ as well as the signal between the upper cutoff at about 12000 cm$^{-1}$ and the end of the detector bandpass at about 16000 cm$^{-1}$ show non-zero values for the no grid case indicating that the measurements performed were affected by the detector non-linearity. However the measurements performed with

390 the reduced intensity by introducing the grid in the parallel beam do not show such high out-of-band intensities. The lower wavenumber out-of-band region shows only noise values and the higher wavenumber region close to the detector bandpass shows values which are higher than the noise but much lower than the signal of the standard measurements. These higher values can be explained by the presence of unintended double passing of the infrared beam in the interferometer that occurs if some radiation is reflected back from the detector system. The presence of the signal, as a result of this double passing,

395 is superimposed on to the non-linearity artifact of the detector in this wavenumber region which makes this spectral region unusable for the determination of non-linearity. The high signal in the out-of-band spectral regions confirms that the TCCON measurements performed during 2017 are affected by the detector non-linearity. A correction method has been developed based on the method described in Hase (2000, chap. 5), it has been tested and applied to the TCCON data. The results of this are shown in the appendix A. The non-linearity corrected TCCON data are henceforth referred to as TCCONmod in this paper.

400 The AirCore measurements performed during the campaign were used to compare with the TCCON and TCCONmod data sets. These results are discussed further in the next section.

[revised manuscript text omitted]
. We will therefore use the TCCONmod data set as our reference data set for further intercomparison studies in the main section of our paper. However, in the appendix B we also show the intercomparison results of the low-resolution measurements relative to the standard TCCON product which is not yet non-linearity corrected.

460 **5.3   Intercomparison results using AirCore as a priori profile**

[revised manuscript text omitted]

are then used for an intercomparison relative to the TCCON as well as for the intercomparison with other low-resolution test instruments. Further details of the intercomparison results are given in appendix C and D, respectively.

**5.6 Humidity dependencies of bias**

The presence of water vapour lines in the retrieval windows can lead to errors in the determination of the Xgas values unless they are fitted well in the forward model. It is therefore necessary to check the influence of the water vapour lines for retrievals performed with the low-resolution instruments. Sodankylä is not the most humid TCCON site. The maximum $XH_2O$ measured by the TCCON is < 6000 ppm during the summer period. In comparison, the TCCON site at Darwin, which is a relatively humid site, show maximum measured $XH_2O$ of < 10000 ppm during the summer period. The year 2017 was relatively dry where the range of $XH_2O$ measured at the Sodankylä site was between 500 and 4500 ppm. A detailed discussion of the bias dependence on the humidity present along the measurement line-of-sight is presented in section F. The results show that the Xgas values derived from the low-resolution instruments during the campaign period showed no dependencies on the humidity along the measurement line-of-sight.

**5.7 Intercomparison of EM27/SUN data processed with PROFFAST and GFIT**

So far, the EM27/SUN (COCCON unit) tested in the framework of the campaign has been investigated using the procedures as recommended by the COCCON network, including the consideration of the individual instrumental line shape (ILS) characterisation and the use of the PREPROCESS and PROFFAST processing chain. This seems appropriate, because otherwise the steps of the established and previously tested procedure for operating the EM27/SUN within COCCON would be skipped.

On the other hand, the separation of instrumental from processing effects provides additional insights and allows to estimate the performance of the spectrometer and the processing chain independently. For this purpose, a short comparison between PREPROCESS & PROFFAST versus the GFIT processing suite as used and validated for TCCON is provided in this section.  The EM27/SUN GGG interferogram processing suite version 2014 developed by Hedelius and Wennberg (2017) was used for processing the EM27/SUN data.

The timeseries of $XCO_2$, $XCH_4$ and XCO processed following the COCCON recommendations (labelled: EM27SUNPF) and using GFIT (labelled: EM27SUNGFIT) and their respective differences and correlation are shown in Fig. 12 & 13, respectively. The biases between the two approaches (EM27SUNPF - EM27SUNGFIT) are listed in the first row of Table 7. The COCCON (EM27SUNPF) $XCO_2$ is biased low with respect to GFIT by about  0.29 ppm, the $XCH_4$ is biased low by 18 ppb and the XCO is biased high by about 1.3 ppb. On most days the intraday random variability (or scatter) is similar for both analysis but the GFIT analysis includes a larger number of outliers from the daily means (Figure 12). No consistent reduction of calibration

biases with respect to TCCON is achieved by applying GFIT instead of PROFFAST. A detailed study of the code differences is needed to understand the differences and is beyond the scope of this paper. Apart from this, the bias in the correlation is very stable between both codes without e.g. noticeable airmass-dependent artefacts or inter-annual drifts. The a-priori profile shapes recommended by TCCON are also applied by COCCON, so the smoothing error should largely cancel out, as both codes predict very similar column sensitivities.

The airmass dependency of Xair retrieved with either code is shown in Fig. 14. The Xair data product is not directly comparable, as PROFFAST applies both an airmass-independent and an airmass-dependent correction on Xair. The Xair values for EM27SUNPF are therefore around one. This is done for exploiting this important diagnostic tool in an optimal manner (while GFIT only calibrates the Xgas products for the target gases) - excursions due to instrumental issues can obviously be detected easier in a calibrated Xair product. Moreover, the definition of Xair in PROFFAST differs from GFIT, as the spectroscopically derived airmass is in the nominator and the pressure derived from the in-situ measurement is in the denominator, which is the opposite of the convention used in GFIT. Therefore, an excursion towards elevated values in PROFFAST is equivalent to a depression in the value reported by GFIT. The comparison between both codes looks plausible, as we find as expected a larger bias and a stronger airmass dependency in the uncalibrated GFIT Xair. The calibration chosen in PROFFAST seems accurate with a slight high bias in the order of 0.2%.

Table 7 presents the biases between the low resolution results achieved with either PROFFAST or GFIT and the TCCON reference (rows 2 and 3). GFIT applied to the EM27/SUN provides a smaller bias in $XCO_2$ (PROFFAST is biased low by about 0.7 ppm, GFIT is biased  low by about 0.4 ppm), while it shows a higher bias in $XCH_4$ (GFIT is biased high by  19 ppb, while no detectable bias is found in the PROFFAST data) and smaller bias in XCO (PROFFAST is biased high by about 4.4 ppb, GFIT is biased high by about 3 ppb).

In summary, the code comparison suggests an excellent performance of the COCCON processing chain. We believe that remaining biases with respect to TCCON can be reduced by further careful adjustment of the calibration factors used in PROFFAST. This work of tying COCCON to TCCON has already been taken up and will be based on several COCCON instruments operated near different TCCON stations in order to minimize the impact of residual instrument or station specific biases. We also plan realization of a COCCON travel standard in this context. Based on our results, we recommend using the COCCON workflow for the processing of raw data collected with the EM27/SUN spectrometer.

**6 Summary and outlook**

The FRM4GHG campaign was successfully executed by comparing four portable remote sensing instruments against the reference TCCON instrument at the Sodankylä site during the year 2017. The EM27/SUN was set up every day at the ambient temperature and pressure and was operated without configuration changes during the whole campaign. The other low-resolution FTIR and the LHR were operated from inside a dedicated temperature controlled container. The instruments needed optimisation and behaved better with a low bias and a high correlation relative to the TCCON instrument afterwards.

In the course of the campaign not only the Vertex70, IRcube and LHR instruments have been improved but also the TCCON instrument by detecting and correcting non-linearity of the detector response. Detecting this issue by comparison with the EM27/SUN shows the potential of this instrument as a traveling standard for TCCON.

The intercomparison results using AirCore profiles as a priori provided interesting insights to the FTS retrievals, its sensitivity to the resolution and the averaging kernels. The AirCore profiles also showed the differences relative to the TCCON a prioris and the resulting biases in the retrievals of the target species. The Xgas calculated from AirCore and compared to the TCCON and the non-linearity corrected TCCON (TCCONmod) data sets show that the latter data set is a better representation of the true atmospheric state.

The EM27/SUN Xgas biases relative to the TCCONmod data were low for the target species except for the high $XCH_4$ bias during the March–May period which is due to the difference in the sensitivity of the high and low-resolution instruments and the a prioris not matching well with the actual profile shape. The EM27/SUN results include a instrument specific bias correction for $XCO_2$ and $XCH_4$ using scaling factors which has been determined independently prior to this study from long-term intercomparison measurements performed at the Karlsruhe TCCON site. It may be that the scaling factor is not optimal for the current location and is also contributing to the bias. This needs to be verified for comparison measurements performed at other TCCON locations. The EM27/SUN Xgas values show high precision and good correlation relative to the reference data sets.

The IRcube Xgas values show relatively high biases which are related to the possible dependence of the signal level on the extended InGaAs detector known to have non-linearity characteristics. The ILS of the IRcube is also less ideal compared to other larger instruments due to the compact short focal length optics. The impact of the ILS on the biases is being further investigated. However the comparison shows low scatter and a good correlation relative to the TCCONmod data.

The Vertex70 was equipped with an extended InGaAs detector which led to identifiable non-linearity effects. The optical path was modified by introducing an aperture stop to avoid saturation and operate in the linear region of the detector to improve the ILS. The bias of the Xgas values, the standard deviation of the difference and the correlation of the modified Vertex70 instrument relative to the TCCONmod data were significantly lower after this instrument modification and comparable to the EM27/SUN results relative to the TCCONmod data.

The LHR was a new instrument deployed under test during this campaign. It showed large scatter and large biases with a strong diurnal variation relative to the TCCONmod and other FTS instruments. The LHR data for the 2017 campaign are not yet able to provide meaningful geophysical information. However this comparison has proven to be invaluable to characterise and understand the instrumental biases and possibly the retrieval biases. Both aspects are currently under investigation and improvements are being developed.

The intercomparison results showed that the non-linearity corrected TCCON data gave a better match with the low-resolution instruments. The standard deviation of the bias and the correlation coefficient was similar for the target species for the non-linearity corrected TCCON data relative to the standard TCCON data.

The intercomparison results of the EM27/SUN data processed with COCCON processing chain showed excellent performance in comparison to the retrieval results from the GFIT processing suite. We recommend the COCCON workflow for the processing of raw data collected with the EM27/SUN spectrometer.

[revised manuscript text omitted]

Inoue, M., Morino, I., Uchino, O., Nakatsuru, T., Yoshida, Y., Yokota, T., Wunch, D., Wennberg, P. O., Roehl, C. M., Griffith, D. W. T., Velazco, V. A., Deutscher, N. M., Warneke, T., Notholt, J., Robinson, J., Sherlock, V., Hase, F., Blumenstock, T., Rettinger, M., Sussmann, R., Kyrö, E., Kivi, R., Shiomi, K., Kawakami, S., De Mazière, M., Arnold, S. G., Feist, D. G., Barrow, E. A., Barney, J., Dubey, M., Schneider, M., Iraci, L. T., Podolske, J. R., Hillyard, P. W., Machida, T., Sawa, Y., Tsuboi, K., Matsueda, H., Sweeney, C., Tans, P. P., Andrews, A. E., Biraud, S. C., Fukuyama, Y., Pittman, J. V., Kort, E. A., and Tanaka, T.: Bias corrections of GOSAT SWIR $XCO_2$ and $XCH_4$ with TCCON data and their evaluation using aircraft measurement data, Atmospheric Measurement Techniques, 9, 3491–3512, https://doi.org/10.5194/amt-9-3491-2016, https://www.atmos-meas-tech.net/9/3491/2016/, 2016.

Jing, Y., Wang, T., Zhang, P., Chen, L., Xu, N., and Ma, Y.: Global Atmospheric CO2 Concentrations Simulated by GEOS-Chem: Comparison with GOSAT, Carbon Tracker and Ground-Based Measurements, Atmosphere, 9, https://doi.org/10.3390/atmos9050175, https://www.mdpi.com/2073-4433/9/5/175, 2018.

Karion, A., Sweeney, C., Tans, P., and Newberger, T.: AirCore: An Innovative Atmospheric Sampling System, Journal of Atmospheric and Oceanic Technology, 27, 1839–1853, https://doi.org/10.1175/2010JTECHA1448.1, https://doi.org/10.1175/2010JTECHA1448.1, 2010.

Keppel-Aleks, G., Toon, G. C., Wennberg, P. O., and Deutscher, N. M.: Reducing the impact of source brightness fluctuations on spectra obtained by Fourier-transform spectrometry, Appl. Opt., 46, 4774–4779, https://doi.org/10.1364/AO.46.004774, 
[revised manuscript text omitted]
 figure shows the same plots as mentioned above for measurements performed on 28 August 2017 at Sodankylä.

[Figure]

**Figure 5.** Top figure - upper panel: XCH$_4$ plotted for TCCONmod and EM27/SUN retrievals with the TCCON a priori and with a modified a priori (calculated using in-situ, AirCore and TCCON map files; labelled with AC in the end) for measurements performed on 15 May 2017 at Sodankylä. Top figure - lower panel: shows the difference between the two retrievals in absolute units. Bottom figure shows the same plots as mentioned above for measurements performed on 28 August 2017 at Sodankylä.

[Figure]

**Figure 6.** Top figure - upper panel: XCO plotted for TCCONmod and EM27/SUN retrievals with the TCCON a priori and with a modified a priori (calculated using in-situ, AirCore and TCCON map files; labelled with AC in the end) for measurements performed on 15 May 2017 at Sodankylä. Top figure - lower panel: shows the difference between the two retrievals in absolute units. Bottom figure shows the same plots as mentioned above for measurements performed on 28 August 2017 at Sodankylä.

[Figure]

**Figure 7.** Timeseries of $XCO_2$ retrievals for TCCONmod, LHR, Vertex70, IRcube and EM27/SUN using the standard procedure using TCCON a priori for measurements performed at Sodankylä in 2017 (top row panel), the difference of $XCO_2$ time series for each instrument relative to the reference TCCONmod results (second row panel). The correlation plots of $XCO_2$ from LHR, Vertex70, IRcube and EM27/SUN instruments vs TCCONmod for all measurements with SZA $< 75°$ : Third row-left panel: LHR vs TCCONmod; third row-right panel: Vertex70 vs TCCONmod; bottom row-left panel: IRcube vs TCCONmod; bottom row-right panel: EM27/SUN vs TCCONmod measurements. The colours represent the measurements performed during the different months of the year.

[Figure]

**Figure 8.** Timeseries of $XCH_4$ retrievals for TCCONmod, Vertex70, IRcube and EM27/SUN using the standard procedure using TCCON a priori for measurements performed at Sodankylä in 2017 (top row panel) and the difference of $XCH_4$ time series for each instrument relative to the reference TCCONmod results (middle row panel). The correlation plots of $XCH_4$ from Vertex70, IRcube and EM27/SUN instruments vs TCCONmod for all measurements with SZA < 75° : Bottom row-left panel: Vertex70 vs TCCONmod; bottom row-middle panel: IRcube vs TCCONmod; bottom row-right panel: EM27/SUN vs TCCONmod measurements. The colours represent the measurements performed during the different months of the year.

[Figure]

**Figure 9.** Timeseries of XCO retrievals for TCCONmod, Vertex70 and EM27/SUN using the standard procedure using TCCON a priori for measurements performed at Sodankylä in 2017 (top row panel) and the difference of XCO time series for each instrument relative to the reference TCCONmod results (middle row panel). The correlation plots of XCO from Vertex70 and EM27/SUN instruments vs TCCONmod for all measurements with SZA < 75° : Bottom row-left panel: Vertex70 vs TCCONmod; bottom row-right panel: EM27/SUN vs TCCONmod measurements. The colours represent the measurements performed during the different months of the year.

[Figure]

**Figure 10.** Timeseries of Xair for TCCONmod, Vertex70, IRcube and EM27/SUN using the standard procedure using TCCON a priori for measurements performed at Sodankylä in 2017 (top row panel) and the difference of Xair time series for each instrument relative to the reference TCCONmod results (middle row panel). The correlation plots of Xair from Vertex70, IRcube and EM27/SUN instruments vs TCCONmod for all measurements with SZA < 75° : Bottom row-left panel: Vertex70 vs TCCONmod; bottom row-middle panel: IRCUBE vs TCCONmod; bottom row-right panel: EM27/SUN vs TCCONmod measurements. The colours represent the measurements performed during the different months of the year.

[Figure]

**Figure 11.** Top plot: XCO$_2$ bias plotted for each instrument relative to non-linearity corrected TCCON (full year–green triangle, short period–magenta triangle), relative to TCCON (full year–red box, short period–blue box) and relative to HR125LR (full year–grey star, short period–orange star). The correlation coefficient of the respective data set are plotted as half filled circles and correspond to the right side y-axis.The XCH$_4$ and XCO biases for each instrument are plotted in the middle and lower panel plots, respectively. A horizontal dashed line at zero is overlayed on each plot to help in the interpretation of the results.

[Figure]

**Figure 12.** Timeseries of $XCO_2$ (first row panel), $XCH_4$ (third row panel) and XCO (fifth row panel) retrieved from EM27/SUN measurements processed following the COCCON recommendations (EM27SUNPF) and using GFIT (EM27SUNGFIT) for measurements performed at Sodankylä in 2017. The difference of $XCO_2$ (second row panel), $XCH_4$ (fourth row panel)  and XCO (sixth row panel) time series for the EM27SUNPF relative to the EM27SUNGFIT analysis as reference.

[Figure]

**Figure 13.** The correlation plots XCO$_2$ (top-left panel), XCH$_4$ (top-right panel) and XCO (bottom panel) retrieved from EM27/SUN measurements processed following the COCCON recommendations (EM27SUNPF) and using GFIT (EM27SUNGFIT) for measurements performed at Sodankylä in 2017.

[Figure]

**Figure 14.** Xair plotted w.r.t the measurement solar zenith angle for retrievals performed with EM27/SUN data following COCCON recommendations (EM27SUNPF) and using GFIT (EM27SUNGFIT) for measurements performed at Sodankylä in 2017.

[Figure]

**Figure A1.** Plot showing the difference of the corrected interferograms - original interferograms vs the original interferograms. The individual corrections are plotted in red, the mean value is plotted as blue open circles and the black line is the fit.

[Figure]

**Figure A2.** Original (top left) and non-linearity corrected (bottom left) spectra; zoom of the out-of-band spectral region (100–3600 cm$^{-1}$) with the original spectra (top right) and non-linearity corrected (bottom right) spectra from the Bruker IFS 125HR at Sodankylä TCCON facility. The colour of the spectrum depends on the interferogram maximum signal at the center burst. The highest values corresponding to the dark red colour are recorded during the noon time when the signal is the highest.

[Figure]

**Figure A3.** Top panel: plot showing the original (red) and non-linearity corrected (black) XCO2 values for one day of measurement performed on 06 September 2017 by the Bruker 125HR Sodankylä TCCON instrument. Middle-panel: shows the difference between the original and the corrected XCO2 values. Lower-panel: shows the relative difference (original - corrected)/corrected in percentage for the XCO2 values.

[Figure]

**Figure A4.** XCO$_2$ plotted for TCCONmod and Vertex70 retrievals with the TCCON a priori and with a modified a priori (calculated using in-situ, AirCore and TCCON map files; labelled with AC in the end) are shown in top row panel and their difference in second row panel for measurements performed on 15 May 2017 at Sodankylä. Third and fourth panel plots show the same plots for 28 August 2017. The same plots for TCCONmod and IRcube retrievals for measurements performed on 15 May 2017 are shown in fifth and sixth row panels and that for the 28 August 2017 are shown in seventh and eight panel plots.

[Figure]

**Figure A5.** XCH$_4$ plotted for TCCONmod and Vertex70 retrievals with the TCCON a priori and with a modified a priori (calculated using in-situ, AirCore and TCCON map files; labelled with AC in the end) are shown in top row panel and their difference in second row panel for measurements performed on 15 May 2017 at Sodankylä. Third and fourth panel plots show the same plots for 28 August 2017. The same plots for TCCONmod and IRcube retrievals for measurements performed on 15 May 2017 are shown in fifth and sixth row panels and that for the 28 August 2017 are shown in seventh and eight panel plots.

[Figure]

**Figure A6.** XCO plotted for TCCONmod and Vertex70 retrievals with the TCCON a priori and with a modified a priori (calculated using in-situ, AirCore and TCCON map files; labelled with AC in the end) are shown in top row panel and their difference in second row panel for measurements performed on 15 May 2017 at Sodankylä. Third and fourth panel plots show the same plots for 28 August 2017.

[revised manuscript text omitted]